# Covered Forest: Fine-grained generalization analysis of graph neural networks

Antonis Vasileiou [1]   Ben Finkelshtein [2]   Floris Geerts [3]   Ron Levie [4]   Christopher Morris [1]

## Abstract

The expressive power of message-passing graph neural networks (MPNNs) is reasonably well understood, primarily through combinatorial techniques from graph isomorphism testing. However, MPNNs' generalization abilities—making meaningful predictions beyond the training set—remain less explored. Current generalization analyses often overlook graph structure, limit the focus to specific aggregation functions, and assume the impractical, hard-to-optimize 0-1 loss function. Here, we extend recent advances in graph similarity theory to assess the influence of graph structure, aggregation, and loss functions on MPNNs' generalization abilities. Our empirical study supports our theoretical insights, improving our understanding of MPNNs' generalization properties.

## 1. Introduction

Graphs represent interactions across life, natural, and formal sciences, such as atomistic systems (Duval et al., 2023; Zhang et al., 2023) or social networks (Easley & Kleinberg, 2010; Lovász, 2012), highlighting the need for machine learning methods for graph data. *Message-passing graph neural networks* (MPNNs) (Gilmer et al., 2017; Scarselli et al., 2009) have recently gained attention, achieving success in areas like drug design (Wong et al., 2023), weather forecasting (Lam et al., 2023), and combinatorial optimization (Cappart et al., 2023; Scavuzzo et al., 2024; Qian et al., 2024).

Despite their success, MPNNs' theoretical properties are underexplored (Morris et al., 2024), with most studies focusing on *expressivity*, i.e., their ability to separate graphs or approximate permutation-invariant graph functions. Ex-

[1]RWTH Aachen University, Germany [2]University of Oxford, UK [3]University of Antwerp, Belgium [4]Technion – Israel Institute of Technology, Israel. Correspondence to: Antonis Vasileiou .

*Proceedings of the $42^{nd}$ International Conference on Machine Learning*, Vancouver, Canada. PMLR 267, 2025. Copyright 2025 by the author(s).

pressivity is analyzed via algorithmic alignment with the 1-*dimensional Weisfeiler–Leman algorithm* (1-WL) (Weisfeiler & Leman, 1968; Weisfeiler, 1976; Morris et al., 2023b), a heuristic for graph isomorphism, or universal approximation theorems (Azizian & Lelarge, 2021; Böker et al., 2023; Chen et al., 2019; Geerts & Reutter, 2022; Maehara & NT, 2019; Rauchwerger et al., 2024). For instance, Morris et al. (2019); Xu et al. (2019) showed that the 1-WL bounds MPNNs' expressivity in distinguishing non-isomorphic graphs, while recent work (Böker et al., 2023; Chen et al., 2022; 2023; Rauchwerger et al., 2024) refines expressivity using advances in graph similarity, such as the *Tree distance* (Böker, 2021).

Understanding when MPNNs generalize to unseen graphs is equally crucial but understudied (Morris et al., 2024). Recent works (Franks et al., 2023; Liao et al., 2021; Morris et al., 2023a; Scarselli et al., 2018) often use *Vapnik–Chervonenkis dimension* (VC dimension) (Vapnik, 1995) or related formalisms to derive generalization bounds based on simplistic graph parameters, e.g., maximum degree. For example, Morris et al. (2023a, Proposition 1 and 2) linked 1-WL expressivity with VC dimension, assuming the 0-1 loss function and specific aggregation functions. Their reliance on 1-WL implies a simplistic notion of *graph similarity*, where graphs are either equivalent or entirely dissimilar, leading to vacuous bounds irrelevant to practice. Moreover, in general, most analyses overlook architectural variations like aggregation functions.

**Present work** Here, we extend modern generalization frameworks based on (data-dependent) *covering numbers* (Xu & Mannor, 2012; Kawaguchi et al., 2022) to address the above shortcomings. Specifically, we investigate how refined notions of graph similarity, or *pseudo-metrics* on graphs, enable smaller graph coverings and tighter analyses of MPNNs' generalization error. See Figure 1 for an illustration of how pseudo-metrics induce distinct geometries, leading to improved generalization analysis. Our analysis also incorporates various architectural choices and loss functions.

Concretely, *(1)* we demonstrate that the choice of pseudo-metric on $n$-order graphs significantly affects generalization analysis. For instance, the *Tree distance* (Böker, 2021) yields a strictly tighter generalization error bound

than Morris et al. (2023a). We provide a general proof technique for this improvement, covering loss functions such as cross-entropy and mean absolute error, unlike Morris et al. (2023a). *(2)* We link our pseudo-metrics to the *Tree Mover's distance* (TMD) (Chuang & Jegelka, 2022), deepening the understanding of TMD, originally defined via a recursive transportation formula. *(3)* We define the 1-MWL, a heuristic for graph isomorphism that characterizes the expressivity of MPNNs with *mean aggregation* and satisfies the Lipschitz property required for our generalization analysis. We also derive a corresponding pseudo-metric on graphs equivalent to 1-MWL in distinguishing non-isomorphic graphs. *(4)* Empirically, our theoretical findings translate into practice, offering nuanced insights into MPNN generalization.

*In summary, our results highlight how refined graph similarity notions improve the understanding of MPNNs' generalization, considering graph structure, architectural choices, and loss functions.*

See Appendix A for a detailed discussion of related work.

## 1.1. Background

In the following, we provide the necessary background on (pseudo-)metric spaces, covering numbers, and MPNNs. We use standard notation for graphs and norms; see Appendix B.

**Metric spaces and coverings** In the remainder of the paper, "distances" between graphs play an essential role, which we make precise by defining a *pseudo-metric* (on the set of graphs). Let $\mathcal{X}$ be a set equipped with a pseudo-metric $d \colon \mathcal{X} \times \mathcal{X} \to \mathbb{R}^+$, i.e., $d$ is a function satisfying $d(x,x) = 0$ and $d(x,y) = d(y,x)$ for $x,y \in \mathcal{X}$, and $d(x,y) \leq d(x,z) + d(z,y)$, for $x,y,z \in \mathcal{X}$. The latter property is called the triangle inequality. The pair $(\mathcal{X}, d)$ is called a *pseudo-metric space*. For $(\mathcal{X}, d)$ to be a *metric space*, $d$ additionally needs to satisfy $d(x,y) = 0 \Rightarrow x = y$, for $x,y \in \mathcal{X}$.[1] See Appendix B for the definition of (Lipschitz/uniform) continuity of a function between pseudometric spaces. Let $(\mathcal{X}, d)$ be a pseudo-metric space. Given an $\varepsilon > 0$, an $\varepsilon$-*cover* of $\mathcal{X}$ is a subset $C \subseteq \mathcal{X}$ such that for all elements $x \in \mathcal{X}$ there is an element $y \in C$ such that $d(x,y) \leq \varepsilon$. Given $\varepsilon > 0$ and a pseudo-metric $d$ on the set $\mathcal{X}$, we define the *covering number* of $\mathcal{X}$, $\mathcal{N}(\mathcal{X}, d, \varepsilon) \coloneqq$

$$\min\{m \mid \text{there is an } \varepsilon\text{-cover of } \mathcal{X} \text{ of cardinality } m\},$$

i.e., the smallest number $m$ such that there exists a $\varepsilon$-cover of cardinality $m$ of the set $\mathcal{X}$ with regard to the pseudo-metric $d$.

The covering number provides a direct way of constructing a partition of $\mathcal{X}$. Let $K \coloneqq \mathcal{N}(\mathcal{X}, d, \varepsilon)$ so that, by definition

---

[1]Observe that computing a metric on the set of graphs $\mathcal{G}$ up to isomorphism is at least as hard as solving the graph isomorphism problem on $\mathcal{G}$.

of the covering number, there is a subset $\{x_1, \ldots, x_K\} \subset \mathcal{X}$ representing an $\varepsilon$-cover of $\mathcal{X}$. We define a partition $\{C_1, \ldots, C_K\}$ where $C_i \coloneqq \{x \in \mathcal{X} \mid d(x, x_i) = \min_{j \in [K]} d(x, x_j)\}$, for $i \in [K]$. To break ties, we take the smallest $i$ in the above. Note that $\mathcal{X} = \bigcup_{i \in [K]} C_i$. The *diameter* of a set is the maximal distance between any two elements in the set. Implied by the definition of an $\varepsilon$-cover and the triangle inequality, each $C_i$ has a diameter of at most $2\varepsilon$.

**Supervised learning on graphs** We define *supervised learning on graphs* as follows. Let $\mathcal{G}$ be the set of all graphs and $\mathcal{Y}$ a set. Unless specified otherwise, $\mathcal{G}$ includes graphs of varying orders with vertex features in arbitrary domains. Later, we restrict $\mathcal{G}$ by bounding graph order or vertex feature domains. Similarly, $\mathcal{Y}$ varies depending on the task, e.g., regression or binary classification. We consider classes $\mathcal{H} \subseteq \mathcal{X}^{\mathcal{G}}$ of *graph embeddings*, i.e., mappings from $\mathcal{G}$ to $\mathcal{X}$ for some set $\mathcal{X}$. A *graph learning algorithm* learns such embeddings from data samples. More formally, let $\mathcal{Z} \coloneqq \mathcal{G} \times \mathcal{Y}$ with a probability distribution $\mu$. A *data sample* $\mathcal{S}$ is a collection of elements from $\mathcal{Z}$, drawn i.i.d. according to $\mu$. We denote $(G, y) \sim \mu$ for an element drawn from $\mathcal{Z}$ under $\mu$, and $\mathcal{S} \sim \mu^k$ for a sample of size $k \in \mathbb{N}$. A *graph learning algorithm* for $\mathcal{H}$ maps each sample $\mathcal{S}$ to a graph embedding $h_{\mathcal{S}} \in \mathcal{H}$. The "goodness" of $h_{\mathcal{S}}$ is assessed using a *loss function* $\ell \colon \mathcal{X} \times \mathcal{Y} \to \mathbb{R}^+$, assumed bounded by some $M > 0$. That is, $|\ell(h(G), y)| \leq M$ for all $h \in \mathcal{H}$, $G \in \mathcal{G}$, and $y \in \mathcal{Y}$. We define the *expected* and *empirical error* as: $\ell_{\exp}(h_{\mathcal{S}}) \coloneqq \mathbb{E}_{(G,y) \sim \mu}\big[\ell(h_{\mathcal{S}}(G), y)\big]$ and $\ell_{\emp}(h_{\mathcal{S}}) \coloneqq \frac{1}{|\mathcal{S}|} \sum_{(G,y) \in \mathcal{S}} \ell(h_{\mathcal{S}}(G), y)$, where $\mathcal{S} \sim \mu^k$ for $k \in \mathbb{N}$. The *generalization error*, $|\ell_{\exp}(h_{\mathcal{S}}) - \ell_{\emp}(h_{\mathcal{S}})|$, is what we aim to bound in this work.

**Message-passing graph neural networks** One particular, well-known class of graph embeddings is MPNNs. Following Gilmer et al. (2017), let $G$ be an attributed graph with initial vertex-feature $\boldsymbol{h}_v^{(0)} \in \mathbb{R}^{d_0}$, $d_0 \in \mathbb{N}$, for $v \in V(G)$. An *MPNN architecture* consists of a stack of $L$ neural network layers for some $L > 0$. In each *layer*, $t \in \mathbb{N}$, we compute a vertex feature $\boldsymbol{h}_v^{(t)} \coloneqq$

$$\mathsf{UPD}^{(t)}\Big(\boldsymbol{h}_v^{(t-1)}, \mathsf{AGG}^{(t)}\big(\{\!\!\{\boldsymbol{h}_u^{(t-1)} \mid u \in N(v)\}\!\!\}\big)\Big) \in \mathbb{R}^{d_t},$$

$d_t \in \mathbb{N}$, for $v \in V(G)$, where $\mathsf{UPD}^{(t)}$ and $\mathsf{AGG}^{(t)}$ are parameterized functions, e.g., neural networks. In the case of graph-level tasks, e.g., graph classification, one uses a *readout*,

$$\boldsymbol{h}_G \coloneqq \mathsf{READOUT}\Big(\{\!\!\{\boldsymbol{h}_v^{(L)} \mid v \in V(G)\}\!\!\}\Big) \in \mathbb{R}^d,$$

to compute a single vectorial representation based on learned vertex features after iteration $L$. Again, READOUT may be a parameterized function.

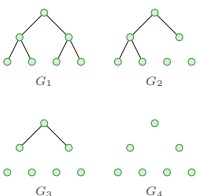 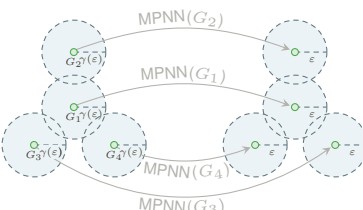 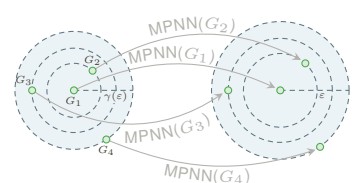

(a) A set of padded binary trees.

(b) Geometry under the trivial 1-WL discrete pseudo-metric. All graphs are equally far apart, not taking into account their similarity, mapping each graph to a unique $\varepsilon$-ball.

(c) Geometry under the Tree distance $\widetilde{\delta}_{\|\cdot\|}$. The similarity of graphs is preserved under MPNNs, leading to a smaller covering and a tighter generalization analysis.

*Figure 1.* Illustrating how the choice of pseudo-metrics influences the geometry and the size of coverings, leading to a tighter generalization analysis.

**Special cases of MPNN layers**   In the following, we discuss special cases of MPNNs, which we will subsequently use in our analyses of MPNNs' generalization abilities. First, for an *unlabeled* $n$-order graph $G$, we assume all initial vertex-features are identical, i.e., $\boldsymbol{h}_v^{(0)} = \boldsymbol{h}_w^{(0)}$, for $v, w \in V(G)$. In this case, we define an *MPNN layer using order-normalized sum aggregation* and *order-normalized readout*, where

$$\boldsymbol{h}_v^{(t)} := \varphi_t\Big(1/|V(G)| \sum_{u \in N(v)} \boldsymbol{h}_u^{(t-1)}\Big), \tag{1}$$

$$\boldsymbol{h}_G := \psi\Big(1/|V(G)| \sum_{u \in V(G)} \boldsymbol{h}_u^{(L)}\Big), \tag{2}$$

for $v \in V(G)$, where $\varphi_t \colon \mathbb{R}^{d_{t-1}} \to \mathbb{R}^{d_t}$ is a $L_{\varphi_t}$-Lipschitz continuous function, with respect to the 2-norm-induced metric, for $d_t \in \mathbb{N}$ and $t \in [L]$, and $\psi \colon \mathbb{R}^{d_L} \to \mathbb{R}^d$ is a $L_\psi$-Lipschitz continuous function.[2] Secondly, for an *attributed* $n$-order graph $(G, a)$, i.e., we set $\boldsymbol{h}_v^{(0)} = a(v)$, for $v \in V(G)$, we define an *MPNN layer using sum aggregation* and *order-normalized readout*, where

$$\boldsymbol{h}_v^{(t)} := \varphi_t\Big(\boldsymbol{h}_v^{(t-1)}\boldsymbol{W}_t^{(1)} + \sum_{u \in N(v)} \boldsymbol{h}_u^{(t-1)}\boldsymbol{W}_t^{(2)}\Big), \tag{3}$$

$$\boldsymbol{h}_G := \psi\Big(1/|V(G)| \sum_{u \in V(G)} \boldsymbol{h}_u^{(L)}\Big), \tag{4}$$

where $\varphi_t$ and $\psi$ are defined as in Equation (1) and $\boldsymbol{W}_t^{(1)}, \boldsymbol{W}_t^{(2)}$ are $d_{t-1} \times d_t$ matrices over $\mathbb{R}$. Under the additional assumption of positive homogeneity of $\varphi_t$, i.e., $\varphi_t(\lambda x) = \lambda \varphi_t(x)$, for $\lambda > 0$, we define an *MPNN layer using mean aggregation* and *order-normalized readout*, where

$$\boldsymbol{h}_v^{(t)} := \varphi_t\Big(\boldsymbol{h}_v^{(t-1)}\boldsymbol{W}_t^{(1)} + 1/|N(v)|\sum_{u \in N(v)} \boldsymbol{h}_u^{(t-1)}\boldsymbol{W}_t^{(2)}\Big), \tag{5}$$

$$\boldsymbol{h}_G := \psi\Big(1/|V(G)| \sum_{u \in V(G)} \boldsymbol{h}_u^{(L)}\Big). \tag{6}$$

---

[2]Implied by Morris et al. (2019), choosing the function $\varphi_t$ appropriately, leads to a 1-WL-equivalent MPNN-layer.

**MPNN classes**   Based on the above three types of MPNN layers, we consider the following classes of MPNNs. Let $\mathcal{G}$ be a class of graphs, we denote the class of all $L$-layer MPNNs following Equation (1) where $\psi$ is represented by a *feed-forward neural network* (FNN) (See Appendix B.1 for a formal definition of FNNs) with Lipschitz constant $L_{\mathsf{FNN}}$ and bounded by $M' \in \mathbb{R}$ by $\mathsf{MPNN}_{L,M',L_{\mathsf{FNN}}}^{\mathrm{ord}}(\mathcal{G})$, i.e., $\mathsf{MPNN}_{L,M',L_{\mathsf{FNN}}}^{\mathrm{ord}}(\mathcal{G}) :=$

$$\Big\{h \colon \mathcal{G} \to \mathbb{R} \;\Big|\; h(G) := \mathsf{FNN}_{\boldsymbol{\theta}} \circ \boldsymbol{h}_G \text{ where } \boldsymbol{\theta} \in \Theta\Big\},$$

where $\Theta$ is the parameter set of FNNs. Analogously, for Equation (3) and Equation (5), we define the classes $\mathsf{MPNN}_{L,M',L_{\mathsf{FNN}}}^{\mathrm{sum}}(\mathcal{G})$ and $\mathsf{MPNN}_{L,M',L_{\mathsf{FNN}}}^{\mathrm{mean}}(\mathcal{G})$, respectively. Finally, we denote the class of all MPNN architectures consisting of $L$ layers with a readout layer represented by an FNN at the $(L+1)$th-layer by $\mathsf{MPNN}_L(\mathcal{G})$.

In Appendix C, we provide the formal definition of the 1-Weisfeiler–Leman algorithm (1-WL), characterizing the distinguishing power of sum aggregation MPNNs (Morris et al., 2019). We also introduce 1-MWL, a variant of 1-WL, characterizing the distinguishing power of *mean aggregation* MPNNs.

## 2. Pseudo-metrics on the set of graphs

We define and study (pseudo-)metrics on the set of graphs, which we later use for a fine-grained generalization analysis of MPNNs; see Appendix D for an extended discussion. These pseudo-metrics are designed to align with MPNNs, reflecting their computational properties. Since the expressiveness of MPNNs is bounded by the 1-WL, we aim to define pseudo-metrics such that two graphs have a distance of 0 if, and only if, they are 1-WL indistinguishable.

Building on Böker (2021), we define the *Tree distance*, as follows. A matrix $\boldsymbol{S} \in [0,1]^{n \times n}$ is a *doubly-stochastic matrix* if $\|\boldsymbol{S}_{i,\cdot}\|_1 = 1$, for $i \in [n]$, and $\|\boldsymbol{S}_{\cdot,j}\|_1 = 1$, for $j \in [n]$. Let $D_n$ denote the set of $n \times n$ doubly-stochastic matrices. Given a matrix norm $\|\cdot\|$, for two graphs $G$ and

$H$ with adjacency matrices $\boldsymbol{A}(G)$ and $\boldsymbol{A}(H)$, respectively, we define the *Tree distance*

$$\delta_{\|\cdot\|}^{\mathcal{T}}(G,H) := \min_{\boldsymbol{S} \in D_n} \|\boldsymbol{A}(G)\boldsymbol{S} - \boldsymbol{S}\boldsymbol{A}(H)\|. \quad (7)$$

Similarly, we extend the above definition to labeled graphs in $\mathcal{G}_{n,d}^{\mathbb{B}}$,[3] resulting in the *labeled Tree distance*, $\widetilde{\delta}_{\|\cdot\|}^{\mathcal{T}}(G,H) :=$

$$\min_{\boldsymbol{S} \in D_n} \|\boldsymbol{A}(G)\boldsymbol{S} - \boldsymbol{S}\boldsymbol{A}(H)\| + \mathrm{Tr}(\boldsymbol{S}^{\mathsf{T}}\boldsymbol{L}(G,H)), \quad (8)$$

where $\boldsymbol{L}(G,H) = [\mathrm{dist}(\ell_G(i), \ell_H(j))]_{i \in V(G), j \in V(H)}$ for some distance function dist between node features (see Appendix D). The following result shows that Equations (7) and (8) are valid pseudo-metrics for many norms.

**Proposition 1.** *For every entry-wise matrix norm or the cut norm $\|\cdot\|$, $\widetilde{\delta}_{\|\cdot\|}^{\mathcal{T}}$ is a pseudo-metric on $\mathcal{G}_{n,d}^{\mathbb{B}}$. Additionally, for two vertex-labeled graphs $G$ and $H$ in $\mathcal{G}_{n,d}^{\mathbb{B}}$, $\widetilde{\delta}_{\|\cdot\|}^{\mathcal{T}}(G,H) = 0$ if, and only, if $G$ and $H$ are 1-WL indistinguishable.* $\square$

Although the Tree distance and its extension to labeled graphs match the 1-WL algorithm in expressivity, they do not consider the number of iterations needed to distinguish two graphs. This limits our ability to provide tighter generalization error bounds for MPNNs with a fixed number of layers. To address this, we define the *Forest distance* on $\mathcal{G}_{n,d}^{\mathbb{R}}$, which matches the expressivity of 1-WL up to $L$ iterations for fixed $L \in \mathbb{N}$.

**Forest distance**  To define the Forest distance, formally, let $(G, a_G)$ and $(H, a_H)$ be attributed graphs of order $n$, where $a_G \colon V(G) \to \mathbb{R}^d \setminus \{\mathbf{o}_{\mathbb{R}^d}\}$ and $a_H \colon V(H) \to \mathbb{R}^d \setminus \{\mathbf{o}_{\mathbb{R}^d}\}$, for $d \in \mathbb{N}$.[4] Given a graph $G$, a node $u \in V(G)$, and a depth $L \in \mathbb{N}$, we define the *unrolling tree* of $u$ at depth $L$ as a rooted tree at $u$ with depth $L$, where each node has as children its neighbors in the graph $G$. This tree is also called the computation tree (see Appendix C for a formal definition). Now, for a fixed $L \in \mathbb{N}$, consider the following multiset of unrolling trees:

$$\mathcal{T}_G^L := \{\!\{\tau(\mathsf{unr}(G,u,L)) \mid u \in V(G)\}\!\}.$$

To account for unrolling trees of different orders, we perform a padding process on all trees in $\mathcal{T}_G^L$. That is, for each vertex of each unrolling tree in the multiset $\mathcal{T}_G^L$, we add children with the label $\mathbf{o}$ until each vertex has exactly $n - 1$ children. Hence, after padding, all trees in $\mathcal{T}_G^L$ will have the exact same structure, though the vertices will have different labels, and we denote their disjoint union, i.e., a

forest, by $F_{G,L}$. We perform the same procedure for the graph $(H, \ell_H)$, resulting in the forest $F_{H,L}$. Note that the forests $F_{G,L}$ and $F_{H,L}$ have the same structure but are possibly non-isomorphic due to their vertex labels not matching. However, we can always find an edge-preserving bijection between $V(F_{G,L})$ and $V(F_{H,L})$. Finally, given a vertex $u \in V(F_{G,L})$, we denote by $l(u) \in [L]$ the level of vertex $u$ in the forest $F_{G,L}$. Now given a weight function $\omega \colon \mathbb{N} \to \mathbb{R}^+$, we define the Forest distance of depth $L$ and weights $\omega$, between $G$ and $H$ as $\mathrm{FD}_{L,\omega}(G,H) :=$

$$\min_{\varphi} \sum_{u \in V(F_{G,L})} \omega(l(u)) \cdot \|a_G(u) - a_H(\varphi(u))\|_2, \quad (9)$$

where the minimum is taken over all edge-preserving bijections $\varphi$ between $V(F_{G,L})$ and $V(F_{H,L})$.[5] When $\omega(l) = 1$, for $l \in \mathbb{N}$, we denote the Forest distance by $\mathrm{FD}_L$. See Figure 2 for an illustration of the distance. The following result shows that $\mathrm{FD}_L^{(\omega)}$ is a valid pseudo-metric.

**Lemma 2.** *For every $\omega \colon \mathbb{N} \to \mathbb{R}^+$, $L \in \mathbb{N}$, the Forest distance $\mathrm{FD}_{L,\omega}$ is a well-defined pseudo-metric on $\mathcal{G}_{n,d}^{\mathbb{R}}$. In addition, for two graphs, $G, H$, $\mathrm{FD}_{L,\omega}(G,H) = 0$ if and only if $G, H$ are 1-WL indistinguishable after $L$ iterations.* $\square$

Now the following result shows that the Forest distance is a simplified version of the TMD defined by Chuang & Jegelka (2022), see Appendix F, providing a streamlined, easy-to-understand definition of the TMD distance while preserving all the essential properties required for our analysis.

**Lemma 3.** *The Forest distance is equivalent to the Tree Mover's distance. That is, for all $n, d, L \in \mathbb{N}, \omega \colon \mathbb{N} \to \mathbb{R}^+$, and for all graphs $G, H \in \mathcal{G}_{n,d}^{\mathbb{R}}$,*

$$\mathrm{TMD}_L^{(\omega)}(G,H) = \mathrm{FD}_{L,\widetilde{\omega}}(G,H),$$

*where $\widetilde{\omega}(n) = \prod_{i=1}^n \omega(i)$, for $i \in \mathbb{N}$.* $\square$

While mathematically equivalent, the Forest distance and TMD have distinct advantages. TMD allows more efficient computation, making it practical for applications, whereas the Forest distance intuitively captures structural differences between graphs.

**Mean Forest distance**  The Forest distance motivates the definition of the *mean-Forest distance*, based on mean unrolling (computation) trees (see Appendix G). This is the first graph pseudo-metric that precisely captures the distinguishing power of mean aggregation MPNNs (see Lemma 29). These MPNNs also satisfy a Lipschitz property regarding the mean-Forest distance (see

---

[3]Note that this pseudometric can also be defined on the space $\mathcal{G}_{n,d}^{\mathbb{R}}$, which includes graphs with continuous vertex features. However, we restrict our proof to Boolean features to establish the equivalence with the 1-WL algorithm.

[4]Without loss of generality, we use the zero vector $\mathbf{o}$ for padding purposes.

[5]In the definition of the Forest distance, we use the Euclidean norm following Chuang & Jegelka (2022). However, we note that any other norm could also be used, as all norms are equivalent in finite-dimensional spaces.

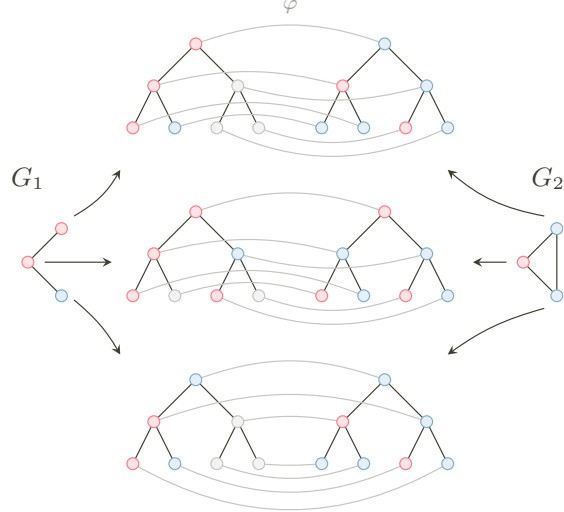

*Figure 2.* An illustration of the computation of the Forest distance for depth $L = 2$ for two labeled graphs $G_1$ and $G_2$. Grey vertices indicate the padded vertices in the unrollings, and $\varphi$ represents an edge-preserving bijection between the two forests.

Lemma 48). We remark that Chuang & Jegelka (2022) (Appendix B.1) introduced the unnormalized tree mover's distance ($\text{TMD}_\omega^{*L}$) and proved its Lipschitz property for mean aggregation MPNNs. However, $\text{TMD}_\omega^{*L}$ is more powerful than the 1-MWL, as it relies on optimal transport between unrolling trees, capturing 1-WL's distinguishing power instead of 1-MWL's.

## 3. Robustness framework

In the following, we introduce the generalization framework for studying the generalization abilities of MPNNs. We refer to Section 1.1 for the formal setup and notation. Slightly modifying the notation from Xu & Mannor (2012), we say that a (graph) learning algorithm for the hypothesis class $\mathcal{H}$, e.g., a class of graph embeddings, on $\mathcal{G} := \mathcal{X} \times \mathcal{Y}$, is $(K, \varepsilon)$-robust, for $K$ and $\varepsilon$, mappings from the set of all possible samples to $\mathbb{N}$ and $\mathbb{R}^+$, respectively, if for all samples $\mathcal{S}$, $\mathcal{Z}$ can be partitioned into $K(\mathcal{S}) > 0$ sets, $\{C_i\}_{i=1}^{K(\mathcal{S})}$, such that the following holds. If $(G, y) \in C_i$, for some $i \in [K]$, then for all $(G', y') \in C_i$,

$$\left| \ell\big(h_\mathcal{S}(G), y\big) - \ell\big(h_\mathcal{S}(G'), y'\big) \right| < \varepsilon(\mathcal{S}),$$

where $h_\mathcal{S}$ is the graph embedding the learning algorithm returns regarding the data sample $\mathcal{S}$. Intuitively, the above definition requires that the difference of the losses of two data points in the same part of the partition is small. Xu & Mannor (2012) showed that a $(K, \epsilon)$-robust learning algorithm implies a bound on the generalization error.

**Theorem 4** (Theorem 3 in Xu & Mannor (2012))**.** *For any* $(K, \epsilon)$-*robust (graph) learning algorithm for* $\mathcal{H}$ *on* $\mathcal{Z}$, *we*

*have that for all* $\delta \in (0,1)$, *with probability at least* $1 - \delta$,

$$|\ell_{\exp}(h_\mathcal{S}) - \ell_{\text{emp}}(h_\mathcal{S})| \leq \epsilon(\mathcal{S}) + M\sqrt{\frac{2K(\mathcal{S})\log(2) + 2\log(1/\delta)}{|\mathcal{S}|}},$$

*where* $h_\mathcal{S}$, *as before, denotes a graph embedding from* $\mathcal{H}$ *returned by the learning algorithm given the data sample* $\mathcal{S}$ *of* $\mathcal{Z}$. *We recall that* $M$ *is the bound on the loss function* $\ell$. $\square$

See Appendix I for the refined results of Kawaguchi et al. (2022) and our extension allowing for different radii within each partition set.

**Connecting robustness, continuity, and expressivity** A natural approach to ensure the robustness of the learning algorithm is to impose continuity assumptions on both the loss functions and the graph embeddings. To formalize this, let us consider two pseudo-metric spaces $(\mathcal{X}, d_\mathcal{X})$ and $(\mathcal{Y}, d_\mathcal{Y})$, and define the pseudo-metric $d_\infty : (\mathcal{X} \times \mathcal{Y}) \times (\mathcal{X} \times \mathcal{Y}) \to \mathbb{R}^+$ as

$$d_\infty((x, y), (x', y')) := \max\{d_\mathcal{X}(x, x'), d_\mathcal{Y}(y, y')\}.$$

Let $\ell : \mathcal{X} \times \mathcal{Y} \to \mathbb{R}^+$ be a $c_\ell$-Lipschitz continuous loss function concerning the metric $d_\infty$ and $(\mathcal{G}, d_\mathcal{G})$ be a pseudo-metric space over a set of graphs. If the covering numbers for the spaces $(\mathcal{G}, d_\mathcal{G})$ and $(\mathcal{Y}, d_\mathcal{Y})$ can be bounded above, then uniform continuity implies robustness, as shown next.

**Proposition 5.** *Let* $(\mathcal{G}, d_\mathcal{G})$, $(\mathcal{X}, d_\mathcal{X})$, *and* $(\mathcal{Y}, d_\mathcal{Y})$ *be pseudo-metric spaces, and let* $\mathcal{H}$ *denote the class of uniformly continuous graph embeddings from* $(\mathcal{G}, d_\mathcal{G})$ *to* $(\mathcal{X}, d_\mathcal{X})$. *If* $\ell : \mathcal{X} \times \mathcal{Y} \to \mathbb{R}^+$ *is a* $c_\ell$-*Lipschitz continuous loss function, regarding* $d_\infty$, *then for any* $\varepsilon > 0$,[6] *a graph learning algorithm for* $\mathcal{H}$ *on* $\mathcal{G} \times \mathcal{Y}$ *is*

$$\big(\mathcal{N}\big(\mathcal{G}, d_\mathcal{G}, \gamma(\varepsilon, \cdot)/2\big) \cdot \mathcal{N}(\mathcal{Y}, d_\mathcal{Y}, \varepsilon/2), c_\ell \varepsilon\big)\text{-robust.}$$

*Here,* $\gamma = \gamma(\varepsilon, h_\mathcal{S})$ *depends on both* $\varepsilon$ *and* $\mathcal{S}$ *is the positive function used in the definition of the uniform continuity of* $h_\mathcal{S} \in \mathcal{H}$. $\square$

**Lipschitzness and equicontinuity of the hypothesis class** By Proposition 5 and Theorem 4, we observe that the generalization bound depends on the function $\gamma$, which, in turn, depends on the specific choice of the MPNN $h_\mathcal{S}$. Ideally, we aim to eliminate this dependency on $\mathcal{S}$ by deriving a uniform bound for all uniformly continuous MPNNs. This can be achieved by showing the existence of a "uniform" choice of $\gamma(\varepsilon)$ that satisfies the uniform continuity definition for all possible MPNNs within a hypothesis class. This property is also known as *equicontinuity* of the hypothesis class.

---

[6]Since $\varepsilon$ is a function in the definition of robustness, we clarify that here $\varepsilon$ refers to a constant function from the set of all possible samples to a positive real number. For simplicity and with a slight abuse of notation, we also denote this constant by $\varepsilon$.

For instance, this holds for all the hypothesis classes of all $c$-Lipschitz MPNNs, sharing the same Lipschitz constant $c$, in which case we can set $\gamma(\varepsilon) = \varepsilon/c$.

**Corollary 6.** *Let $(\mathcal{G}, d_{\mathcal{G}})$, $(\mathcal{X}, d_{\mathcal{X}})$, and $(\mathcal{Y}, d_{\mathcal{Y}})$ be pseudo-metric spaces, and let $\mathcal{H}$ denote the class of $c_{\mathcal{H}}$-Lipschitz continuous graph embeddings from $(\mathcal{G}, d_{\mathcal{G}})$ to $(\mathcal{X}, d_{\mathcal{X}})$. Assume that the loss function is $c_\ell$-Lipschitz, regarding $d_\infty$. Then, graph learning algorithms for $\mathcal{H}$ are $\left(\mathcal{N}\left(\mathcal{G}, d_{\mathcal{G}}, \varepsilon/(2c_{\mathcal{H}})\right) \cdot \mathcal{N}(\mathcal{Y}, d_{\mathcal{Y}}, \varepsilon/2), c_\ell\varepsilon\right)$-robust.* $\square$

As will become clear later, we will use Corollary 6 to derive our generalization bounds in the regression setting, assuming a Lipschitz-continuous loss function. However, for the classification setting, we will specifically use the cross-entropy loss, which is Lipschitz-continuous, concerning only one argument. In this case, we will need a slightly modified version of Corollary 6 to establish our generalization bounds, resulting in the following proposition, which follows from Xu & Mannor (2012, Theorem 14)

**Proposition 7.** *Let $(\mathcal{G}, d_{\mathcal{G}})$, $(\mathcal{X}, d_{\mathcal{X}})$, and $(\mathcal{Y}, d_{\mathcal{Y}})$ be pseudo-metric spaces, let $\mathcal{H}$ be a class of graph embeddings from $(\mathcal{G}, d_{\mathcal{G}})$ to $(\mathcal{X}, d_{\mathcal{X}})$, and let $\ell \colon \mathcal{X} \times \mathcal{Y} \to \mathbb{R}^+$ be a loss function. For a graph learning algorithm for $\mathcal{H}$ on $\mathcal{G} \times \mathcal{Y}$, assume that there exists a positive $\gamma > 0$ such that for all samples $\mathcal{S}$ and $G_1, G_2 \in \mathcal{G}$, $y_1, y_2 \in \mathcal{Y}$, $\max\{d_{\mathcal{G}}(G_1, G_2), d_{\mathcal{Y}}(y_1, y_2)\} \leq \gamma \implies |\ell(h_{\mathcal{S}}(G_1), y_1) - \ell(h_{\mathcal{S}}(G_2), y_2)| < \epsilon(\mathcal{S})$, then, $\mathcal{H}$ is $(\mathcal{N}(\mathcal{Y}, d_{\mathcal{Y}}, \gamma/2) \cdot \mathcal{N}(\mathcal{G}, d_{\mathcal{G}}, \gamma/2), \epsilon)$-robust.* $\square$

See Appendix J.1 for more details on the relationship between robustness and continuity.

## 4. Fine-grained generalization analysis of MPNNs

In this section, we derive the main generalization results of this work using the robustness framework described in Section 3. First, in Appendix L, we recover the VC-dimension bounds from Morris et al. (2023a) within the robustness framework, showing that these bounds can be obtained using a trivial discrete pseudo-metric between graphs, defined as 1 when the graphs are 1-WL, distinguishable and 0 otherwise, leading to vacuous generalization bounds. This motivates us to analyze the impact of alternative pseudo-metrics that yield more fine-grained generalization bounds within the robustness framework compared to the trivial 1-WL-based pseudo-metric. Our analysis considers different aggregation functions, vertex labeling schemes, and loss functions.

We begin by exploring the generalization abilities of MPNN architectures using sum aggregation. Specifically, we consider unlabeled graphs and analyze MPNN layers defined in Equation (1), i.e., order-normalized sum-aggregation MPNNs, using the pseudo-metrics defined in Equation (7),

i.e., the Tree distance. This leads to Proposition 9, Theorem 10, and Proposition 12 (for a class of binary trees). Next, we consider labeled graphs and analyze MPNN layers defined in Equation (3), i.e., sum-aggregation MPNNs, using the pseudo-metrics defined in Equation (9), i.e., the Forest distance. This yields Proposition 15 and Proposition 16. These results can be extended to mean-aggregation MPNNs using the mean-Forest distance, as described in Appendix M.

We first consider unlabeled $n$-order graphs, i.e., every vertex has the same label. We analyze the binary classification setting. Following Section 1.1 and Equation (1), here we investigate the class $\mathsf{MPNN}^{\mathrm{ord}}_{L, M', L_{\mathsf{FNN}}}(\mathcal{G}_n)$. We further consider the *binary cross-entropy loss* $\ell \colon \mathbb{R} \times \{0, 1\} \to \mathbb{R}$, where $\ell(x, y) \coloneqq y\log(\sigma(x)) + (1-y)\log(1-\sigma(x))$, where $\sigma$ is the sigmoid function. It is known that $\ell$ is Lipschitz on the first argument (since it has a bounded first-order derivative) for some Lipschitz constant $L_\ell$.

In Proposition 1 in Section 1.1, we observed that the Tree distance of Equation (7) induces various pseudo-metrics that are equivalent to the 1-WL in terms of expressivity. Utilizing the analysis in Böker et al. (2023), we show that the Tree distance using the cut norm satisfies the uniform continuity property using the same $\gamma(\varepsilon)$ for all MPNNs in $\mathsf{MPNN}^{\mathrm{ord}}_{L, M', L_{\mathsf{FNN}}}(\mathcal{G}_n)$, i.e., it is equicontinuous. Formally, we consider the following variant of the Tree distance on $\mathcal{G}_n$,

$$\delta^{\mathcal{T}}_{\square}(G, H) \coloneqq \min_{\boldsymbol{S} \in D_n} \|\boldsymbol{A}(G)\boldsymbol{S} - \boldsymbol{S}\boldsymbol{A}(H)\|_{\square},$$

where $\|\cdot\|_{\square}$ denotes the *cut norm*, see Equation (11) in Appendix B for details.

The following result shows the equicontinuity property of MPNNs.

**Theorem 8.** *For all $\varepsilon > 0, L \in \mathbb{N}$ there exists $\gamma(\varepsilon) > 0$ such that for $n \in \mathbb{N}, M' \in \mathbb{R}, G, H \in \mathcal{G}_n$ and all MPNN architectures in $\mathsf{MPNN}^{\mathrm{ord}}_{L, M', L_{\mathsf{FNN}}}(\mathcal{G}_n)$, we have*

$$\delta^{\mathcal{T}}_{\square}(G, H) < n^2 \cdot \gamma(\varepsilon) \implies \|\boldsymbol{h}_G - \boldsymbol{h}_H\|_2 \leq \varepsilon. \quad \square$$

The proof is a direct implication of Theorems 29 and 31 in Böker et al. (2023) using the uniform continuity of MPNNs concerning the *Prokhorov* metric (Theorem 29) and then the $\varepsilon$-$\delta$ equivalence between the Prokhorov metric and the Tree distance $\delta^{\mathcal{T}}_{\square}$ (Theorem 31). It is important to note that, in the above result, we proved that all functions in $\mathsf{MPNN}^{\mathrm{ord}}_{L, M', L_{\mathsf{FNN}}}(\mathcal{G}_n)$, are uniformly continuous sharing the same $\gamma(\varepsilon)$, or equivalently the hypothesis class $\mathsf{MPNN}^{\mathrm{ord}}_{L, M', L_{\mathsf{FNN}}}(\mathcal{G}_n)$ satisfies the equicontinuity property. This property is our primary motivation for choosing the tree distance as the pseudo-metric.

Based on the above result, we define the following non-decreasing function $\bar{\gamma} \colon \mathbb{R}^+ \cup \{+\infty\} \to \mathbb{R}^+ \cup \{+\infty\}$

where $\gamma(+\infty) := +\infty$, for all $G, H \in \mathcal{G}_n$, for all $h \in$ $\mathsf{MPNN}^{\mathrm{ord}}_{L,M',L_{\mathsf{FNN}}}(\mathcal{G}_n)$,

$$\bar{\gamma}(\epsilon) := \sup\{\delta > 0 \,|\, \delta^{\mathcal{T}}_{\square}(G,H) \leq n^2\delta \Rightarrow \|h_G - h_H\| \leq \epsilon\},$$

and define its generalized inverse function $\bar{\gamma}^{\leftarrow}: \mathbb{R}^+ \cup \{+\infty\} \to \mathbb{R}^+ \cup \{+\infty\}$, where $\mathbb{R}^+ = (0, +\infty)$, as

$$\bar{\gamma}^{\leftarrow}(y) := \inf\{\epsilon > 0 \mid \gamma(\epsilon) \geq y\}.$$

Note that $\bar{\gamma}^{\leftarrow}$ is also non-decreasing. We further make the convention that $\inf\{\emptyset\} = +\infty$. We now establish the following family of generalization bounds for the class $\mathsf{MPNN}^{\mathrm{ord}}_{L,M',L_{\mathsf{FNN}}}(\mathcal{G}_n)$ based on the results from Proposition 7.

**Proposition 9.** *For $n, L \in \mathbb{N}$, $M' \in \mathbb{R}$, and $\varepsilon > \frac{1}{2}d^*$, where $d^*$ is the minimum non-zero $\delta^{\mathcal{T}}_{\square}$ distance between two graphs in $\mathcal{G}_n$, any graph learning algorithm for the class $\mathsf{MPNN}^{\mathrm{ord}}_{L,M',L_{\mathsf{FNN}}}(\mathcal{G}_n)$ is*

$$\left(2\mathcal{N}(\mathcal{G}_n, \delta^{\mathcal{T}}_{\square}, \varepsilon), L_\ell \cdot L_{\mathsf{FNN}} \cdot \bar{\gamma}^{\leftarrow}\left(\frac{2\varepsilon}{n^2}\right)\right)\text{-robust.}$$

*Hence, for any sample $\mathcal{S}$ and $\delta \in (0,1)$, with probability at least $1 - \delta$,*

$$|\ell_{\exp}(h_{\mathcal{S}}) - \ell_{\mathrm{emp}}(h_{\mathcal{S}})| \leq L_\ell \cdot L_{\mathsf{FNN}} \cdot \bar{\gamma}^{\leftarrow}\left(\frac{2\varepsilon}{n^2}\right)$$
$$+ M\sqrt{\frac{4\mathcal{N}(\mathcal{G}_n, \delta^{\mathcal{T}}_{\square}, \varepsilon)\log(2) + 2\log(1/\delta)}{|\mathcal{S}|}},$$

*where $M$ is an upper bound for the loss function $\ell$ and $L_\ell$ is the Lipschitz constant of $\ell(\cdot, y)$, and $y \in \{0,1\}$.* $\square$

As can be observed by Proposition 9, the tree distance—a more fine-grained metric compared to the trivial 1-$\mathsf{WL}$ metric (which is 1 for 1-$\mathsf{WL}$-distinguishable graphs and 0 otherwise)—enables deriving a wider range of generalization bounds by allowing any real number as the radius for the covering number. Choosing the radius $\varepsilon$ is critical for tight generalization bounds: larger $\varepsilon$ reduces the covering number but increases the first term of the bound, while smaller $\varepsilon$ has the opposite effect. Balancing this trade-off is important, though computing the covering number as a function of $\varepsilon$ is an extremely hard problem. For a given $n$, the covering number can instead be bounded in terms of $\varepsilon$ and $m_n = |\mathcal{G}_n/_{\sim_{\mathsf{WL}}}|$, yielding covering number bounds dependent on $\varepsilon$. For $n$-order graphs $\mathcal{G}_n$ and $\varepsilon = 4k$, a cover of size $m_n/(k+1)$ can be constructed for all $k \in \mathbb{N}$, leading to a family of bounds parametrized by $k$.

**Theorem 10.** *For $n, L \in \mathbb{N}$, and $M' \in \mathbb{R}$, for any graph learning algorithm for the class $\mathsf{MPNN}^{\mathrm{ord}}_{L,M',L_{\mathsf{FNN}}}(\mathcal{G}_n)$ and for any sample $\mathcal{S}$ and $\delta \in (0,1)$, with probability at least*

$1 - \delta$, *we have*

$$|\ell_{\exp}(h_{\mathcal{S}}) - \ell_{\mathrm{emp}}(h_{\mathcal{S}})| \leq L_\ell \cdot L_{\mathsf{FNN}} \cdot \bar{\gamma}^{\leftarrow}\left(\frac{8k}{n^2}\right)$$
$$+ M\sqrt{\frac{4\frac{m_n}{k+1}\log(2) + 2\log\left(\frac{1}{\delta}\right)}{|\mathcal{S}|}}. \tag{10}$$

*for $k \in \mathbb{N}$, where $M$ is an upper bound on the loss function $\ell$, and $m_n = |\mathcal{G}_n/_{\sim_{\mathsf{WL}}}|$* $\square$

In Appendix I in Proposition 36, we combine the extended robustness definition along with Theorem 35 to extend the generalization bounds from Theorem 10 to the class of unlabeled graphs with at most $n$ vertices, denoted as $\mathcal{G}_{\leq n}$. The proof of Proposition 36 is omitted, as it directly follows from the tree construction described in the proof of Theorem 10.

**Tighter bounds on the covering number** The proof of Theorem 10 establishes an upper bound on the covering number, decreasing linearly with the radius. We can use this bound to determine the optimal radius minimizing the generalization bound in Equation (10). However, it remains unclear if this upper bound on the covering number is tight or if the linear decay can be improved. Moreover, the bound $m_n/(k+1)$ relies on the number of equivalence classes $m_n$, which is hard to compute.

To address this, we analyze specific graph classes that yield tighter bounds and simplify the computation of 1-$\mathsf{WL}$-distinguishable graphs. First, consider the class of unordered, unlabeled, full binary trees on $n$ vertices, denoted $\mathcal{T}^{(2)}_n$, where $n = 2j+1$ and $j \in \mathbb{N}$. This class includes rooted trees where each non-root vertex has 0 or 2 children. The root need not be explicitly specified here since it is the only vertex with degree 2; all others have degree 1 or 3. The graphs in $\mathcal{T}^{(2)}_n$ are also called *Otter trees*, named after Richard Otter's work on their enumeration (Otter, 1948). The total number of Otter trees on $n$ vertices, for $n = 2j+1$, is given by the *Wedderburn–Etherington number* $w_j$, with $j \in \mathbb{N}$. Although no closed-form formula exists for $w_j$, recursive methods enable efficient computation. Additionally, the following asymptotic result holds.

**Lemma 11** ((Finch, 2005, p. 295)). *The asymptotic growth of $w_n$ is given by $w_n \sim A \cdot n^{-3/2} \cdot b^n$, for a positive constant $A$ whose precise value is not relevant for our purposes and $b \approx 2.4832$.* $\square$

Based on this lemma, we show that for sufficiently large $n$, we can bound the covering number of the set of Otter trees with a function that decreases *exponentially* regarding the radius of the cover leading. This results in the following (tighter) generalization bounds for MPNNs on the space of Otter trees.

**Proposition 12.** *For $L \in \mathbb{N}, M' \in \mathbb{R}$ and sufficiently large $n \in \mathbb{N}$, for any graph learning algorithm on*

$\mathrm{MPNN}_{L,M',L_{\mathsf{FNN}}}^{\mathrm{ord}}(\mathcal{T}_{2n+1}^{(2)})$ *and for any sample* $\mathcal{S}$, *with* $\delta \in (0,1)$, *with probability at least* $1 - \delta$, *we have*

$$|\ell_{\exp}(h_{\mathcal{S}}) - \ell_{\mathrm{emp}}(h_{\mathcal{S}})| \leq L_\ell \cdot L_{\mathsf{FNN}} \cdot \bar{\gamma}^{\leftarrow}\left(\frac{16k}{(2n+1)^2}\right)$$
$$+ M\sqrt{\frac{4^{w_n/b^{2k}}\log(2) + 2\log(1/\delta)}{|\mathcal{S}|}},$$

*where* $k \in \mathbb{N}$, $M$ *is an upper bound on the loss function* $\ell$, $b \approx 2.4832$, *and* $w_n = |\mathcal{T}_{2n+1}^{(2)}|$. □

Note that using the above bound with a radius mildly dependent on $n$, say $\log_b(n)$, we can derive an upper bound on the covering number that decreases quadratically with $n$.

Additionally, in Appendix N, we derive a graph class (denoted as $\mathcal{F}_n$) that leads to non-constant improvement for a constant $\varepsilon$ and lift the above results to the *regression setting*.

**Vertex-labeled graphs** Finally, in this section, we aim to extend the previous results to the space of discretely-labeled $n$-order graphs, where vertex features are drawn from a finite collection of $d$ elements, i.e., $\mathcal{G}_{n,d}^{\mathbb{B}}$. For ease of notation, we consider one-hot encoding of the labels. Thus, all vertices are initially labeled with a $d$-dimensional vector, having a 1 in one position and 0 elsewhere. Furthermore, we consider the set of graphs with bounded-degree $q$, for $q \in \mathbb{N}$. We denote this set as $\mathcal{G}_{n,d,q}$, and the number of equivalence classes induced by the 1-$\mathsf{WL}$ after $L$ iterations on this set by $m_{n,d,q,L}$, or simply $m_{n,d,q}$ when $L = n - 1$.

To establish our generalization bounds based on the robustness framework, we use the Forest distance $\mathrm{FD}_L$ as defined in Section 2. We begin with the following simple lemma, bounding the Forest distance between two graphs differing by either one edge or one label. Moreover, the following lemma explains the assumption of bounded degree in our graph class. Without this assumption, the bound below would grow with $n$, making it impossible to establish uniform generalization bounds.

**Lemma 13.** *For* $d, q \in \mathbb{N}$, *there exists a constant* $b(d, q, L)$ *such that for* $n \in \mathbb{N}$ *and* $G, H \in \mathcal{G}_{n,d,q}$, *if* $G$ *can be derived by either deleting an edge or changing a single vertex feature of* $H$, *then* $\mathrm{FD}_L(G, H) \leq b(d, q, L)$, *where* $b(d, q, L) = 2q^L \frac{1-q^L}{1-q}$. □

In the following, we analyze the generalization power of $\mathrm{MPNN}_{L,M',L_{\mathsf{FNN}}}^{\mathrm{sum}}(\mathcal{G}_{n,d,q})$. Similar to the case of unlabeled graphs, we consider the cross-entropy function $\ell \colon \mathbb{R} \times \{0,1\} \to \mathbb{R}$, defined by $\ell(x, y) = y\log(\sigma(x)) + (1 - y)\log(1 - \sigma(x))$, where $\sigma(x)$ is the sigmoid function. As with unlabeled graphs, our generalization analysis requires MPNNs to be Lipschitz continuous with respect to the chosen pseudo-metric. The following result demonstrates the Lipschitz continuity prop-

erty of MPNNs in $\mathrm{MPNN}_{L,M',L_{\mathsf{FNN}}}^{\mathrm{sum}}(\mathcal{G}_{n,d,q})$ concerning the Forest distance. While our analysis is restricted to $\mathcal{G}_{n,d,q}$, we state and prove the following result in the more general setting of graphs in $\mathcal{G}_{n,d}^{\mathbb{R}}$, i.e., with real-valued vertex features.

**Lemma 14.** *For* $L, n, d \in \mathbb{N}$, $M' \in \mathbb{R}$ *and for all MPNNs in* $\mathrm{MPNN}_{L,M',L_{\mathsf{FNN}}}^{\mathrm{sum}}(\mathcal{G}_{n,d}^{\mathbb{R}})$, *we have*

$$\|\boldsymbol{h}_G - \boldsymbol{h}_H\|_2 \leq \frac{1}{n}C(L)L_\psi \prod_{i=1}^{L} L_{\varphi_i}\mathrm{FD}_L(G, H),$$

*for* $G, H \in \mathcal{G}_{n,d}^{\mathbb{R}}$, *where* $C(L)$ *is a constant that depends on* $L$ *and the Lipschitz constants of the MPNN layers.* □

We are now ready to present the generalization bound extension for labeled graphs.

**Proposition 15.** *For* $\varepsilon > 0$ *and* $n, L, d, q \in \mathbb{N}$, $M' \in \mathbb{R}$, *any graph learning algorithm for* $\mathrm{MPNN}_{L,M',L_{\mathsf{FNN}}}^{\mathrm{sum}}(\mathcal{G}_{n,d,q})$ *is*

$$(2\mathcal{N}(\mathcal{G}_{n,d,q}, \mathrm{FD}_L, \varepsilon), L_\ell \cdot L_{\mathsf{FNN}} \cdot C_{\mathrm{FD}_L} \cdot 2\varepsilon)\text{-robust.}$$

*Hence, we have the following generalization bounds. For any sample* $\mathcal{S}$ *and* $\delta \in (0,1)$, *with probability at least* $1 - \delta$,

$$|\ell_{\exp}(h_{\mathcal{S}}) - \ell_{\mathrm{emp}}(h_{\mathcal{S}})| \leq \widetilde{C}\varepsilon$$
$$+ M\sqrt{\frac{\mathcal{N}(\mathcal{G}_{n,d,q}, \mathrm{FD}_L, \varepsilon)4\log(2) + 2\log(1/\delta)}{|\mathcal{S}|}},$$

*for* $\varepsilon > 0$, *where* $\widetilde{C} = 2/nL_\ell L_{\mathsf{FNN}}C_{\mathrm{FD}_L}$, $C_{\mathrm{FD}_L} = C(L)L_\psi \prod_{i=1}^{L} L_{\varphi_i}$ *is the Lipschitz constant in Lemma 14 and* $M$ *is an upper bound of the loss function* $\ell$. □

Using a similar construction as Theorem 10, below, we upper bound the covering number in the generalization bound by a function of the corresponding radius and $m_{n,d,q,L}$, leading to the following generalization bound.

**Proposition 16.** *For* $n, q, d, L \in \mathbb{N}$, *and* $M' \in \mathbb{R}$, *for any graph learning algorithm for the class* $\mathrm{MPNN}_{L,M',L_{\mathsf{FNN}}}^{\mathrm{sum}}(\mathcal{G}_{n,d,q})$ *and for any sample* $\mathcal{S}$ *and* $\delta \in (0,1)$, *with probability at least* $1 - \delta$, *we have*

$$|\ell_{\exp}(h_{\mathcal{S}}) - \ell_{\mathrm{emp}}(h_{\mathcal{S}})| \leq 2\widetilde{C}b(d, q, L)k$$
$$+ M\sqrt{\frac{\frac{m_{n,d,q,L}}{k+1}4\log(2) + 2\log(1/\delta)}{|\mathcal{S}|}},$$

*for* $k \in \mathbb{N}$, *where* $\widetilde{C} = 2/nL_\ell L_{\mathsf{FNN}}C_{\mathrm{FD}_L}$, $C_{\mathrm{FD}_L} = C(L)L_\psi \prod_{i=1}^{L} L_{\varphi_i}$ *is the Lipschitz constant in Lemma 14 and* $M$ *is an upper bound of the loss function* $\ell$. □

In Appendix M, we lift the results to MPNNs using *mean aggregation*. Additionally, in Section 5, we discuss our analysis's limitations and future work.

## 5. Limitations, possible road maps, and future work

While our results are the first to successfully incorporate non-trivial graph similarities, architectural choices, and loss functions into the generalization analysis, many open questions remain regarding MPNN generalization properties. First, our techniques are tailored towards discretely-labeled graphs, e.g., not accounting for real-valued labels. While Rauchwerger et al. (2024) did a first step in this direction, their generalization bounds are implicit or existential, i.e., they do not derive concrete upper bounds on the size of the covering, only showing the existence of a finite covering via the compactness of the investigated pseudo-metric spaces. Secondly, while our analysis can be extended to other aggregation functions, e.g., weighted mean, it does not account for other commonly used architectural choices, such as normalization layers or skip connections. Thirdly, although the experimental results in Section 6 indicate that our generalization analysis holds in practice to some extent, it does not explain why gradient descent-based algorithms converge to generalizing solutions. Hence, future work should extend our results to graphs with real-valued features and study whether gradient descent-based algorithms can converge to parameter assignments inducing the desired coverings.

## 6. Experimental study

In the following, we investigate to what extent our theoretical results translate into practice. Specifically, we answer the following questions.

**Q1** To what extent do the empirical covering numbers for different graph families match the theoretical upper bounds derived in Section 4?

**Q3** How are the Forest distance and MPNN outputs correlated?

**Q3** How does the covering number influence the generalization performance of MPNNs?

See https://github.com/benfinkelshtein/CoveredForests for source code and instructions to reproduce all results.

See Appendix P for an overview of employed data sets, neural architectures, experimental protocols, and model configurations.

**Results and discussion** In the following, we answer **Q1-Q3**.

**Q1** See Figures 8 and 9 in the appendix. Across the graph classes $\mathcal{G}_n$, $\mathcal{T}_n^{(2)}$, $\mathcal{F}_n$, and all real-world datasets, we ob-

*Table 1.* The empirical generalization gap computed using the 1-norm and our generalization bound based on estimates of the Lipschitz constant and the upper bound on the loss function.

| | Dataset | | | |
| --- | --- | --- | --- | --- |
| | MUTAG | NCI1 | MCF-7H | OGBG-MOLHIV |
| Train loss | $1.890_{\pm0.078}$ | $0.535_{\pm0.007}$ | $0.327_{\pm0.026}$ | $0.144_{\pm0.006}$ |
| Test loss | $1.546_{\pm0.054}$ | $0.588_{\pm0.151}$ | $0.325_{\pm0.010}$ | $0.151_{\pm0.005}$ |
| Generalization gap | $0.344_{\pm0.095}$ | $0.053_{\pm0.151}$ | $0.002_{\pm0.028}$ | $0.007_{\pm0.008}$ |
| Our bound (Optimal $\varepsilon$) | 0.705 | 0.060 | 0.007 | 0.015 |
| Our bound ($\varepsilon = 0$) | 0.946 | 0.079 | 0.008 | 0.019 |

serve that the covering number increases with larger graph orders or smaller radii, aligning with the expected behavior. Figure 10 in the appendix demonstrates that the covering number bound presented in Theorem 10 is tighter than the number of 1-WL indistinguishable graphs, $m_n$, when compared to the optimal cover, $\mathcal{N}(\cdot, \delta_1^{\mathcal{T}}, \varepsilon)$, highlighting the usefulness of our upper bound. This observation holds even though the optimal cover is based on the 1-norm while our bound is derived using the cut norm, further solidifying the improved bound introduced in Theorem 10.

**Q2** See Figure 11 in the appendix. We observe a strong correlation between the Forest distance and the MPNN output variations, indicated by a high Pearson correlation coefficient across varying datasets and MPNN layers. This finding supports the validity of defining the Forest distance's Lipschitz constant in Lemma 14, showing that it captures the computation of MPNNs.

**Q3** See Table 1. The results demonstrate that the covering number bound presented in Proposition 15 is tight in real-world settings, in the sense that the empirical generalization gap and the computed bound are of the same order of magnitude, closely reflecting the real-world generalization behavior. Furthermore, incorporating the upper bound presented in Proposition 16 yields an even tighter bound over simply equating the covering number to the total number of 1-WL indistinguishable graphs.

## 7. Conclusion

Here, we focused on understanding how choosing different pseudo-metrics on the set of graphs and capturing the computation of different MPNN architectures allow for a tighter analysis of the generalization error of MPNNs. Unlike previous works, our refined analysis allows us to account for non-trivial graph similarities, the impact of different aggregation functions, and various practically relevant loss functions. Our empirical study confirmed the validity of our theoretical findings. *Overall, our theoretical framework constitutes an essential initial step in unraveling how graph structure and architectural choices influence the MPNNs' generalization properties.*

## Acknowledgements

Antonis Vasileiou is funded by the German Research Foundation (DFG) within Research Training Group 2236/2 (UnRAVeL). Ben Finkelshtein is supported by the Clarendon Scholarship. Christopher Morris is partially funded by a DFG Emmy Noether grant (468502433) and RWTH Junior Principal Investigator Fellowship under Germany's Excellence Strategy. Ron Levie is supported by a grant from the United States-Israel Binational Science Foundation (BSF), Jerusalem, Israel, and the United States National Science Foundation (NSF), (NSF-BSF, grant No. 2024660), and by the Israel Science Foundation (ISF grant No. 1937/23). We thank Erik Müller for crafting the figures.

## Impact statement

This paper aims to advance the field of machine learning by providing tight bounds that enhance the theoretical understanding of MPNNs. While our findings hold significant potential for societal impact, this discussion is beyond the immediate scope of this work.

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

# A. Related work

In the following, we discuss relevant related work.

**MPNNs** Recently, MPNNs (Gilmer et al., 2017; Scarselli et al., 2009) emerged as the most prominent graph machine learning architecture. Notable instances of this architecture include, e.g., Duvenaud et al. (2015); Hamilton et al. (2017); Kipf & Welling (2017), and Veličković et al. (2018), which can be subsumed under the message-passing framework introduced in Gilmer et al. (2017). In parallel, approaches based on spectral information were introduced in, e.g., Bruna et al. (2014); Defferrard et al. (2016); Gama et al. (2019); Kipf & Welling (2017); Levie et al. (2019), and Monti et al. (2017)—all of which descend from early work in Baskin et al. (1997); Goller & Küchler (1996); Kireev (1995); Merkwirth & Lengauer (2005); Micheli & Sestito (2005); Micheli (2009); Scarselli et al. (2009), and Sperduti & Starita (1997).

**Generalization frameworks** Vapnik & Chervonenkis (1964; 1971); Vapnik (1998) laid out the theory of statistical learning, introducing the VC dimension and its strong connection to uniform convergence; see also Mohri et al. (2018). Xu & Mannor (2012) introduced an alternative approach based on a notion of "robustness.". Here, if a testing sample is "close" to a training sample, then the expected error is close to the training error, implying a bound on the generalization error; see Section 3. They also showed that so-called *weak robustness* is sufficient and necessary for generalization. Recently, Kawaguchi et al. (2022) proposed data-driven variations of the framework of Xu & Mannor (2012), allowing for a tighter analysis in practice. Sokolic et al. (2017) used Xu & Mannor (2012) to showcase how invariant machine learning architectures, regarding some group action, exhibit a smaller generalization error than their non-invariant counterparts.

**Generalization abilities of MPNNs and GNNs** Scarselli et al. (2018) used classical techniques from learning theory (Karpinski & Macintyre, 1997) to show that MPNNs' VC dimension (Vapnik, 1995) with piece-wise polynomial activation functions on a *fixed* graph, under various assumptions, is in $\mathcal{O}(P^2 n \log n)$, where $P$ is the number of parameters and $n$ is the order of the input graph; see also Hammer (2001). We note here that Scarselli et al. (2018) analyzed a different type of MPNN not aligned with modern MPNN architectures (Gilmer et al., 2017); see also D'Inverno et al. (2024). Garg et al. (2020) showed that the empirical Rademacher complexity (see, e.g., Mohri et al. (2018)) of a specific, simple MPNN architecture, using sum aggregation and specific margin loss, is bounded in the maximum degree, the number of layers, Lipschitz constants of activation functions, and parameter matrices' norms. We note here that their analysis assumes weight sharing across layers. Recently, Karczewski et al. (2024) lifted this approach to $E(n)$-equivariant MPNNs (Satorras et al., 2023). Liao et al. (2021) refined the results of Garg et al. (2020) via a PAC-Bayesian approach, further refined in Ju et al. (2023). See Lee et al. (2024) for (transductive) PAC-Bayesian generalization bounds for knowledge graphs. Maskey et al. (2022; 2024) assumed that data is generated by random graph models, leading to MPNNs' generalization analysis depending on the (average) number of vertices of the graphs. In addition, Levie (2023) and Rauchwerger et al. (2024) defined metrics on attributed graphs, resulting in a generalization bound for MPNNs depending on the covering number of these metrics. Verma & Zhang (2019) studied the generalization abilities of 1-layer MPNNs in a transductive setting based on algorithmic stability. Similarly, Esser et al. (2021) used stochastic block models to study the transductive Rademacher complexity (El-Yaniv & Pechyony, 2007; Tolstikhin & Lopez-Paz, 2016) of standard MPNNs. See also Tang & Liu (2023) for refined results. For semi-supervised vertex classification, Baranwal et al. (2021) studied the classification of a mixture of Gaussians, where the data corresponds to the vertex features of a stochastic block model, deriving conditions under which the mixture model is linearly separable using the GCN layer (Kipf & Welling, 2017). Recently, Morris et al. (2023a) made progress connecting MPNNs' expressive power and generalization ability via the Weisfeiler–Leman hierarchy. They studied the influence of graph structure and the parameters' encoding lengths on MPNNs' generalization by tightly connecting 1-WL's expressivity and MPNNs' VC dimension. They derived that MPNNs' VC dimension depends tightly on the number of equivalence classes computed by the 1-WL over a given set of graphs. Moreover, they showed that MPNNs' VC dimension depends logarithmically on the number of colors computed by the 1-WL and polynomially on the number of parameters. Since relying on the 1-WL, their analysis implicitly assumes a discrete pseudo-metric space, where two graphs are either exactly equal or far apart. One VC lower bound reported in Morris et al. (2023a) was tightened in Daniëls & Geerts (2024) to MPNNs restricted to using a single layer and a width of one. In addition, Pellizzoni et al. (2024) extended the analysis of Morris et al. (2023a) to node-individualized MPNNs and devised a Rademacher-complexity-based approach using a covering number argument (Bartlett et al., 2017). Similar to us, they also assume MPNNs that are (Lipschitz) continuous regarding a given pseudo-metric. However, unlike the present work, they do not derive such pseudo-metric and do not provide an explicit bound on the covering number. Franks et al. (2024) studied MPNNs' VC dimension assuming linearly separable data and showed a tight relationship to the data's margin, also partially explaining when more expressive

architectures lead to better generalization. Li et al. (2024) build on the margin-based generalization framework proposed by Chuang et al. (2021), which is based on $k$-Variance and the Wasserstein distance. They provide a method to analyze how expressiveness affects graph embeddings' inter- and intra-class concentration. Kriege et al. (2018) leveraged results from graph property testing (Goldreich, 2010) to study the sample complexity of learning to distinguish various graph properties, e.g., planarity or triangle freeness, using graph kernels (Borgwardt et al., 2020; Kriege et al., 2020). Finally, Yehudai et al. (2021) showed negative results for MPNNs' generalization ability to larger graphs.

**Graph (pseudo-)distances**  Based on older ideas in Dell et al. (2018); Dvořák (2010); Tinhofer (1991), Böker (2021) defined the *Tree distance*, a pseudo-metric on the set of graphs, and showed a tight correspondence between the former and homomorphism densities regarding trees, which also implies equivalence to the 1-WL. Böker et al. (2023) later lifted the result to specific MPNNs using sum aggregation, deriving uniform continuity. Based on this, Sverdlov et al. (2024) defined an extension of the Tree distance to account for labeled graphs and derived a specific MPNN architecture that exhibits a bi-Lipschitz property. Chuang & Jegelka (2022) defined the TMD distance between the computation graphs of specific MPNNs via a hierarchical optimal transport problem, showed equivalence to the 1-WL, and derived Lipschitz constants. In addition, they studied MPNNs' generalization abilities under distribution shifts. Levie (2023) defined the graphon-signal cut distance, showed that MPNNs are Lipschitz continuous concerning this metric, and derived a bound on the covering number of the space of attributed graphs under this metric. As a result of the Lipschitz continuity and finite covering, he derived a generalization bound for MPNNs. However, the graphon-signal cut distance topology does not give the coarsest topology under which MPNNs are continuous. At the same time, in the present work, we find such a maximally coarse topology, which, in principle, reduces the covering number (asymptotically in the radius). Most recently, Rauchwerger et al. (2024) extended Böker et al. (2023) to attributed graphs, i.e., the vertex features are in $\mathbb{R}^{1 \times d}$, deriving a universal approximation theorem for MPNNs over such graphs and generalization bounds for MPNN using a covering number argument based on the technique from Levie (2023). Unlike the present work, they do not derive explicit bounds on the covering number and only study specific MPNN layers.

## B. Detailed notation

Here, we outlined our notation in more detail.

**Basic notations**  Let $\mathbb{N} := \{1, 2, \ldots\}$ and $\mathbb{N}_0 := \mathbb{N} \cup \{0\}$. The set $\mathbb{R}^+$ denotes the set of non-negative real numbers. For $n \in \mathbb{N}$, let $[n] := \{1, \ldots, n\} \subset \mathbb{N}$. We use $\{\!\{\ldots\}\!\}$ to denote multisets, i.e., the generalization of sets allowing for multiple, finitely many instances for each of its elements. For two non-empty sets $X$ and $Y$, let $Y^X$ denote the set of functions from $X$ to $Y$. Given a set $X$ and a subset $A \subset X$, we define the indicator function $1_A \colon X \to \{0,1\}$ such that $1_A(x) = 1$ if $x \in A$, and $1_A(x) = 0$ otherwise. Let $\boldsymbol{M}$ be an $n \times m$ matrix, $n > 0$ and $m > 0$, over $\mathbb{R}$, then $\boldsymbol{M}_{i,\cdot}$, $\boldsymbol{M}_{\cdot,j}$, $i \in [n]$, $j \in [m]$, are the $i$th row and $j$th column, respectively, of the matrix $\boldsymbol{M}$. Let $\boldsymbol{N}$ be an $n \times n$ matrix, $n > 0$, then the *trace* $\text{Tr}(\boldsymbol{N}) := \sum_{i \in [n]} N_{ii}$. In what follows, $\mathbf{o}$ denotes an all-zero vector with an appropriate number of components.

**Norms**  Given a vector space $V$, a *norm* is a function $\|\cdot\| \colon V \to \mathbb{R}^+$ which satisfies the following properties. For all vectors $\boldsymbol{u}, \boldsymbol{v} \in V$ and scalar $s \in \mathbb{R}$, we have (i) *non-negativity,* $\|\boldsymbol{v}\| \geq 0$ with $\|\boldsymbol{v}\| = 0$ if, and only if, $\boldsymbol{v} = \mathbf{o}$; (ii) *scalar multiplication,* $\|s\boldsymbol{v}\| = |s| \|\boldsymbol{v}\|$; and the (iii) *triangle inequality* holds, $\|\boldsymbol{u} + \boldsymbol{v}\| \leq \|\boldsymbol{u}\| + \|\boldsymbol{v}\|$. When $V$ is some real vector space, say $\mathbb{R}^{1 \times d}$, for $d > 0$, here, and in the remainder of the paper, $\|\cdot\|_1$ and $\|\cdot\|_2$ refer to the 1-*norm* $\|\boldsymbol{x}\|_1 := |x_1| + \cdots + |x_d|$ and 2-*norm* $\|\boldsymbol{x}\|_2 := \sqrt{x_1^2 + \cdots + x_d^2}$, respectively, for $\boldsymbol{x} \in \mathbb{R}^{1 \times d}$.

When considering the vector space $\mathbb{R}^{n \times n}$ of square $n \times n$ matrices, a *matrix norm* $\|\cdot\|$ is a norm as described above, with the additional property that $\|\boldsymbol{M}\boldsymbol{N}\| \leq \|\boldsymbol{M}\|\|\boldsymbol{N}\|$ for all matrices $\boldsymbol{M}$ and $\boldsymbol{N} \in \mathbb{R}^{n \times n}$. A matrix norm is called *induced*, or *operator norm*, if $\|\boldsymbol{M}\| := \sup_{\boldsymbol{x} \in \mathbb{R}^n, \|\boldsymbol{x}\|=1} \|\boldsymbol{M}\boldsymbol{x}\|$, for some vector norm $\|\cdot\|$ on $\mathbb{R}^n$. A matrix norm is called *element-wise* if it is defined in terms of a vector norm $\|\cdot\|$ by interpreting $\boldsymbol{M}$ as a vector in $\mathbb{R}^{n^2}$. However, not every vector norm results in an element-wise matrix norm. For instance, the norm $\|\boldsymbol{M}\|_\infty := \max_{i,j \in [n]} |m_{ij}|$ is a valid norm on $\mathbb{R}^{n^2}$ but does not satisfy the conditions of a matrix norm on $\mathbb{R}^{n \times n}$. Later in the paper, we use the *cut norm* on $\mathbb{R}^{n \times n}$, defined by

$$\|\boldsymbol{M}\|_\square := \max_{S \subset [n], T \subset [n]} \Big| \sum_{i \in S, j \in T} m_{ij} \Big|. \tag{11}$$

Finally, for $\boldsymbol{p} \in \mathbb{R}^{1 \times d}, d > 0$, and $\varepsilon > 0$, the *ball* $B_{\|\cdot\|}(\boldsymbol{p}, \varepsilon, d) := \{\boldsymbol{s} \in \mathbb{R}^{1 \times d} \mid \|\boldsymbol{p} - \boldsymbol{s}\| \leq \varepsilon\}$; when the norm is clear from the context, we omit the subscript.

**Graphs** An *(undirected) graph* $G$ is a pair $(V(G), E(G))$ with *finite* sets of *vertices* $V(G)$ and *edges* $E(G) \subseteq \{\{u, v\} \subseteq V(G) \mid u \neq v\}$. *vertices* or *nodes* $V(G)$ and *edges* $E(G) \subseteq \{\{u, v\} \subseteq V(G) \mid u \neq v\}$. The *order* of a graph $G$ is its number $|V(G)|$ of vertices. If not stated otherwise, we set $n := |V(G)|$ and call $G$ an *n-order graph*. We denote the set of all $n$-order (undirected) graphs by $\mathcal{G}_n$ and the set of all (undirected) graphs up to $n$ vertices by $\mathcal{G}_{\leq n}$. In a *directed graph*, we define $E(G) \subseteq V(G)^2$, where each edge $(u, v)$ has a direction from $u$ to $v$. Given a directed graph $G$ and vertices $u, v \in V(G)$, we say that $v$ is a *child* of $u$ if $(u, v) \in E(G)$. A (directed) graph $G$ is called *connected* if, for any $u, v \in V(G)$, there exist $r \in \mathbb{N}$ and $\{u_1, \ldots, u_r\} \subseteq V(G)$, such that $(u, u_1), (u_1, u_2), \ldots, (u_r, v) \in E(G)$, and analogously for undirected graphs by replacing directed edges with undirected ones. We say that a graph $G$ is *disconnected* if it is not connected. For a graph $G$ and an edge $e \in E(G)$, we denote by $G \setminus e$ the *graph induced by removing* the edge $e$ from $G$. For an $n$-order graph $G \in \mathcal{G}_n$, assuming $V(G) = [n]$, we denote its *adjacency matrix* by $\boldsymbol{A}(G) \in \{0,1\}^{n \times n}$, where $\boldsymbol{A}(G)_{vw} = 1$ if, and only if, $\{v, w\} \in E(G)$. The *neighborhood* of a vertex $v \in V(G)$ is denoted by $N_G(v) := \{u \in V(G) \mid \{v, u\} \in E(G)\}$, where we usually omit the subscript for ease of notation, and the *degree* of a vertex $v$ is $|N_G(v)|$. A graph $G$ is a *tree* if it is connected, but $G \setminus e$ is disconnected for any $e \in E(G)$. A tree or a disjoint collection of trees is known as a *forest*.

A *rooted tree* $(G, r)$ is a tree where a specific vertex $r$ is marked as the *root*. For a rooted (undirected) tree, we can define an implicit direction on all edges as pointing away from the root; thus, when we refer to the *children* of a vertex $u$ in a rooted tree, we implicitly consider this directed structure. For $S \subseteq V(G)$, the graph $G[S] := (S, E_S)$ is the *subgraph induced by* $S$, where $E_S := \{(u, v) \in E(G) \mid u, v \in S\}$. A *(vertex-)labeled graph* is a pair $(G, \ell_G)$ with a graph $G = (V(G), E(G))$ and a (vertex-)label function $\ell_G : V(G) \to \Sigma$, where $\Sigma$ is an arbitrary countable label set. For a vertex $v \in V(G)$, $\ell_G(v)$ denotes its *label*. A Boolean *(vertex-)d-labeled graph* is a pair $(G, \ell_G)$ with a graph $G = (V(G), E(G))$ and a label function $\ell_G : V(G) \to \{0,1\}^d$. We denote the set of all $n$-order Boolean $d$-labeled graphs as $\mathcal{G}_{n,d}^{\mathbb{B}}$. An *attributed graph* is a pair $(G, a_G)$ with a graph $G = (V(G), E(G))$ and an (vertex-)attribute function $a_G : V(G) \to \mathbb{R}^{1 \times d}$, for $d > 0$. That is, contrary to labeled graphs, vertex annotations may be from an uncountable set. The *attribute* or *feature* of $v \in V(G)$ is $a_G(v)$. We denote the class of all $n$-order graphs with $d$-dimensional, real-valued vertex features by $\mathcal{G}_{n,d}^{\mathbb{R}}$.

Two graphs $G$ and $H$ are *isomorphic* if there exists a bijection $\varphi : V(G) \to V(H)$ that preserves adjacency, i.e., $(u, v) \in E(G)$ if, and only if, $(\varphi(u), \varphi(v)) \in E(H)$. In the case of labeled graphs, we additionally require that $\ell_G(v) = \ell_H(\varphi(v))$ for $v \in V(G)$. Moreover, we call the equivalence classes induced by $\simeq$ *isomorphism types* and denote the isomorphism type of $G$ by $\tau(G)$. A *graph class* is a set of graphs that is closed under isomorphism. Given two graphs $G$ and $H$ with disjoint vertex sets, we denote their disjoint union by $G \mathbin{\dot\cup} H$.

## B.1. Continuity on metric spaces

Let $(\mathcal{X}, d_{\mathcal{X}})$ and $(\mathcal{Y}, d_{\mathcal{Y}})$ be two pseudo-metric spaces. A function $f : \mathcal{X} \to \mathcal{Y}$ is called $c_f$-*Lipschitz continuous* if, for $x, x' \in \mathcal{X}$,

$$d_{\mathcal{Y}}(f(x), f(x')) \leq c_f \cdot d_{\mathcal{X}}(x, x').$$

We also define a weaker notion of continuity, known as *uniform continuity*. Specifically, we say that $f$ is uniformly continuous if, for every $\varepsilon > 0$, there exists a $\gamma(\varepsilon, f) > 0$, possibly dependent on $\varepsilon$ and $f$, such that, for $x, x' \in \mathcal{X}$,

$$d_{\mathcal{X}}(x, x') \leq \gamma(\varepsilon, f) \Rightarrow d_{\mathcal{Y}}(f(x), f(x')) \leq \varepsilon.$$

It is easy to check that $c_f$-Lipschitz continuity implies uniform continuity with $\gamma(\varepsilon, f) = \varepsilon/c_f$.

**Feed-forward neural networks** An $L$-layer *feed-forward neural network* (FNN), for $L \in \mathbb{N}$, is a parametric function $\mathsf{FNN}_{\boldsymbol{\theta}}^{(L)} : \mathbb{R}^{1 \times d} \to \mathbb{R}$, $d > 0$, where $\boldsymbol{\theta} := (\boldsymbol{W}^{(1)}, \ldots, \boldsymbol{W}^{(d)}) \subseteq \Theta$ and $\boldsymbol{W}^{(i)} \in \mathbb{R}^{d \times d}$, for $i \in [L-1]$, and $\boldsymbol{W}^{(L)} \in \mathbb{R}^{d \times 1}$, where

$$\boldsymbol{x} \mapsto \sigma\Big(\cdots \sigma\Big(\sigma\Big(\boldsymbol{x}\boldsymbol{W}^{(1)}\Big)\boldsymbol{W}^{(2)}\Big) \cdots \boldsymbol{W}^{(L)}\Big) \in \mathbb{R},$$

for $\boldsymbol{x} \in \mathbb{R}^{1 \times d}$. Here, the function $\sigma : \mathbb{R} \to \mathbb{R}$ is an *activation function*, applied component-wisely, e.g., a *rectified linear unit* (ReLU), where $\sigma(x) := \max(0, x)$. For an FNN where we do not need to specify the number of layers, we write $\mathsf{FNN}_{\boldsymbol{\theta}}$.

# C. The 1-dimensional Weisfeiler–Leman algorithm

The 1-*dimensional Weisfeiler–Leman algorithm* (1-WL) or *color refinement* is a well-studied heuristic for the graph isomorphism problem, originally proposed by Weisfeiler & Leman (1968).[7] Intuitively, the algorithm determines if two graphs are non-isomorphic by iteratively coloring or labeling vertices. Given an initial coloring or labeling of the vertices of both graphs, e.g., their degree or application-specific information, in each iteration, two vertices with the same label get different labels if the number of identically labeled neighbors is unequal. These labels induce a vertex partition, and the algorithm terminates when, after some iteration, the algorithm does not refine the current partition, i.e., when a *stable coloring* or *stable partition* is obtained. Then, if the number of vertices annotated with a specific label is different in both graphs, we can conclude that the two graphs are not isomorphic. It is easy to see that the algorithm cannot distinguish all non-isomorphic graphs (Cai et al., 1992). However, it is a powerful heuristic that can successfully decide isomorphism for a broad class of graphs (Arvind et al., 2015; Babai & Kucera, 1979).

Formally, let $(G, \ell_G)$ be a labeled graph. In each iteration, $t > 0$, the 1-WL computes a *vertex coloring* $C_t^1 \colon V(G) \to \mathbb{N}$, depending on the coloring of the neighbors. That is, in iteration $t > 0$, we set

$$C_t^1(v) := \mathsf{RELABEL}\Big(\big(C_{t-1}^1(v), \{\!\{ C_{t-1}^1(u) \mid u \in N(v) \}\!\}\big)\Big),$$

for vertex $v \in V(G)$, where RELABEL injectively maps the above pair to a unique natural number, which has not been used in previous iterations. In iteration 0, the coloring $C_0^1 := \ell_G$ is used.[8] To test whether two graphs $G$ and $H$ are non-isomorphic, we run the above algorithm in "parallel" on both graphs. If the two graphs have a different number of vertices colored $c \in \mathbb{N}$ at some iteration, the 1-WL *distinguishes* the graphs as non-isomorphic. Moreover, if the number of colors between two iterations, $t$ and $(t + 1)$, does not change, i.e., the cardinalities of the images of $C_t^1$ and $C_{i+t}^1$ are equal, or, equivalently,

$$C_t^1(v) = C_t^1(w) \iff C_{t+1}^1(v) = C_{t+1}^1(w),$$

for all vertices $v, w \in V(G \dot\cup H)$, then the algorithm terminates. For such $t$, we define the *stable coloring* $C_\infty^1(v) = C_t^1(v)$, for $v \in V(G \dot\cup H)$. The stable coloring is reached after at most $\max\{|V(G)|, |V(H)|\}$ iterations (Grohe, 2017). Finally, when considering graphs of a fixed order $n$, we define the equivalence class of a graph $G \in \mathcal{G}_n$ induced by 1-WL as $[G] = \{G' \in \mathcal{G}_n \mid G, G' \text{ are 1-WL indistinguishable}\}$ and the *quotient space* consisting of the equivalence classes as $\mathcal{G}_n/_{\sim_{\mathsf{WL}}} := \{[G] \mid G \in \mathcal{G}_n\}$. We similarly, use $\mathcal{G}_n/_{\sim_{\mathsf{WL}_L}}$ for the quotient space of equivalences classes after $L$ iterations, for $L \in \mathbb{N}$.

**Connection between the 1-WL and MPNNs**  Morris et al. (2019) and Xu et al. (2019) showed that any possible MPNN architecture is limited by the 1-WL in terms of distinguishing non-isomorphic graphs. Furthermore, for the MPNNs defined in Equation (3), Morris et al. (2019) showed that, under appropriate parameter selections, they can achieve expressivity equivalent to that of the 1-WL; see Grohe (2021) and Morris et al. (2023b) for details. Similarly, building on Morris et al. (2019), it is straightforward to show that the MPNNs defined in Equation (1) can distinguish the same pairs of non-isomorphic $n$-order graphs as the 1-WL.

**Unrollings characterization for 1-dimensional Weisfeiler–Leman algorithm**  Following Morris et al. (2020b), given an $n$-order labeled graph $(G, \ell_G)$, we define the *unrolling tree* of depth $L \in \mathbb{N}_0$ for a vertex $u \in V(G)$, denoted as $\mathsf{unr}(G, u, L)$, inductively as follows.

1. For $L = 0$, we consider the trivial tree as an isolated vertex labeled $\ell_G(u)$.

2. For $L > 0$, we consider the root vertex with label $\ell_G(u)$ and, for $v \in N(u)$, we attach the subtree $\mathsf{unr}(G, v, L - 1)$ under the root.

The above unrolling tree construction characterizes the 1-WL algorithm through the following lemma.

**Lemma 17** (Folklore, see, e.g., Morris et al. (2020a)). *For $L \in \mathbb{N}_0$, given a labeled graph $(G, \ell_G)$ and vertices $u, v \in V(G)$, the following are equivalent.*

---

[7]Strictly speaking, the 1-WL and color refinement are two different algorithms. That is, the 1-WL considers neighbors and non-neighbors to update the coloring, resulting in a slightly higher expressive power when distinguishing vertices in a given graph; see (Grohe, 2021) for details. Following customs in the machine learning literature, we consider both algorithms to be equivalent.

[8]Here, we implicitly assume an injective function from $\Sigma$ to $\mathbb{N}$.

*1. The vertices $u$ and $v$ have the same color after $L$ iterations of the 1-WL.*

*2. The unrolling trees $\mathsf{unr}(G, u, L)$ and $\mathsf{unr}(G, v, L)$ are isomorphic.* □

### C.1. The mean 1-dimensional Weisfeiler–Leman algorithm

Similarly to 1-WL, we can define the mean-aggregation 1-WL (1-MWL) on a labeled graph $(G, \ell_G)$ and give a similar characterization based on unrolling trees denoted as $\mathsf{m\text{-}unr}(G, u, L)$. We will later use the 1-MWL to characterize the expressive power of MPNN layers using mean aggregation. Before we describe the 1-MWL algorithm and the unrolling construction, we introduce some notation. For a given multiset $X$, we denote by $\mathsf{set}(X)$ the set consisting of all distinct elements of $X$. Additionally, for each $x \in \mathsf{set}(X)$, we denote by $\mathsf{mul}_X(x)$ the multiplicity of the element $x \in X$ in the multiset. We define

$$\mathsf{freq}(X) \coloneqq \left\{ \left( x, \frac{\mathsf{mul}_X(x)}{|X|} \right) \;\middle|\; x \in \mathsf{set}(X) \right\},$$

mapping each multiset to a set of pairs, encoding the frequency of each element in the multiset. Also, given a directed tree $T$, for each vertex $u \in V(T)$, we denote by $\mathsf{ch}_T(u)$ the set of children of the vertex $u \in V(T)$, i.e., vertices that are one level below $u$ and are connected to $u$. Moreover, for a vertex $u \in V(T)$, we denote by $\mathsf{sub}_T(u)$ the rooted directed subtree under vertex $u$ with root $u$. Finally, for vertex $u \in V(T)$, we define $\mathsf{des}_T(u) = \{\!\{ \tau(\mathsf{sub}_T(v)) \mid v \in \mathsf{ch}_T(u) \}\!\}$.

Given a labeled graph $(G, \ell_G)$, the 1-MWL computes a vertex coloring $C_t^{1,\mathrm{m}} \colon V(G) \to \mathbb{N}$. That is, in iteration $t > 0$, we set

$$C_t^{1,\mathrm{m}}(v) = \mathsf{RELABEL}(C_{t-1}^{1,\mathrm{m}}(v), \mathsf{freq}(M_{t-1}(v))),$$

for $v \in V(G)$, where RELABEL injectively maps the above pair to a unique natural number, which has not been used in previous iterations, and $M_t(v) = \{\!\{ C_t^{1,\mathrm{m}}(u) \mid u \in N(v) \}\!\}$. In iteration 0, we set $C_0^{1,\mathrm{m}} \coloneqq \ell_G$.

**Unrolling characterization for the 1-MWL.** Given a labeled graph $(G, \ell)$ and a vertex $u \in V(G)$, we define the *mean unrolling tree* of depth $L$ $\mathsf{m\text{-}unr}(G, u, L)$, characterizing the 1-MWL up to $L$ iterations. For a vertex $u \in V(G)$, starting from the original unrolling tree $T = \mathsf{unr}(G, u, L)$ of depth $L$, considering the root as level 0, we prune the tree as follows.

1. Starting from level $l = L - 1$.

2. For all vertices $u$ at level $l$, we consider the multiset $\mathsf{des}_T(u)$. Let $c$ be the greatest common factor of the elements in $\{\mathsf{mul}_{\mathsf{dec}_T(u)}(a) \mid a \in \mathsf{set}(\mathsf{des}_T(u))\}$, then for each $a \in \mathsf{set}(\mathsf{dec}_T(u))$, we drop $\frac{c-1}{c}\mathsf{mul}_{\mathsf{des}_T(u)}(a)$ copies of $a$ from $\mathsf{dec}_T(u)$, and we also prune the corresponding subtrees from $T$, leading to an updated tree.

3. If $l = 0$, the pruning is done; otherwise, we set $l = l - 1$ and repeat step 2 with the updated tree $T$.

We call the above-described algorithm the *mean-pruning* of the tree $T$. See Figure 3 for an illustration of the mean-pruning process and the mean-unrolling tree. The above mean unrolling tree construction characterizes the 1-MWL through the following result.

**Proposition 18.** *For $L \in \mathbb{N}_0$, given a labeled graph $(G, \ell_G)$ and vertices $u, v \in V(G)$, the following are equivalent.*

*1. The vertices $u, v$ have the same color after $L$ iterations of the 1-MWL.*

*2. The trees $\mathsf{m\text{-}unr}(G, u, L)$ and $\mathsf{m\text{-}unr}(G, v, L)$ are isomorphic.* □

**Connection between the 1-MWL and MPNNs** Similarly to the results by Morris et al. (2019), we show in Proposition 21 in Appendix C.2 that any possible MPNN architecture described by Equation (5) is limited by 1-MWL in terms of distinguishing non-isomorphic graphs. Furthermore, again, for the MPNNs considered in Equation (5), we show that, under appropriate parameter selection, they can achieve expressivity equivalent to 1-MWL.

### C.2. Mean 1-WL unrollings characterization

Here, we show the correspondence between mean unrollings, the 1-MWL, and mean-aggregation MPNNs. We begin by showing a characterization of the 1-MWL through the mean unrollings defined in Appendix C.1

We first show the following auxiliary lemma, which shows that the computation of $\mathsf{m\text{-}unr}(\cdot)$ at depth $L + 1$ induces a finer partition of the set of vertices compared to the partition induced by $\mathsf{m\text{-}unr}(\cdot)$ at depth $L$.

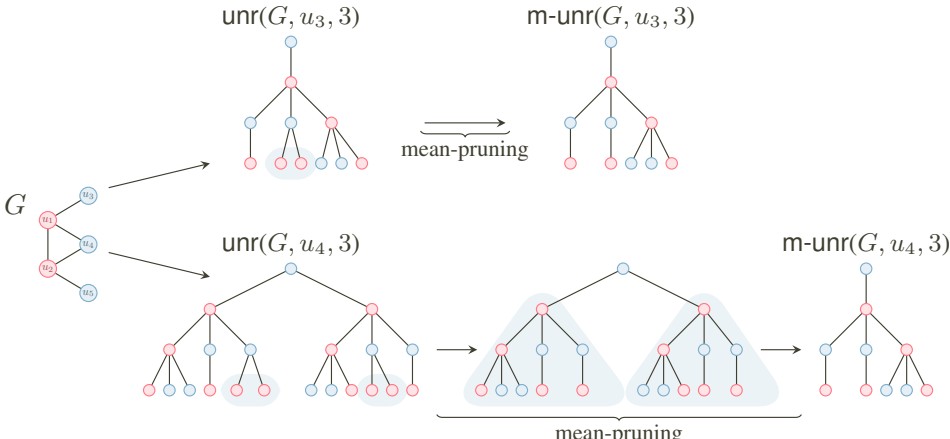

*Figure 3.* An illustration of the unrolling trees and the process of obtaining the mean unrolling trees from the unrolling trees through mean pruning.

**Lemma 19.** *For $L \in \mathbb{N}$, given a labeled graph $(G, \ell_G)$ and vertices $u, v \in V(G)$, the following holds,*

$$\text{m-unr}(G, u, L+1) \simeq \text{m-unr}(G, v, L+1) \implies \text{m-unr}(G, u, L) \simeq \text{m-unr}(G, v, L).$$

*Proof.* As described in Appendix C.1, $\text{m-unr}(G, v, L+1)$ is derived by applying the mean-pruning process to $\text{unr}(G, v, L+1)$. Consider the labeled tree $(T_1, \ell_{T_1}) \coloneqq \text{unr}(G, v, L+1)$. We define the $L$-depth labeled tree $(T_1', \ell_{T_1'})$ induced by $(T_1, \ell_{T_1})$ by relabeling all vertices at level $L$ of $T_1$ and then remove all leaves, i.e., vertices at level $L+1$. We relabel vertices at level $L$ as follows.

For a vertex $x$ at level $L$ of $T_1$ with label $\ell_{T_1}(x)$, let $M_{T_1,x}$ denote the multiset of labels of its children, i.e.,

$$M_{T_1,x} = \{\!\{ \ell_{T_1}(y) \mid y \in \text{ch}_{T_1}(x) \}\!\}.$$

We compute the new label

$$\ell_{T_1'}(x) = \text{RELABEL}(\ell_{T_1}(x), \text{freq}(M_{T_1,x})). \tag{12}$$

Similarly, let $(T_2, \ell_{T_2}) = \text{unr}(G, u, L+1)$, and define $(T_2', \ell_{T_2'})$ analogously. By definition, $\text{m-unr}(G, u, L+1) \simeq \text{m-unr}(G, v, L+1)$ if, and only if, the trees $T_1''$ and $T_2''$, obtained by applying the mean-pruning process to $T_1'$ and $T_2'$, respectively, are isomorphic. Now, if $T_1'' \simeq T_2''$, it directly follows that $\text{m-unr}(G, u, L) \simeq \text{m-unr}(G, v, L)$. The reasoning is as follows: the trees $T_1'$ and $T_2'$ up to level $L-1$ are structurally identical to $\text{unr}(G, u, L+1)$ and $\text{unr}(G, v, L+1)$, respectively. Furthermore, if two vertices in layer $L$ of $T_1'$ have the same label, then these vertices must have also had the same label before the relabeling step (i.e., in $\text{unr}(G, u, L+1)$). This follows directly from Equation (12). Therefore, if applying the mean-pruning process to $T_1'$ and $T_2'$ results in isomorphic graphs, applying the mean-pruning process to $\text{unr}(G, u, L)$ and $\text{unr}(G, v, L)$ must also yield isomorphic graphs. □

We next show the correspondence between mean unrollings and the 1-MWL.

**Proposition 20** (Proposition 18 in the main paper)**.** *For $L \in \mathbb{N}_0$, given a labeled graph $(G, \ell_G)$ and vertices $u, v \in V(G)$, the following are equivalent.*

1. *Vertices $u$ and $v$ have the same color after $L$ iterations of the 1-MWL.*

2. *The trees $\text{m-unr}(G, u, L)$ and $\text{m-unr}(G, v, L)$ are isomorphic.*

*Proof.* For the proof of both directions of the proposition, we will need the following observation.

**Observation 1** (Recursive formulation of mean unrolling)**.** *For each $L \in \mathbb{N}$, $u \in V(G)$, the mean unrolling $\text{m-unr}(G, u, L+1)$ can be constructed by considering a root vertex with label $\ell_G(u)$ and attaching to this root vertex all $\text{m-unr}(G, w, L)$, for $w \in N(u)$, and finally performing the process described in step 2 of the $m$-unrolling construction on level $l = 1$.*

We now prove the two directions.

*1 ⇒ 2:* By induction on $L$. For $L = 0$, the result is clear. For the induction hypothesis, we assume that for some $L \in \mathbb{N}$, and for all vertices $u, v \in V(G)$, if $C_L^{1,\mathrm{m}}(u) = C_L^{1,\mathrm{m}}(v)$, then m-unr$(G, u, L) \simeq$ m-unr$(G, v, L)$. Now, for vertices $u, v \in V(G)$, if $C_{L+1}^{1,\mathrm{m}}(u) = C_{L+1}^{1,\mathrm{m}}(v)$, then by definition, freq$(M_L(u)) =$ freq$(M_L(v))$, where $M_L(v) = \{\!\{ C_L^{1,\mathrm{m}}(w) \mid w \in N(v) \}\!\}$. We consider the tree $T_u$ with root $u$ and attach all m-unr$(G, w, L)$, for $w \in N(u)$. Analogously, we consider the tree $T_v$ with root $v$. By the observation above, we drop (if necessary) all the subtrees rooted at level 1 of $T_u$ as described in step 2 of the m-unr operation, leading to a new tree $T_u'$ which is isomorphic to m-unr$(G, u, L + 1)$. We do the same on $T_v$, leading to $T_v' \simeq$ m-unr$(G, v, L + 1)$. Now, using the induction hypothesis, since freq$(M_L(u)) =$ freq$(M_L(v))$, there exists a bijection $\sigma$ between the subtrees $t$ rooted at level 1 of $T_u'$ and the subtrees rooted at level 1 of $T_v'$, such that $t \simeq \sigma(t)$, for all subtrees $t$. Finally, because $C_{L+1}^{1,\mathrm{m}}(u) = C_{L+1}^{1,\mathrm{m}}(v)$ implies $C_0^{1,\mathrm{m}}(u) = C_0^{1,\mathrm{m}}(v)$, the roots of $T_u'$ and $T_v'$ have the same color, implying that m-unr$(G, u, L + 1) \simeq$ m-unr$(G, v, L + 1)$.

*2 ⇒ 1:* For the reverse direction, we again proceed by induction. If m-unr$(G, u, L + 1) \simeq$ m-unr$(G, v, L + 1)$, the observation above and the inductive hypothesis directly imply that freq$(M_L(u)) =$ freq$(M_L(v))$. Furthermore, by Lemma 19, m-unr$(G, u, L + 1) \simeq$ m-unr$(G, v, L + 1)$ implies m-unr$(G, u, L) \simeq$ m-unr$(G, v, L)$, which concludes the proof. □

Next, we show that mean-aggregation MPNNs as defined in Equation (5) are limited by 1-MWL in distinguishing non-isomorphic graphs. In addition, we prove that under an appropriate parameter selection, both algorithms can achieve exactly the expressivity of the 1-MWL. Formally,

**Proposition 21.** *Let $(G, \ell_G) \in \mathcal{G}_{n,d}^{\mathbb{B}}$ be a labeled graph and $C_t^{1,\mathrm{m}}$ be the coloring function of 1-MWL on iteration $t > 0$. Then for all $t \geq 0, u, v \in V(G)$ and for all MPNNs described by Equation (5), we have that,*

$$C_t^{1,\mathrm{m}}(u) = C_t^{1,\mathrm{m}}(v) \implies \boldsymbol{h}_u^{(t)} = \boldsymbol{h}_v^{(t)}.$$

*Additionally, for $L > 0$, there exists a choice of parameters $\boldsymbol{W}_t'^{(1)}$, $\boldsymbol{W}_t'^{(2)}$, $\varphi_t'$, $t \in [L]$, such that if $\boldsymbol{h}'^{(t)}$ is the MPNN induced by $\boldsymbol{W}_t'^{(1)}$, $\boldsymbol{W}_t'^{(2)}$, $\varphi_t'$, we have that:*

$$\boldsymbol{h}_u'^{(t)} = \boldsymbol{h}_v'^{(t)} \implies C_t^{1,\mathrm{m}}(u) = C_t^{1,\mathrm{m}}(v).$$

*Proof.* We recall that for a multiset $X$, freq$(X)$ is defined as

$$\mathsf{freq}(X) := \left\{ \left( x, \frac{\mathsf{mul}_X(x)}{|X|} \right) \;\middle|\; x \in \mathsf{set}(X) \right\}.$$

We now define the following class of MPNN architectures:

$$\boldsymbol{h}_v^{(t)} := \mathsf{UPD}^{(t)}\left( \boldsymbol{h}_v^{(t-1)}, \mathsf{AGG}^{(t)}\left( \mathsf{freq}\left( \{\!\{ \boldsymbol{h}_w^{(t-1)} \mid w \in N(v) \}\!\} \right) \right) \right), \tag{13}$$

where $\mathsf{UPD}^{(t)}$ and $\mathsf{AGG}^{(t)}$ are two parameterized functions. It suffices to show that the MPNNs described by Equation (5) are a special case of the MPNNs in Equation (13). The proof then proceeds by induction exactly as in Morris et al. (2019, Theorem 1). To verify this, we set

$$\mathsf{AGG}^{(t)}(X) = \sum_{(a_1, a_2) \in \mathsf{freq}(X)} a_1 \cdot a_2 \cdot \boldsymbol{W}_{t-1}^{(2)},$$

$$\mathsf{UPD}^{(t)}(x_1, x_2) = \varphi_t\left( x_1 \cdot \boldsymbol{W}_{t-1}^{(1)} + x_2 \right).$$

The other direction follows by straightforward adaption of the proof of Morris et al. (2019, Theorem 2) by adapting Lemma 11 and 14.

□

## D. Pseudo-metrics on graph

First, let us consider (pseudo-)metrics on the set of $n$-order unlabeled graphs $\mathcal{G}_n$, for $n \in \mathbb{N}$. The straightforward way to define a metric on $\mathcal{G}_n$ is through the adjacency matrices of the graphs. To that, for $n \in \mathbb{N}$, a matrix $\boldsymbol{P} \in \{0,1\}^{n \times n}$ is a *permutation matrix* if $\|\boldsymbol{P}_{i,\cdot}\|_1 = 1$, for $i \in [n]$, and $\|\boldsymbol{P}_{\cdot,j}\|_1 = 1$, for $j \in [n]$. More generally, a matrix $\boldsymbol{S} \in [0,1]^{n \times n}$ is a *doubly-stochastic matrix* if $\|\boldsymbol{S}_{i,\cdot}\|_1 = 1$, for $i \in [n]$, and $\|\boldsymbol{S}_{\cdot,j}\|_1 = 1$, for $j \in [n]$. Let $P_n$ and $D_n$ denote the sets of $n \times n$ permutation and doubly-stochastic matrices, respectively. Now observe that graphs $G$ and $H \in \mathcal{G}_n$ are isomorphic if, and only, if there exists a permutation matrix $\boldsymbol{P} \in P_n$ such that $\boldsymbol{A}(G)\boldsymbol{P} = \boldsymbol{P}\boldsymbol{A}(H)$. Similarly, from Tinhofer (1991), it follows that the graphs $G$ and $H$ are not distinguished by 1-WL if, and only, if there exist of a doubly-stochastic matrix $\boldsymbol{S}$ such that $\boldsymbol{A}(G)\boldsymbol{S} = \boldsymbol{S}\boldsymbol{A}(H)$.

Based on the above, for $n \in \mathbb{N}$, let $G, H \in \mathcal{G}_n$ and $\|\cdot\|$ is a norm in $\mathbb{R}^{n \times n}$, we define the following pseudo-metric on $\mathcal{G}_n$,

$$\delta_{\|\cdot\|}(G, H) := \min_{\boldsymbol{P} \in P_n} \|\boldsymbol{A}(G)\boldsymbol{P} - \boldsymbol{P}\boldsymbol{A}(H)\|.$$

By definition, it holds that $\delta_{\|\cdot\|}(G, H) = 0$ if, and only, if $G$ and $H$ are isomorphic.

Secondly, we extend the above definition to the set of labeled $n$-order graphs. For simplicity and to be aligned with Morris et al. (2023a), we consider $d$-dimensional Boolean features, though the following distance can also be defined in the set of real-valued features. Given two labeled $n$-order graphs $(G, \ell_G), (H, \ell_H) \in \mathcal{G}_{n,d}^{\mathbb{B}}$ and a distance function dist$: \{0,1\}^d \times \{0,1\}^d \to \mathbb{R}^+$, assuming $V(G) = V(H) = [n]$, we define the *distance matrix* $\boldsymbol{L}(G, H) = [\text{dist}(\ell_G(i), \ell_H(j))]_{i \in V(G), j \in V(H)}$. Following Bento & Ioannidis (2019), we define the following metric on $\mathcal{G}_{n,d}^{\mathbb{B}}$, for any entry-wise or operator norm $\|\cdot\|$,

$$\widetilde{\delta}_{\|\cdot\|}(G, H) := \min_{\boldsymbol{P} \in P_n} \|\boldsymbol{A}(G)\boldsymbol{P} - \boldsymbol{P}\boldsymbol{A}(H)\| + \text{Tr}(\boldsymbol{P}^\mathsf{T}\boldsymbol{L}(G, H)).$$

To see the effect of the second term, suppose that dist is the discrete distance function. Then, $\min_{\boldsymbol{P} \in P_n} \text{Tr}(\boldsymbol{P}^\mathsf{T}\boldsymbol{L}(G, H))$ finds a label-preserving vertex permutation. Again, we can easily verify that $\widetilde{\delta}_{\|\cdot\|}(G, H) = 0$ if, and only, if $(G, \ell_G)$ and $(H, \ell_H)$ are isomorphic.

## E. Proof of the pseudo-metric property and 1-WL equivalence

In this section, we prove that the family of distances defined in $\mathcal{G}_{n,d}^{\mathbb{B}}$ by Equation (8) constitutes a well-defined pseudo-metric on the set of graphs. We then show that this family of pseudo-metrics is equivalent in expressivity to the 1-WL algorithm, i.e., we show Proposition 1.

To establish the pseudo-metric property, we use the result by Bento & Ioannidis (2019) showing that for every entry-wise norm, the following distance defines a pseudo-metric on $\mathcal{G}_{n,d}^{\mathbb{B}}$,

$$\overline{\delta}_{\|\cdot\|}^{\mathcal{T}}(G, H) := \min_{\boldsymbol{S} \in D_n} \|\boldsymbol{A}(G)\boldsymbol{S} - \boldsymbol{S}\boldsymbol{A}(H)\| + \text{Tr}(\boldsymbol{S}^\mathsf{T}\boldsymbol{L}(G, H)),$$

where

$$L_{i,j} = \text{dist}(\ell_G(i), \ell_H(j)),$$

for some distance function dist (e.g., dist$(\ell_G(i), \ell_H(j) = 1$ if $\ell_G(i) \neq \ell_H(j)$ and 0 otherwise.)

However, the cut norm is not an entry-wise norm, and thus, we need to prove the pseudo-metric property separately. Nevertheless, we follow the argumentation used for entry-wise norms. To achieve this, we first establish the following auxiliary lemma.

**Lemma 22.** *Let $\boldsymbol{A}$ be the adjacency matrix of an $n$-order graph, then for all doubly-stochastic matrices $\boldsymbol{S} \in D_n$, the following inequalities hold,*

$$\|\boldsymbol{A}\boldsymbol{S}\|_\square \leq \|\boldsymbol{A}\|_\square,$$
$$\|\boldsymbol{S}\boldsymbol{A}\|_\square \leq \|\boldsymbol{A}\|_\square.$$

*Proof.* By the Birkhoff-von-Neumann theorem, each doubly-stochastic matrix can be written as a convex combination of permutation matrices. That is, if $\boldsymbol{S} \in D_n$, there exist a $k \in \mathbb{N}$, $\lambda_1, \dots \lambda_k \in [0,1]$, and permutation matrices $\boldsymbol{P}_1, \dots, \boldsymbol{P}_k$,

such that $\boldsymbol{S} = \sum_{i=1}^{k} \lambda_i \boldsymbol{P_i}$. Therefore, if $\boldsymbol{A}$ is the adjacency matrix of an $n$-order graph, then

$$\|\boldsymbol{AS}\|_\square = \|\boldsymbol{A} \sum_{i=1}^{k} \lambda_i \boldsymbol{P_i}\|_\square \le \sum_{i=1}^{k} \lambda_i \|\boldsymbol{AP_i}\|_\square = \sum_{i=1}^{k} \lambda_i \|\boldsymbol{A}\|_\square = \|\boldsymbol{A}\|_\square,$$

where the first inequality is derived from the sub-additivity and the absolute homogenicity property of the cut norm. In addition, because the cut norm is invariant under permutations, it holds that $\|\boldsymbol{P_i A}\|_\square = \|\boldsymbol{A}\|_\square$. We similarly can prove the second inequality. $\qquad\square$

We are now ready to show the pseudo-metric property of $\overline{\delta_\square^{\mathcal{T}}}$ on $\mathcal{G}_{n,d}^{\mathbb{B}}$.

**Proposition 23** (First part of Proposition 1 in the main paper)**.** *The distance function $\overline{\delta_\square^{\mathcal{T}}}$ defines a pseudo-metric on $\mathcal{G}_{n,d}^{\mathbb{B}}$, for $d \in \mathbb{N}$*

*Proof.* Let the graphs $(G_A, \ell_{G_A}), (G_B, \ell_{G_B}), (G_C, \ell_{G_C}) \in \mathcal{G}_{n,d}^{\mathbb{B}}$.
*Symmetry.* To prove the symmetry property, i.e., $\overline{\delta_\square^{\mathcal{T}}}(G_A, G_B) = \overline{\delta_\square^{\mathcal{T}}}(G_B, G_A)$, we use the following properties:

1. For all adjacency matrices $\boldsymbol{A}$, $\|\boldsymbol{A}\|_\square = \|\boldsymbol{A}^\intercal\|_\square$.
2. For all $\boldsymbol{S} \in D_n, \boldsymbol{S}^\intercal \in D_n$.
3. $\mathrm{Tr}(\boldsymbol{A}^\intercal \boldsymbol{B}) = \mathrm{Tr}(\boldsymbol{A}\boldsymbol{B}^\intercal)$.

We then have,

$$\begin{aligned}
\overline{\delta_\square^{\mathcal{T}}}(G_A, G_B) &= \min_{\boldsymbol{S} \in D_n} \|\boldsymbol{A}(G_A)\boldsymbol{S} - \boldsymbol{S}\boldsymbol{A}(G_B)\|_\square + \mathrm{Tr}(\boldsymbol{S}^\intercal \boldsymbol{L}(G_A, G_B)) \\
&= \min_{\boldsymbol{S} \in D_n} \|\boldsymbol{S}\boldsymbol{A}(G_B) - \boldsymbol{A}(G_A)\boldsymbol{S}\|_\square + \mathrm{Tr}(\boldsymbol{S}^\intercal \boldsymbol{L}(G_A, G_B)) \\
&= \min_{\boldsymbol{S} \in D_n} \|\boldsymbol{A}(G_B)\boldsymbol{S}^\intercal - \boldsymbol{S}^\intercal \boldsymbol{A}(G_A)\|_\square + \mathrm{Tr}(\boldsymbol{S}\boldsymbol{L}(G_A, G_B)^\intercal) \\
&= \min_{\boldsymbol{S}^\intercal \in D_n} \|\boldsymbol{A}(G_B)\boldsymbol{S}^\intercal - \boldsymbol{S}^\intercal \boldsymbol{A}(G_A)\|_\square + \mathrm{Tr}((\boldsymbol{S}^\intercal)^\intercal \boldsymbol{L}(G_A, G_B)^\intercal) \\
&= \min_{\boldsymbol{S}^\intercal \in D_n} \|\boldsymbol{A}(G_B)\boldsymbol{S}^\intercal - \boldsymbol{S}^\intercal \boldsymbol{A}(G_A)\|_\square + \mathrm{Tr}((\boldsymbol{S}^\intercal)^\intercal \boldsymbol{L}(G_B, G_A)) \\
&= \min_{\boldsymbol{S} \in D_n} \|\boldsymbol{A}(G_B)\boldsymbol{S} - \boldsymbol{S}\boldsymbol{A}(G_A)\|_\square + \mathrm{Tr}(\boldsymbol{S}^\intercal \boldsymbol{L}(G_B, G_A)) \\
&= \overline{\delta_\square^{\mathcal{T}}}(G_B, G_A),
\end{aligned}$$

as desired.

*Triangle inequality.* We define $\boldsymbol{S}' := \arg\min_{\boldsymbol{S} \in D_n} \|\boldsymbol{A}(G_A)\boldsymbol{S} - \boldsymbol{S}\boldsymbol{A}(G_B)\|_\square + \mathrm{Tr}(\boldsymbol{S}^\intercal \boldsymbol{L}(G_A, G_B))$ and $\boldsymbol{S}'' = \arg\min_{\boldsymbol{S} \in D_n} \|\boldsymbol{A}(G_B)\boldsymbol{S} - \boldsymbol{S}\boldsymbol{A}(G_C)\|_\square + \mathrm{Tr}(\boldsymbol{S}^\intercal \boldsymbol{L}(G_B, G_C))$. It is easy to verify that the product of two doubly-stochastic matrices is a doubly-stochastic matrix and, therefore,

$$\overline{\delta_\square^{\mathcal{T}}}(G_A, G_C) \le \|\boldsymbol{A}(G_A)\boldsymbol{S}'\boldsymbol{S}'' - \boldsymbol{S}'\boldsymbol{S}''\boldsymbol{A}(G_C)\|_\square + \mathrm{Tr}((\boldsymbol{S}'\boldsymbol{S}'')^\intercal \boldsymbol{L}(G_A, G_C)).$$

Therefore, it suffices to show that

$$\|\boldsymbol{A}(G_A)\boldsymbol{S}'\boldsymbol{S}'' - \boldsymbol{S}'\boldsymbol{S}''\boldsymbol{A}(G_C)\|_\square \le \|\boldsymbol{A}(G_A)\boldsymbol{S}' - \boldsymbol{S}'\boldsymbol{A}(G_B)\|_\square + \|\boldsymbol{A}(G_A)\boldsymbol{S}'' - \boldsymbol{S}''\boldsymbol{A}(G_C)\|_\square$$

and,

$$\mathrm{Tr}((\boldsymbol{S}'\boldsymbol{S}'')^\intercal \boldsymbol{L}(G_B, G_C)) \le \mathrm{Tr}((\boldsymbol{S}')^\intercal \boldsymbol{L}(G_A, G_B)) + \mathrm{Tr}((\boldsymbol{S}'')^\intercal \boldsymbol{L}(G_B, G_C)).$$

For the first inequality, we use Lemma 22 and the triangle inequality for the cut norm,

$$\begin{aligned}
\|\boldsymbol{A}(G_A)\boldsymbol{S}'\boldsymbol{S}'' - \boldsymbol{S}'\boldsymbol{S}''\boldsymbol{A}(G_C)\|_\square &\le \|\boldsymbol{A}(G_A)\boldsymbol{S}'\boldsymbol{S}'' - \boldsymbol{S}'\boldsymbol{A}(G_B)\boldsymbol{S}''\|_\square \\
&\qquad + \|\boldsymbol{S}'\boldsymbol{A}(G_B)\boldsymbol{S}'' - \boldsymbol{S}'\boldsymbol{S}''\boldsymbol{A}(G_C)\|_\square \\
&\le \|\boldsymbol{A}(G_A)\boldsymbol{S}' - \boldsymbol{S}'\boldsymbol{A}(G_B)\|_\square + \|\boldsymbol{A}(G_B)\boldsymbol{S}'' - \boldsymbol{S}''\boldsymbol{A}(G_C)\|_\square.
\end{aligned}$$

For the second inequality, we use the notation $\text{dist}(\ell_G(i), \ell_H(j))$ for the distance between the label of vertex $i \in V(G)$ and the label of vertex $j \in V(H)$ (e.g., $\text{dist}(\ell_G(i), \ell_H(j)) = \mathbf{1}_{\ell_G(i) \neq \ell_H(j)}$, where $\mathbf{1}$ is the indicator function.)

$$
\begin{aligned}
\text{Tr}((\boldsymbol{S}'\boldsymbol{S}'')^\intercal \boldsymbol{L}(G_A, G_C)) &= \sum_{i,j \in [n]} \sum_{k \in [n]} s'_{i,k} s''_{k,j} \, \text{dist}(\ell_{G_A}(i), \ell_{G_C}(j)) \\
&\leq \sum_{i,j \in [n]} \sum_{k \in [n]} s'_{i,k} s''_{k,j} (\text{dist}(\ell_{G_A}(i), \ell_{G_B}(k)) + \text{dist}(\ell_{G_B}(k), \ell_{G_C}(j))) \\
&= \sum_{i,k \in [n]} s'_{i,k} \, \text{dist}(\ell_{G_A}(i), \ell_{G_B}(k)) \sum_{j \in [n]} s''_{k,j} \\
&\qquad\qquad + \sum_{k,j \in [n]} s''_{k,j} \, \text{dist}(\ell_{G_B}(k), \ell_{G_C}(j)) \sum_{i \in [n]} s'_{i,k} \\
&\leq \text{Tr}((\boldsymbol{S}')^\intercal \boldsymbol{L}(G_A, G_B)) + \text{Tr}((\boldsymbol{S}'')^\intercal \boldsymbol{L}(G_B, G_C)),
\end{aligned}
$$

where the first inequality follows from the triangle inequality of metric $\text{dist}$, and the last inequality from $\|\boldsymbol{S}''\|_1 \leq 1$ and $\|\boldsymbol{S}'\|_1 \leq 1$. $\qquad\square$

We now show the equivalence of the pseudo-metric $\bar{\delta}_\square^{\mathcal{T}}$ and the 1-WL in distinguishing non-isomorphic graphs.

**Lemma 24** (Second part of Proposition 1 in the main paper). *For all $n, d \in \mathbb{N}$ and $G, H \in \mathcal{G}_{n,d}^{\mathbb{B}}$ we have that $\bar{\delta}_\square^{\mathcal{T}}(G, H) = 0$ if, and only, if $G$ and $H$ are 1-WL indistinguishable.*

*Proof.* For all $n, d \in \mathbb{N}$ and $G, H \in \mathcal{G}_{n,d}^{\mathbb{B}}$, we simplify the notation as follows. Once an adjacency matrix $\boldsymbol{A}(G)$ is fixed for an $n$-order graph $G$, we refer to the $i$-th vertex of $G$ as the vertex corresponding to the $i$-th row of $\boldsymbol{A}(G)$, for $i \in [n]$. We define an injection $\theta \colon \mathbb{B}^d \to \{n+2, n+3, \ldots\}$. Then, we introduce a transformation $T_\theta \colon \mathcal{G}_{n,d}^{\mathbb{B}} \to \mathcal{G}_{\leq m'}$, which maps graphs $G \in \mathcal{G}_{n,d}^{\mathbb{B}}$ to unlabeled graphs $G' \in \mathcal{G}_{\leq m'}$, where $m'$ is sufficiently large. The transformation $T_\theta$ is defined as follows.

Given $(G, \ell_G) \in \mathcal{G}_{n,d}^{\mathbb{B}}$, for each $u \in V(G)$, we add $\theta(\ell_G(u))$ new vertices, denoted $u_1, \ldots, u_{\theta(\ell_G(u))}$, to the graph $G$. These new vertices form a clique and one of them (without loss of generality, $u_1$) is connected to $u$. Formally, we define:

$$
T_\theta((G, \ell_G)) = G',
$$

where

$$
V(G') = V(G) \cup \left( \bigcup_{u \in V(G)} \left( \bigcup_{i=1}^{\theta(\ell_G(u))} \{u_i\} \right) \right),
$$

and

$$
E(G') = E(G) \cup \left( \bigcup_{u \in V(G)} \{u, u_1\} \right) \cup \left( \bigcup_{u \in V(G)} \left( \bigcup_{\substack{i,j=1 \\ i \neq j}}^{\theta(\ell_G(u))} \{u_i, u_j\} \right) \right).
$$

Let $m$ denote the number of added vertices, i.e., $m = \sum_{u \in V(G)} \theta(\ell_G(u))$. The adjacency matrix $\boldsymbol{A}(G')$ of the transformed graph $G'$ can then be represented in the following block structure:

$$
\boldsymbol{A}(G') = \begin{bmatrix} \boldsymbol{A}(G) & \boldsymbol{B} \\ \boldsymbol{B}^\intercal & \boldsymbol{D} \end{bmatrix},
$$

where $\boldsymbol{B} \in \mathbb{R}^{n \times m}$ and $\boldsymbol{D} \in \mathbb{R}^{m \times m}$ are symmetric.

Now, let $G, H \in \mathcal{G}_{n,d}^{\mathbb{B}}$ and their transformed graphs be $G' = T_\theta(G)$ and $H' = T_\theta(H)$. We aim to prove the equivalence of the following statements, completing the proof.

1. $\delta_{\|\cdot\|}^{\mathcal{T}}(G', H') = 0$.

2. $G'$ and $H'$ are 1-WL-indistinguishable.

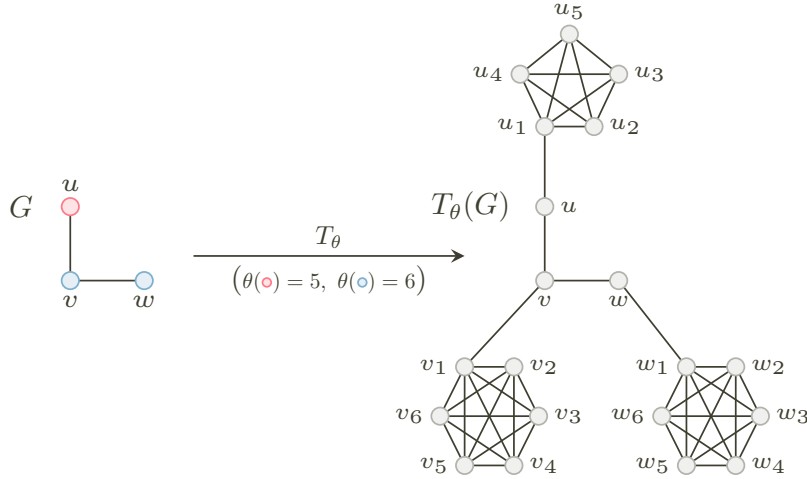

*Figure 4.* An illustration of the $T_\theta$ transformation of a labeled graph $G$ into an unlabeled graph $T_\theta(G)$. For illustration reasons the colors red and blue have chosen as the initial node features in $\mathbb{R}_d^{\mathbb{B}}$.

3. $G$ and $H$ are 1-WL-indistinguishable.

4. $\widetilde{\delta}_{\|\cdot\|}^{\mathcal{T}}(G, H) = 0$.

The equivalences follow as outlined below:

- 1. $\Leftrightarrow$ 2.: This is a direct result of Tinhofer (1986).

- 2. $\Rightarrow$ 3. Let $C_t'^{(1)}$ and $C_t^{(1)}$ represent the coloring functions of the 1-WL algorithm applied to the graphs $G'$ and $(G, \ell_G)$, respectively. We aim to show that for all $u, v \in V(G)$, if $C_\infty'^{(1)}(u) = C_\infty'^{(1)}(v)$, then it follows that $C_\infty^{(1)}(u) = C_\infty^{(1)}(v)$. To do this, we will prove by induction that for each $t$, if $C_{t+2}'^{(1)}(u) = C_{t+2}'^{(1)}(v)$, then $C_t^{(1)}(u) = C_t^{(1)}(v)$.

  For $t = 0$, if $C_2'^{(1)}(u) = C_2'^{(1)}(v)$, we conclude that $\ell_G(u) = \ell_G(v)$. Otherwise, the vertices $u$ and $v$ would have different cliques attached to them, resulting in distinct colors in the second iteration. Now, assume the induction hypothesis, i.e., $C_{t+2}'^{(1)}(u) = C_{t+2}'^{(1)}(v) \implies C_t^{(1)}(u) = C_t^{(1)}(v)$, for $t \in \mathbb{N}$. Next, suppose that $C_{t+3}'^{(1)}(u) = C_{t+3}'^{(1)}(v)$. We further note that the added vertices in $G'$, even from the first iteration, have different colors from those of the original vertices since their degrees are strictly larger than $n + 1$, while the original vertices have degrees strictly smaller than $n + 1$. That is, $|N_{G'}(u_i)| > n - 1$ and $|N_{G'}(u)| < n + 1$ for all $u \in V(G), i \in [\theta(\ell_G(u))]$. By the definition of the 1-WL algorithm and applying the induction hypothesis, we obtain the following,

$$C_t^{(1)}(u) = C_t^{(1)}(v),$$
$$\{\!\!\{ C_t^{(1)}(w) \mid w \in N_{G'}(u) \}\!\!\} = \{\!\!\{ C_t^{(1)}(w) \mid w \in N_{G'}(v) \}\!\!\}.$$

  By the second equality and the fact that the added vertices, namely $\{u_i\}_{i \in \theta(\ell_G(u))}$ for $u \in V(G)$, will always have distinct stable colors from the original vertices (i.e., vertices in $V(G)$), we deduce that

$$\{\!\!\{ C_t^{(1)}(w) \mid w \in N_G(u) \}\!\!\} = \{\!\!\{ C_t^{(1)}(w) \mid w \in N_G(v) \}\!\!\}.$$

  This completes the induction.

- 3. $\Rightarrow$ 2. It suffices to show that $C_t^{(1)}(u) = C_t^{(1)}(v)$ implies the following three equalities.

  (a) $C_t'^{(1)}(u) = C_t'^{(1)}(v)$, for all $u, v \in V(G)$,
  (b) $C_t'^{(1)}(u_1) = C_t'^{(1)}(v_1)$,
  (c) $C_t'^{(1)}(u_i) = C_t'^{(1)}(v_j)$, for all $i, j \in [\theta(\ell_G(u))]$.

We can prove $(a) \implies (b)$ and $(b) \implies (c)$ by induction. First, we show that if $C'^{(1)}_t(u_1) = C'^{(1)}_t(v_1)$, then $C'^{(1)}_{t+1}(u_i) = C'^{(1)}_{t+1}(v_j)$ for all $u, v \in V(G)$ and $i, j \in [\theta(\ell_G(u))]$ (implying $(b) \implies (c)$). Similarly, we can show by induction that $C'^{(1)}_t(u) = C'^{(1)}_t(v)$ implies $C'^{(1)}_{t+1}(u_1) = C'^{(1)}_{t+1}(v_1)$ for all $u, v \in V(G)$ (implying $(a) \implies (b)$).

Next, we use again induction to prove $C^{(1)}_t(u) = C^{(1)}_t(v) \implies (a)$. If $C^{(1)}_{t+1}(u) = C^{(1)}_{t+1}(v)$, by the induction hypothesis and the definition of the 1-WL algorithm, we obtain.

$$C'^{(1)}_t(u) = C'^{(1)}_t(v),$$

$$\{\!\!\{ C'^{(1)}_t(w) \mid w \in N_G(u) \}\!\!\} = \{\!\!\{ C'^{(1)}_t(w) \mid w \in N_G(v) \}\!\!\}.$$

By (b) and (c), we conclude that $\{\!\!\{ C'^{(1)}_t(w) \mid w \in N_{G'}(u) \}\!\!\} = \{\!\!\{ C'^{(1)}_t(w) \mid w \in N_{G'}(v) \}\!\!\}$. Hence, we have $C'^{(1)}_{t+1}(u) = C'^{(1)}_{t+1}(v)$, completing the proof.

- 1. $\Rightarrow$ 4.: Assume $\delta^{\mathcal{T}}_{\|\cdot\|}(G', H') = \|A(G')S' - S'A(H')\| = 0$ for some doubly stochastic matrix $S'$. By Theorem 3.5.11 in Grohe (2017), if $C_\infty$ denotes the stable coloring under the 1-WL algorithm on $G', H'$, then $S'$ can be explicitly expressed as

$$S'_{i,j} = \begin{cases} 1/|V(G) \cap C_\infty^{-1}(c)|, & \text{if } C_\infty(i) = C_\infty(j) = c \text{ for some color } c, \\ 0, & \text{otherwise.} \end{cases}$$

The newly added vertices in $G'$ and $H'$ always have stable colors distinct from those of the original vertices since their degrees are strictly larger than $n + 1$, while the original vertices have degrees strictly smaller than $n + 1$. Consequently, $S'$ has a block diagonal structure

$$S' = \begin{bmatrix} S & 0 \\ 0 & \widetilde{S} \end{bmatrix},$$

where $S \in D_n$ and $\widetilde{S} \in D_m$. Substituting into the equation $\|A(G')S' - S'A(H')\| = 0$, we deduce that

$$\|A(G)S - SA(H)\| = 0.$$

The term $\mathrm{Tr}(S^\top L(G, H))$ can be written as

$$\mathrm{Tr}(S^\top L(G, H)) = \sum_{i \in V(G), j \in V(H)} S_{i,j} L(G, H)_{i,j}.$$

If $\sum_{i \in V(G), j \in V(H)} S_{i,j} L(G, H)_{i,j} \neq 0$, then, there exists $i \in V(G)$ and $j \in V(H)$ for which $S_{i,j} > 0$ and $\ell_G(i) \neq \ell_H(j)$ which is a contradiction since $S_{i,j} = S'_{i,j} > 0$ and if $\ell_G(i) \neq \ell_H(j)$, there would have been attached different vertices to vertices $i$ and $j$, implying that they would have different stable colors. Hence, $\delta^{\mathcal{T}}_{\Box}(G, H) = 0$.

- 4. $\Rightarrow$ 1.: Suppose $\widetilde{\delta}^{\mathcal{T}}_{\|\cdot\|}(G, H) = 0$. We construct a block diagonal doubly stochastic matrix $S' \in D_{n+m}$, consisting of two blocks in $D_n$ and $D_m$ respectively, such that $\|A(G')S' - S'A(H')\| = 0$.

Assume $\widetilde{\delta}^{\mathcal{T}}_{\Box}(G, H) = 0$. This implies that there exists $S \in D_n$, s.t. $\|A(G)S - SA(H)\| = 0$ and $\mathrm{Tr}(S^\top L(G, H)) = 0$.

We choose to order the vertices being ordered by their colors vertices in $G$ and $H$. That is, a vertex $u \in V(G)$ comes before vertex $v \in V(G)$ if the stable coloring satisfies $C_\infty(u) < C_\infty(v)$. Following the described ordering, and Theorem 3.5.11 in Grohe (2017), then $S$ is symmetric, satisfying $S_{i,j} > 0$ if, and only, if vertices $i, j$ have the same color when the 1-WL algorithm acts on $G$ and $H$, without considering initial labels. Moreover, $\mathrm{Tr}(S^\top L(G, H)) = 0$ implies that $S_{i,j} > 0 \implies \ell_G(i) = \ell_G(j)$. Therefore, if two vertices have the same stable color after 1-WL without considering initial labels, then these two vertices also have the same initial labels. This implies that the adjacency matrices of $G', H'$ can be written as follows,

$$A(G') = \begin{bmatrix} A(G) & B \\ B^\top & D \end{bmatrix},$$

$$A(H') = \begin{bmatrix} A(H) & B \\ B^\top & D \end{bmatrix},$$

where $\boldsymbol{B} \in \mathbb{R}^{n \times m}$, such that

$$\boldsymbol{B}_{i,j} = \begin{cases} 1, & i \in \{2, \ldots n\} \text{ and } j = 1 + \sum_{k=1}^{i-1} \theta(\ell_G(k)), \\ 0, & \text{otherwise}, \end{cases}$$

and $\boldsymbol{D} \in \mathbb{R}^{m \times m}$ is a block diagonal symmetric matrix of the attached cliques. We now define $\boldsymbol{S}'$ to be

$$\boldsymbol{S}' = \begin{bmatrix} \boldsymbol{S} & 0 \\ 0 & \widetilde{\boldsymbol{S}} \end{bmatrix},$$

where $\widetilde{\boldsymbol{S}}$ is defined as follows,

$$\widetilde{\boldsymbol{S}} = \begin{bmatrix} \boldsymbol{K}_{1,1} & \boldsymbol{K}_{1,1} & \cdots & \boldsymbol{K}_{1,n} \\ \boldsymbol{K}_{2,1} & \boldsymbol{K}_{2,2} & \cdots & \boldsymbol{K}_{2,n} \\ \vdots & \vdots & \ddots & \vdots \\ \boldsymbol{K}_{n,1} & \boldsymbol{K}_{n,2} & \cdots & \boldsymbol{K}_{n,n} \end{bmatrix},$$

where $\boldsymbol{K}_{i,j} \in \mathbb{R}^{\theta(\ell_G(i)) \times \theta(\ell_G(j))}$ is

$$\boldsymbol{K}_{i,j} = \begin{bmatrix} S_{i,j} & 0 & \cdots & 0 \\ 0 & 0 & \cdots & 0 \\ \vdots & \vdots & \ddots & \vdots \\ 0 & 0 & \cdots & 0 \end{bmatrix}.$$

It is then easy to verify that, $\widetilde{\boldsymbol{S}}$ is a doubly stochastic and symmetric matrix satisfying $\widetilde{\boldsymbol{S}} \boldsymbol{B}^{\mathsf{T}} = \boldsymbol{B}^{\mathsf{T}} \boldsymbol{S}$ which by symmetry implies $\boldsymbol{B} \widetilde{\boldsymbol{S}} = \boldsymbol{S} \boldsymbol{B}$. Finally, it is easy to verify that $\boldsymbol{D} \widetilde{\boldsymbol{S}} = \widetilde{\boldsymbol{S}} \boldsymbol{D}$, by showing that $\boldsymbol{D} \widetilde{\boldsymbol{S}}$ is symmetric and using the symmetry of $\boldsymbol{D}$ and $\widetilde{\boldsymbol{S}}$. All together, this implies

$$\boldsymbol{A}(G') \boldsymbol{S}' = \begin{bmatrix} \boldsymbol{A}(G)\boldsymbol{S} & \boldsymbol{B}\widetilde{\boldsymbol{S}} \\ \boldsymbol{B}^{\mathsf{T}}\boldsymbol{S} & \boldsymbol{D}\widetilde{\boldsymbol{S}} \end{bmatrix} = \begin{bmatrix} \boldsymbol{S}\boldsymbol{A}(H) & \boldsymbol{S}\boldsymbol{B} \\ \widetilde{\boldsymbol{S}}\boldsymbol{B}^{\mathsf{T}} & \widetilde{\boldsymbol{S}}\boldsymbol{D} \end{bmatrix} = \boldsymbol{S}'\boldsymbol{A}(H'),$$

as desired.

This concludes the proof. $\qquad\square$

## F. The Tree Mover's distance

The following introduces the Tree Mover's Distance (TMD), defined through the optimal transport problem. Following the notation from the original paper by Chuang & Jegelka (2022), we introduce the optimal transport problem and the Wasserstein distance. We then define an optimal transport problem between the sets of unrolled computation trees corresponding to the vertices of two labeled graphs. Next, we define a distance between labeled rooted trees and extend this definition to the space of labeled graphs, using the optimal transport problem and the unrolling trees of graphs' vertices.

**The Optimal Transport** We begin with a brief introduction to *Optimal Transport* (OT) and the *Wasserstein distance*. Let $X = \{x_i\}_{i=1}^m$ and $Y = \{y_j\}_{j=1}^m$ be two multisets of $m$ elements each. Let $\boldsymbol{C} \in \mathbb{R}^{m \times m}$ be the transportation cost for each pair, i.e., $C_{ij} = d(x_i, y_j)$, where $d$ is a distance function between $X$ and $Y$. The Wasserstein distance is defined through the following minimization problem,

$$\mathrm{OT}_d^*(X, Y) := \min_{\boldsymbol{\gamma} \in \Gamma(X,Y)} \frac{\langle \boldsymbol{C}, \boldsymbol{\gamma} \rangle}{m}, \quad \Gamma(X, Y) = \{\boldsymbol{\gamma} \in \mathbb{R}_+^{m \times m} \mid \boldsymbol{\gamma} \mathbf{1}_m = \boldsymbol{\gamma}^\top \mathbf{1}_m = \mathbf{1}_m\},$$

where $\langle \cdot, \cdot \rangle$ is defined as the sum of the elements resulting from the entry-wise multiplication of two matrices with equal dimensions, and $\mathbf{1}_m$ denotes the all ones $m$-dimensional column vector. In the following, we use the unnormalized version of the Wasserstein distance,

$$\mathrm{OT}_d(X, Y) := \min_{\boldsymbol{\gamma} \in \Gamma(X,Y)} \langle \boldsymbol{C}, \boldsymbol{\gamma} \rangle = m \cdot \mathrm{OT}_d^*(X, Y).$$

**Distance between rooted trees via hierarchical OT**  Let $(T, r)$ denote a rooted tree. We further let $\mathcal{T}_v$ be the multiset of subtrees of $T$ consisting of subtrees rooted at the children of $v$. Determining whether two trees are similar requires iteratively examining whether the subtrees in each level are similar. By recursively computing the optimal transportation cost between their subtrees, we define the distance between two rooted trees $(T_a, r_a), (T_b, r_b)$. However, the number of subtrees could be different for $r_a$ and $r_b$, i.e., $|\mathcal{T}_{r_a}| \neq |\mathcal{T}_{r_b}|$. To compute the OT between sets with different sizes, we augment the smaller set with *blank trees*.

**Definition 25.** A blank tree $T_0$ is a tree (graph) that contains a single vertex and no edge, where the vertex feature is the zero vector $\mathbf{0}_p \in \mathbb{R}^p$, and $T_0^n$ denotes a multiset of $n$ blank trees.

**Definition 26.** Given two multisets of trees $\mathcal{T}_u, \mathcal{T}_v$, define $\rho$ to be a function that augments a pair of trees with blank trees as follows,

$$\rho \colon (\mathcal{T}_v, \mathcal{T}_u) \to \left( \mathcal{T}_v \cup T_0^{\max(|\mathcal{T}_u| - |\mathcal{T}_v|, 0)}, \ \mathcal{T}_u \cup T_0^{\max(|\mathcal{T}_v| - |\mathcal{T}_u|, 0)} \right).$$

We now recursively define a distance between rooted trees using the optimal transport problem.

**Definition 27.** The distance between two trees rooted $(T_a, r_a), (T_b, r_b)$ is defined recursively as:

$$\mathrm{TD}_\omega(T_a, T_b) := \begin{cases} \|\boldsymbol{x}_{r_a} - \boldsymbol{x}_{r_b}\| + \omega(L) \cdot \mathrm{OT}_{\mathrm{TD}_w}(\rho(T_{r_a}, T_{r_b})) & \text{if } L > 1, \\ \|\boldsymbol{x}_{r_a} - \boldsymbol{x}_{r_b}\| & \text{otherwise,} \end{cases}$$

where $\|\cdot\|$ is a norm on the feature space, $L = \max(\mathrm{Depth}(T_a), \mathrm{Depth}(T_b))$ and $\omega \colon \mathbb{N} \to \mathbb{R}^+$ is a depth-dependent weighting function.

**Extension to labeled graphs**  Below, we extend the above distance between rooted trees to the TMD on labeled graphs by calculating the optimal transportation cost between the graphs' unrolling trees.

**Definition 28.** Given two graphs $G, H$ and $L > 0$, the tree mover's distance between $G$ and $H$ is defined as

$$\mathrm{TMD}_\omega^L(G, H) = \mathrm{OT}_{\mathrm{TD}_\omega}(\rho(\mathcal{T}_G^L, \mathcal{T}_H^L)),$$

where $\mathcal{T}_G^L$ and $\mathcal{T}_H^L$ are multisets of the depth-$L$ unrolling trees of graphs $G$ and $H$, respectively.

# G. Mean-Forest distance

**Normalized Tree Mover's distance**  We can scale the vertex features of the forests to extend the definition of the Forest distance to the normalized TMD; see Chuang & Jegelka (2022). This distance satisfies the Lipschitz property of mean aggregation MPNNs, as shown in Chuang & Jegelka (2022, Appendix B.). However, this distance is strictly more expressive than the mean-1-WL algorithm, meaning that there are graphs that are not distinguished by the 1-MWL but have a positive normalized TMD. To see this, consider the distance between two star graphs, one with a root labeled 1 and two neighbors labeled 1 and 2, and the other with a root labeled 1 and four neighbors, two of which are labeled 1 and the other two with 2. It is clear that the 1-MWL cannot distinguish these two simple graphs. However, the normalized TMD between these graphs is positive because the number of vertices at the first level differs. This motivates us to define the mean-Forest distance through the mean unrollings defined in Section 1.1.

**Mean-Forest distance**  Here, we define the *mean-Forest distance*, providing a pseudo-metric aligned with the mean-aggregation MPNN models and preserving the expressivity of the 1-MWL. That is, for a given attributed $n$-order graph $(G, \ell_G)$ with labels in $\mathbb{R}^{1 \times d}$ and a fixed $L \in \mathbb{N}$, consider the following multiset of mean-unrolling trees,

$$\bar{\mathcal{T}}_G^L = \{\!\!\{ \tau\,(\mathsf{m\text{-}unr}(G, u, L)) \mid u \in V(G) \}\!\!\}.$$

In analogy to the use of the normalized Wasserstein distance to define the normalized TMD in Chuang & Jegelka (2022), here we scale all vertex features to align our distance with the mean aggregation scheme described by Equation (5), thereby ensuring the Lipschitz property. For each vertex $u$ in each tree, we divide its vertex feature by $s(u)$, where $s(u)$ is defined inductively as follows. For the root $r$, $s(r) = 1$. For any other vertex $u$, with parent $v$, we define $s(u)$ as the product of the number of children of $v$ and $s(v)$. Next, we perform a padding process on all trees in $\bar{\mathcal{T}}_G^L$ as follows. For each vertex of each tree, we add children with the vertex feature $\mathbf{o}$ until each vertex has exactly $n - 1$ children. We then construct the forest consisting of all $n$ trees and denote this forest by $F_{G,L}^{\mathrm{m}}$.

The mean-Forest distance between two labeled graphs $G$ and $H$ is defined as

$$\mathrm{FD}_L^{\mathrm{m}}(G, H) = \min_{\varphi} \sum_{u \in V(F_{G,L}^{\mathrm{m}})} \|\ell_{\mathcal{F}_{G,L}^{\mathrm{m}}}(u) - \ell_{\mathcal{F}_{H,L}^{\mathrm{m}}}(\varphi(u))\|_2,$$

where the minimum is taken over all edge-preserving bijections between $V(F_{G,L}^{\mathrm{m}})$ and $V(F_{H,L}^{\mathrm{m}})$. Now, the following result shows that the mean-Forest distance is a valid pseudo-metric.

**Lemma 29.** *The mean-Forest distance* $\mathrm{FD}_L^{\mathrm{m}}$ *is a well-defined pseudo-metric on* $\mathcal{G}_{n,d}^{\mathbb{R}}$, *for* $L \in \mathbb{N}$. *Additionally, for two graphs* $G, H$, $\mathrm{FD}_L^{\mathrm{m}}(G, H) = 0$ *if, and only, if* $G, H$ *are* 1-*MWL indistinguishable after* $L$ *iterations.* $\square$

## H. Properties of the Forest distance and the mean-Forest distance

Here, we prove the main results regarding the Forest distance and mean-Forest distance as defined in Section 2. We start by showing the equivalence between the Forest distance and the Tree Mover's distance, as stated in Lemma 3. Next, we establish that the mean-Forest distance is a well-defined pseudo-metric on the set of graphs and has equivalent expressivity to 1-MWL; see Lemma 29. We then show the Lipschitz-continuity property of sum and mean aggregation MPNNs concerning the Forest distance and mean-Forest distance, respectively, i.e., Lemma 14 and Lemma 48. Finally, we derive an upper bound for the covering number of graph class $\mathcal{G}_{n,d,q}$ regarding the Forest distance, as stated in Proposition 16.

We begin by showing the equivalence between the Forest distance and the TMD. Specifically, we prove Lemma 3 using a recursive formula for the Forest distance for the simple case where the weight $\omega \equiv 1$. The proof for general weight function $\omega$ remains the same. To proceed, we will introduce some further notations. Recall the definition of the Forest distance, for $L \in \mathbb{N}$ and graphs $G, H \in \mathcal{G}_{n,d}^{\mathbb{R}}$, the Forest distance

$$\mathrm{FD}_L(G, H) = \min_{\varphi} \sum_{u \in V(F_{G,L})} \|\ell_G(u) - \ell_H(\varphi(u))\|_2,$$

where $F_{G,L}$ denotes the forest consisting of all padded trees in $\mathcal{T}_G^L$, and the minimum is taken over all edge-preserving bijections $\varphi$ between $V(F_{G,L})$ and $V(F_{H,L})$. We denote the set of forests consisting of depth-$L$ complete trees (i.e., non-leaf vertex has exactly $n - 1$ children), with vertex labels from $\mathbb{R}^d$, by $\mathcal{F}_L$. For two forests $F_1, F_2 \in \mathcal{F}_L$, we define $\overline{\mathsf{iso}}(F_1, F_2)$ as the set of edge-preserving bijections between $V(F_1)$ and $V(F_2)$. If $F_1$ and $F_2$ contain a different number of trees, we pad the smaller forest with depth-$L$ complete trees with $\mathbf{o}$ vertex labels. The *forest transport function* $\mathrm{FT}_L$ between two forests in $\mathcal{F}_L$ is then defined as

$$\mathrm{FT}_L(F_1, F_2) \coloneqq \min_{\varphi \in \overline{\mathsf{iso}}(F_1, F_2)} \left( \sum_{u \in V(F_1)} \|\boldsymbol{x}_u^{(F_1)} - \boldsymbol{x}_{\varphi(u)}^{(F_2)}\|_2 \right),$$

where $\boldsymbol{x}_u^{(F_j)} \in \mathbb{R}^{1 \times d}$ represents the label of vertex $u$ in $F_j$ for $j \in \{1,2\}$. Observing this, we immediately find that

$$\mathrm{FD}_L(G, H) = \mathrm{FT}_L(F_{G,L}, F_{H,L}).$$

Now, for a rooted tree $T$ with root $r$, let $\mathcal{T}_r$ denote the multiset containing all subtrees rooted at the children of $r$ in $T$, and let $F_r$ represent the forest of padded (with $\mathbf{o}$ labeled vertices) subtrees rooted at $r$'s children. Additionally, given a forest $F$ composed of rooted trees, denote by $\mathsf{roots}(F)$ the set of roots of the trees in $F$. The following lemma provides a recursive formula for the Forest distance.

**Lemma 30.** *For forests* $F_1, F_2 \in \mathcal{F}_L$,

$$\mathrm{FT}_L(F_1, F_2) = \min_{\substack{\sigma:\, \mathsf{roots}(F_1) \to \mathsf{roots}(F_2) \\ \textit{bijection}}} \left( \sum_{r \in \mathsf{roots}(F_1)} \left( \|\boldsymbol{x}_r^{(F_1)} - \boldsymbol{x}_{\sigma(r)}^{(F_2)}\|_2 + \mathrm{FT}_{L-1}(F_r, F_{\sigma(r)}) \right) \right).$$

*Proof.* We prove the lemma by substituting the definition of $\mathrm{FT}_{L-1}(F_r, F_{\sigma(r)})$ into the recursive formula. By definition of the forest transport function, we have,

$$\mathrm{FT}_{L-1}(F_r, F_{\sigma(r)}) = \min_{\varphi \in \overline{\mathsf{iso}}(F_r, F_{\sigma(r)})} \left( \sum_{u \in V(F_r)} \|\boldsymbol{x}_u^{(F_1)} - \boldsymbol{x}_{\varphi(u)}^{(F_2)}\|_2 \right).$$

Substituting this into the recursive formula yields that

$$\min_{\substack{\sigma:\,\text{roots}(F_1)\to\text{roots}(F_2) \\ \text{bijection}}} \left( \sum_{r\in\text{roots}(F_1)} \left( \|\boldsymbol{x}_r^{(F_1)} - \boldsymbol{x}_{\sigma(r)}^{(F_2)}\|_2 + \text{FT}_{L-1}(F_r, F_{\sigma(r)}) \right) \right)$$

is equal to

$$\min_{\substack{\sigma:\,\text{roots}(F_1)\to\text{roots}(F_2) \\ \text{bijection}}} \left( \sum_{r\in\text{roots}(F_1)} \left( \|\boldsymbol{x}_r^{(F_1)} - \boldsymbol{x}_{\sigma(r)}^{(F_2)}\|_2 \right.\right.$$
$$\left.\left. + \min_{\varphi\in\overline{\text{iso}}(F_r, F_{\sigma(r)})} \left( \sum_{u\in V(F_{r,F_1})} \|\boldsymbol{x}_u^{(F_r)} - \boldsymbol{x}_{\varphi(u)}^{(F_{\sigma(r)})}\|_2 \right) \right) \right)$$

which is equal to

$$\min_{\substack{\sigma:\,\text{roots}(F_1)\to\text{roots}(F_2) \\ \text{bijection}}} \left( \sum_{r\in\text{roots}(F_1)} \min_{\varphi\in\overline{\text{iso}}(F_r, F_{\sigma(r)})} \left( \|\boldsymbol{x}_r^{(F_1)} - \boldsymbol{x}_{\sigma(r)}^{(F_2)}\|_2 + \sum_{u\in V(F_r)} \|\boldsymbol{x}_u^{(F_r)} - \boldsymbol{x}_{\varphi(u)}^{(F_{\sigma(r)})}\|_2 \right) \right),$$

which in turn is equal to $\text{FT}_L(F_1, F_2)$. Here, the last equality follows from the fact that computing a bijection between the roots of the trees in two forests and then computing edge-preserving bijections between the subtrees of these roots is equivalent to directly computing edge-preserving bijections between the two forests. $\qquad\square$

Next, using Lemma 30, we show the equivalence of Forest distance and TMD.

**Lemma 31** (Lemma 3 (for $\omega\equiv 1$) in the main paper)**.** *The Forest distance is equivalent to the TMD when $\omega\equiv 1$. That is, for graphs $G, H\in\mathcal{G}_{n,d}^{\mathbb{R}}$,*
$$\text{TMD}^L(G, H) = \text{FD}_L(G, H).$$

*Proof.* We prove by induction on $L$ that if $\mathcal{T}_1$ and $\mathcal{T}_2$ are multisets of $n$ depth-$L$ trees, and $F_1$ and $F_2$ are the padded forests induced by $\mathcal{T}_1$ and $\mathcal{T}_2$, respectively, then
$$\text{OT}_{\text{TD}}(\rho(\mathcal{T}_1, \mathcal{T}_2)) = \text{FT}_L(F_1, F_2),$$

where OT, TD, and $\rho$ denote the optimal transport problem, Tree distance, and padding function, respectively, as defined in Appendix F. Setting $\mathcal{T}_1 = \mathcal{T}_G^L$ and $\mathcal{T}_2 = \mathcal{T}_H^L$ implies the equivalency between the two distances. For $L = 0$, consider multisets $\mathcal{T}_1, \mathcal{T}_2$ of trivial "trees," i.e., isolated vertices. By the optimal transport problem definition, if we assume $\mathcal{T}_1 = \mathcal{T}_2 = [n]$, we have
$$\text{OT}_{\text{TD}}(\rho(\mathcal{T}_1, \mathcal{T}_2)) = \min_{\boldsymbol{S}\in D_n} \left( \sum_{u\in\mathcal{T}_1, v\in\mathcal{T}_2} S_{u,v}\|\boldsymbol{x}_u^{(G)} - \boldsymbol{x}_v^{(H)}\|_2 \right),$$

which is a minimization problem with a convex feasible set and linear objective function. Therefore, the minimum is attained on an extreme point. The Birkhoff-von Neumann theorem shows that the extreme points for doubly stochastic matrices are the permutation matrices. Since $\varphi\in\overline{\text{iso}}(F^{G,L}, F^{H,L})$ is equivalent to choosing a permutation matrix, we conclude that,

$$\text{OT}_{\text{TD}}(\rho(\mathcal{T}_1, \mathcal{T}_2)) = \min_{\boldsymbol{P}\in P_n} \left( \sum_{u\in\mathcal{T}_1, v\in\mathcal{T}_2} P_{u,v}\|\boldsymbol{x}_u^{(G)} - \boldsymbol{x}_v^{(H)}\|_2 \right) = \text{FT}_0(F_1, F_2).$$

For the inductive hypothesis, we assume that
$$\text{OT}_{\text{TD}}(\rho(\mathcal{T}_1, \mathcal{T}_2)) = \text{FT}_{L-1}(F_1, F_2),$$

for multisets $\mathcal{T}_1, \mathcal{T}_2$ of $(L-1)$-depth trees and their corresponding forests $F_1, F_2$. Now, let us consider two multisets of $L$-depth trees, $\mathcal{T}_1$ and $\mathcal{T}_2$, with associated forests $F_1$ and $F_2$. With slight abuse of notation, we introduce a canonical ordering

for the elements in the multisets $\mathcal{T}_1$ and $\mathcal{T}_2$ such that each $t \in \mathcal{T}_1$ can be used interchangeably as a tree or as an index for the entries of the permutation matrix. Additionally, for each rooted tree $t \in \mathcal{T}_1$, we denote the root by $r_t$. We compute the optimal transport $\mathrm{OT}_{\mathrm{TD}}(\rho(\mathcal{T}_1, \mathcal{T}_2))$ using the definition from Appendix F as follows,

$$
\begin{aligned}
\mathrm{OT}_{\mathrm{TD}}(\rho(\mathcal{T}_1, \mathcal{T}_2)) &= \min_{\boldsymbol{P} \in P_n} \left( \sum_{t \in \mathcal{T}_1, t' \in \mathcal{T}_2} P_{t,t'} \, \mathrm{TD}(t, t') \right) \\
&= \min_{\boldsymbol{P} \in P_n} \left( \sum_{t \in \mathcal{T}_1, t' \in \mathcal{T}_2} P_{t,t'} \left( \| \boldsymbol{x}_{r_t}^{(t)} - \boldsymbol{x}_{r_{t'}}^{(t')} \|_2 + \mathrm{OT}_{\mathrm{TD}}(\rho(\mathcal{T}_{r_t}, \mathcal{T}_{r_{t'}})) \right) \right) \\
&= \min_{\boldsymbol{P} \in P_n} \left( \sum_{t \in \mathcal{T}_1, t' \in \mathcal{T}_2} P_{t,t'} \left( \| \boldsymbol{x}_{r_t}^{(t)} - \boldsymbol{x}_{r_{t'}}^{(t')} \|_2 + \mathrm{FT}_{L-1}(F_{r_t}, F_{r_{t'}}) \right) \right) \\
&= \mathrm{FD}_L(F_1, F_2),
\end{aligned}
$$

where the last two equalities follow from the inductive hypothesis and from Lemma 30, respectively. □

The following results relate the mean Forest distances to the 1-MWL in distinguishing non-isomorphic graphs.

**Lemma 32** (Lemma 29 in Appendix G). *For $L \in \mathbb{N}$, the mean-Forest distance $\mathrm{FD}_L^{(m)}$ is a well-defined pseudo-metric on $\mathcal{G}_{n,d}^{\mathbb{R}}$. Additionally, $\mathrm{FD}_L^{(m)}(G, H) = 0$ if, and only, if $G, H$ are 1-MWL indistinguishable after $L$ iterations.*

*Proof.* We recall the definition of the mean-Forest distance between two graphs $G, H \in \mathcal{G}_{n,d}^{\mathbb{R}}$, which can be written as

$$
\mathrm{FD}_L^{(m)}(G, H) = \min_{\substack{\boldsymbol{P} \in P_{n_F} \\ \boldsymbol{A}(F_{G,L}^{(m)})\boldsymbol{P} = \boldsymbol{P}\boldsymbol{A}(F_{H,L}^{(m)})}} (\mathrm{Tr}(\boldsymbol{P}^{\mathsf{T}} \boldsymbol{L}(G, H))),
$$

where $L_{u,v}^{(G,H)} = \| x_u^{(G)} - x_v^{(H)} \|_2$, and $n_F$ is the total number of vertices in the forest after the padding process i.e., $n_F = (n-1)^{L+1} - 1$.

It is easy to verify that $\mathrm{FD}_L^{(m)}(G, G) = 0$ and $\mathrm{FD}_L^{(m)}(G, H) = \mathrm{FD}_L^{(m)}(H, G)$ for all $G, H \in \mathcal{G}_{n,w}^{\mathbb{R}}$. Now, for the triangle inequality, consider $G_A, G_B, G_C \in \mathcal{G}_{n,w}^{\mathbb{R}}$ and $\boldsymbol{P}', \boldsymbol{P}''$ as the permutation matrices arising from the minimization formula of $\mathrm{FD}_L^{(m)}(G_A, G_B)$ and $\mathrm{FD}_L^{(m)}(G_B, G_C)$, respectively. Note that $\boldsymbol{P}'\boldsymbol{P}''$ is a permutation matrix satisfying

$$
\boldsymbol{A}(F_{G_A,L}^{(m)})\boldsymbol{P}'\boldsymbol{P}'' = \boldsymbol{P}'\boldsymbol{A}(F_{G_B,L}^{(m)})\boldsymbol{P}'' = \boldsymbol{P}'\boldsymbol{P}''\boldsymbol{A}(F_{G_C,L}^{(m)}),
$$

implying that

$$
\mathrm{FD}_L^{(m)}(G_A, G_C) \le \mathrm{Tr}((\boldsymbol{P}'\boldsymbol{P}'')^{\mathsf{T}}\boldsymbol{L}(G_A, G_C)).
$$

Following the same derivations as in Proposition 23, we show that

$$
\mathrm{Tr}((\boldsymbol{P}'\boldsymbol{P}'')^{\mathsf{T}}\boldsymbol{L}(G_A, G_C)) \le \mathrm{Tr}((\boldsymbol{P}')^{\mathsf{T}}\boldsymbol{L}(G_A, G_B)) + \mathrm{Tr}((\boldsymbol{P}'')^{\mathsf{T}}\boldsymbol{L}(G_B, G_C)),
$$

which proves the triangle inequality.

The additional property: $\mathrm{FD}_L^{(m)}(G, H) = 0$ if, and only if, $G$ and $H$ are 1-MWL indistinguishable after $L$ iterations, follows directly from the unrolling characterization of 1-MWL as described in Proposition 18.

□

**Computational complexity** Throughout the paper, we discuss two main distances: the tree distance (with either the cut norm or the $L^2$-norm) and the forest distance. As shown in Lemma 3, the forest distance is equivalent to the Tree Mover's Distance (TMD) introduced by Chuang & Jegelka (2022). This equivalence allows for efficient computation via dynamic programming (as described in the original paper), with a time complexity of $O(t(n) + Lnt(q))$, where $q$ is the maximum degree of a node (in both graphs), $n$ is the number of nodes, and $t(m) = O(m^3 \log(m))$.

The tree distance with the $L^2$-norm can be computed in polynomial time (specifically, an $\epsilon$-approximation), as detailed in (Nesterov & Nemirovskii, 1994, Section 6.3.3). In contrast, computing the tree distance with the cut norm involves a significantly harder combinatorial optimization problem. One of the most well-known approximation techniques for this case is described in (Alon & Naor, 2004). Additional relevant works include Alon et al. (2002); Frieze & Kannan (1999). It is worth mentioning that for the theoretical results in our paper, either the tree distance with the cut norm or with the 2-norm can be used interchangeably, as they define the same topology as discussed in (Böker et al., 2023).

## I. Extended robustness framework

Kawaguchi et al. (2022) improved the generalization bound in Theorem 4 by establishing a data-dependent bound that reduces the dependency on $K$ from $\sqrt{K}$ to $\log(K)$, as shown in the following theorem.

**Theorem 33** (Theorem 3 in Kawaguchi et al. (2022))**.** *For any $(K, \epsilon)$-robust (graph) learning algorithm for $\mathcal{H}$ on $\mathcal{Z}$, with partition $\{C_k\}_{k=1}^{K(\mathcal{S})}$, we have that for all $\delta \in (0,1)$, with probability at least $1 - \delta$, for a sample $\mathcal{S}$ drawn from $\mathcal{Z}$ according to $\mu$, it holds that*

$$\ell_{\exp}(h_\mathcal{S}) \leq \ell_{\mathrm{emp}}(h_\mathcal{S}) + \epsilon(\mathcal{S}) + \zeta(h_\mathcal{S})\left(\left(\sqrt{2} + 1\right)\sqrt{\frac{|\mathcal{T}_\mathcal{S}|\log(2K/\delta)}{|\mathcal{S}|}} + \frac{2|\mathcal{T}_\mathcal{S}|\log(2K/\delta)}{|\mathcal{S}|}\right),$$

*where $\mathcal{I}_k^\mathcal{S} = \{i \in [|\mathcal{S}|] \mid (x_i, y_i) \in C_i\}$, $\zeta(h_\mathcal{S}) = \max_{(x,y) \in \mathcal{Z}}\{\ell(h_\mathcal{S}(x), y))\}$, and $\mathcal{T}_\mathcal{S} = \{k \in [K] \mid |\mathcal{I}_k^\mathcal{S}| \geq 1\}$.* $\square$

Here, we introduce an extended definition of robustness that allows us to choose different radii for the various partition sets. This modification makes the first term of the generalization bounds in Theorem 4 data-dependent.

**Definition 34.** A (graph) learning algorithm for $\mathcal{H}$ is $(K, \boldsymbol{\varepsilon})$-*robust*, with $K$, $\boldsymbol{\varepsilon} = (\varepsilon_1, \ldots \varepsilon_K)$ mapping from the set of all possible samples to $\mathbb{N}$ and $(0, \infty)^{K(\mathcal{S})}$, respectively, if $\mathcal{Z}$ can be partitioned into $K(\mathcal{S})$ sets, $\{C_i\}_{i=1}^{K(\mathcal{S})}$, such that for all samples $\mathcal{S}$, and for any $(G, y) \in \mathcal{S}$, the following holds. If $(G, y) \in C_i$ for some $i \in \{1, \ldots, K\}$, then for all $(G', y') \in C_i$, we have

$$\left|\ell(h_\mathcal{S}(G), y) - \ell(h_\mathcal{S}(G'), y')\right| < \varepsilon_i(\mathcal{S}),$$

where $h_\mathcal{S}$ is the graph embedding the learning algorithm returns concerning the data sample $\mathcal{S}$.

Building on our refined definition of robustness, we derive the following generalization bound. The proof is nearly identical to that of Theorem 3 in Xu & Mannor (2012); however, for completeness, we provide the proof in Appendix J.

**Theorem 35.** *For any $(K, \boldsymbol{\varepsilon})$-robust (graph) learning algorithm for $\mathcal{H}$ on $\mathcal{Z}$, we have that for all $\delta \in (0,1)$, with probability at least $1 - \delta$,*

$$|\ell_{\exp}(h_\mathcal{S}) - \ell_{\mathrm{emp}}(h_\mathcal{S})| \leq \sum_{i=1}^{K} \frac{|\{s \in \mathcal{S} \mid s \in C_i\}|}{|\mathcal{S}|} \epsilon_i(\mathcal{S}) + M\sqrt{\frac{2K\log(2) + 2\log(1/\delta)}{|\mathcal{S}|}}.$$

*where $h_\mathcal{S}$, as before, denotes a graph embedding from $\mathcal{H}$ returned by the learning algorithm given the data sample $\mathcal{S}$ of $\mathcal{Z}$. We recall that $M$ is the bound on the loss function $\ell$.* $\square$

Based on Theorem 35 and the proof technique outlined in Theorem 10, we present the following generalization bound for the class of unlabeled graphs consisting of at most $n$ vertices (i.e., $\mathcal{G}_{\leq n}$)

**Proposition 36.** *For $n, L \in \mathbb{N}, M' \in \mathbb{R}$, for any graph learning algorithm for the class $\mathrm{MPNN}_{L, M', L_{\mathsf{FNN}}}^{\mathrm{ord}}(\mathcal{G}_{\leq n})$ and for any sample $\mathcal{S}$ and $\delta \in (0,1)$, with probability at least $1 - \delta$, we have*

$$|\ell_{\exp}(h_\mathcal{S}) - \ell_{\mathrm{emp}}(h_\mathcal{S})| \leq L_\ell \cdot L_{\mathsf{FNN}} \cdot \sum_{i=1}^{n} \frac{|\{s \in \mathcal{S} \mid s \in \mathcal{G}_i \times \mathcal{Y}\}|}{|\mathcal{S}|} \overline{\gamma}^{\leftarrow}\left(\frac{8k}{i^2}\right) + M\sqrt{\frac{4^{m_{\leq n}/k + 1}\log(2) + 2\log(1/\delta)}{|\mathcal{S}|}},$$

*for $k \in \mathbb{N}$, where $M$ is an upper bound on the loss function $\ell$, and $m_{\leq n} := |\mathcal{G}_{\leq n}/{\sim_{\mathsf{WL}}}|$.* $\square$

## J. Missing proofs from Section 3

The following results extends the robustness based generalization bounds utilizing the generalized robustness property.

**Theorem 37** (Theorem 35 in Appendix I). *If we have a $(K, \varepsilon)$-robust (graph) learning algorithm for $\mathcal{H}$ on $\mathcal{Z}$, then for all $\delta \in (0,1)$, with probability at least $1 - \delta$, for a sample $\mathcal{S}$ drawn from $\mathcal{Z}$ according to $\mu$, it holds that*

$$|\ell_{\exp}(h_{\mathcal{S}}) - \ell_{\emp}(h_{\mathcal{S}})| \leq \sum_{i=1}^{K(\mathcal{S})} \frac{|\{s \in \mathcal{S} \mid s \in C_i\}|}{|\mathcal{S}|} \epsilon_i(\mathcal{S}) + M\sqrt{\frac{2K(\mathcal{S})\log(2) + 2\log(1/\delta)}{|\mathcal{S}|}}.$$

*where $h_{\mathcal{S}}$, as before, denotes a graph embedding from $\mathcal{H}$ returned by the learning algorithm given the data sample $\mathcal{S}$ of $\mathcal{Z}$. We recall that $M$ is the bound on the loss function $\ell$.*

*Proof.* Let $N_i = \{s \in \mathcal{S} \mid s \in C_i\}$. Note that $(|N_1|, \ldots, |N_K|)$ is an i.i.d multinomial random variable with parameters $|\mathcal{S}|$ and $(\mu(C_1), \ldots, \mu(C_K))$, where $\mu$ is the unknown distribution on $\mathcal{Z}$ from which our sample is drawn. The following holds by the Bretagnolle-Huber-Carol inequality (see Proposition A.6 of van der Vaart et al. (1996)),

$$\Pr\left( \sum_{i=1}^{K} \left( \frac{|N_i|}{|\mathcal{S}|} - \mu(C_i) \right) \geq \lambda \right) \leq 2K \exp\left( -\frac{n\lambda^2}{2} \right).$$

Thus, with probability at least $1 - \delta$, we have:

$$\sum_{i=1}^{K} \left| \frac{|N_i|}{|\mathcal{S}|} - \mu(C_i) \right| \leq \sqrt{\frac{2K\log 2 + 2\ln(1/\delta)}{|\mathcal{S}|}}.$$

The discrepancy between the expected and empirical losses can be bounded as follows:

$$|\ell_{\exp}(h_{\mathcal{S}}) - \ell_{\emp}(h_{\mathcal{S}})|$$
$$= \left| \sum_{i=1}^{K} \mathbb{E}(\ell(h_{\mathcal{S}}, z)|z \in C_i)\mu(C_i) - \frac{1}{|\mathcal{S}|} \sum_{s \in \mathcal{S}} \ell(h_{\mathcal{S}}, s) \right|$$
$$\leq \left| \sum_{i=1}^{K} \mathbb{E}(\ell(h_{\mathcal{S}}, z)|z \in C_i)\frac{|N_i|}{|\mathcal{S}|} - \frac{1}{|\mathcal{S}|} \sum_{s \in \mathcal{S}} \ell(h_{\mathcal{S}}, s) \right|$$
$$+ \left| \sum_{i=1}^{K} \mathbb{E}(\ell(h_{\mathcal{S}}, z)|z \in C_i)\mu(C_i) - \sum_{i=1}^{K} \mathbb{E}(\ell(h_{\mathcal{S}}, z)|z \in C_i)\frac{|N_i|}{|\mathcal{S}|} \right|$$
$$\leq \left| \frac{1}{|\mathcal{S}|} \sum_{i=1}^{K} \sum_{s \in N_i} \max_{z_2 \in C_i} |\ell(h_{\mathcal{S}}, s) - \ell(h_{\mathcal{S}}, z_2)| \right| + \left| \max_{z \in \mathcal{Z}} |\ell(h_{\mathcal{S}}, z)| \sum_{i=1}^{K} \left| \frac{|N_i|}{|\mathcal{S}|} - \mu(C_i) \right| \right|$$
$$\leq \sum_{i=1}^{K} \frac{|\{s \in \mathcal{S} \mid s \in C_i\}|}{|\mathcal{S}|} \epsilon_i(\mathcal{S}) + M\sqrt{\frac{2K\log(2) + 2\log(1/\delta)}{|\mathcal{S}|}},$$

where the first inequality is due to triangle inequality, the second by definition of $N_i$, the third by definition of the generalized robustness property and the bound $M$. $\qquad \square$

The following result links continuity and robustness.

**Proposition 38** (Proposition 5 in the main paper). *Let $(\mathcal{G}, d_{\mathcal{G}})$, $(\mathcal{X}, d_{\mathcal{X}})$, and $(\mathcal{Y}, d_{\mathcal{Y}})$ be pseudo-metric spaces, and let $\mathcal{H}$ denote the class of uniformly continuous graph embeddings from $(\mathcal{G}, d_{\mathcal{G}})$ to $(\mathcal{X}, d_{\mathcal{X}})$. If $\ell : \mathcal{X} \times \mathcal{Y} \to \mathbb{R}^+$ is a $c_\ell$-Lipschitz continuous loss function, regarding $d_\infty$, then for any $\varepsilon > 0$, a graph learning algorithm for $\mathcal{H}$ on $\mathcal{G} \times \mathcal{Y}$ is*

$$\left( \mathcal{N}\big(\mathcal{G}, d_{\mathcal{G}}, \gamma/2\big) \cdot \mathcal{N}(\mathcal{Y}, d_{\mathcal{Y}}, \varepsilon/2), c_\ell\varepsilon \right)\text{-robust.}$$

*Where $\gamma(\cdot, h)$ be the positive function $\gamma$ used in definition of the uniform continuity of $h \in \mathcal{H}$*

*Proof.* Let $\{C_i\}_{i=1}^{K_1}$ be a partition corresponding to a $\gamma(\varepsilon, h_{\mathcal{S}})/2$-cover of $\mathcal{G}$, and let $\{Y_j\}_{j=1}^{K_2}$ be a partition of $\mathcal{Y}$ corresponding to an $\varepsilon/2$-cover of $\mathcal{Y}$. Then we have $K_1 = \mathcal{N}\big(\mathcal{G}, d_{\mathcal{G}}, \gamma(\varepsilon, h_{\mathcal{S}})/2\big)$ and $K_2 = \mathcal{N}(\mathcal{Y}, d_{\mathcal{Y}}, \varepsilon/2)$. Letting $K = K_1 K_2$, we

define the partition $D_{i,j} := C_i \times Y_j$, for $i \in [K_1]$ and $j \in [K_2]$ of $\mathcal{G} \times \mathcal{Y}$. We now verify that this partition satisfies the robustness conditions.

To this end, consider a sample $(G, y) \in \mathcal{S}$ from $\mathcal{G} \times \mathcal{Y}$ and assume $(G, y) \in D_{i,j}$, along with another element $(G', y') \in D_{i,j}$. We need to show that

$$\left| \ell\big(h_{\mathcal{S}}(G), y\big) - \ell\big(h_{\mathcal{S}}(G'), y'\big) \right| \le c_\ell \varepsilon.$$

Since $G$ and $G'$ both belong to $C_i$, we have $d_{\mathcal{G}}(G, G') \le \gamma(\varepsilon, h_{\mathcal{S}})$ and this implies that $d_{\mathcal{X}}\big(h_{\mathcal{S}}(G), h_{\mathcal{S}}(G')\big) \le \varepsilon$. Similarly, since $y$ and $y'$ both belong to $Y_j$, we have $d_{\mathcal{Y}}(y, y') \le \varepsilon$. Thus, using the fact that the loss function $\ell$ is $c_\ell$-Lipschitz continuous, we obtain

$$\begin{aligned}
\left| \ell\big(h_{\mathcal{S}}(G), y\big) - \ell\big(h_{\mathcal{S}}(G'), y'\big) \right| &\le c_\ell \max\big\{ d_{\mathcal{X}}\big(h_{\mathcal{S}}(G), h_{\mathcal{S}}(G')\big), d_{\mathcal{Y}}(y, y') \big\} \\
&\le c_\ell \max\{\varepsilon, \varepsilon\} = c_\ell \varepsilon,
\end{aligned}$$

as desired. $\qquad\square$

### J.1. Robustness and expressiveness

To apply Theorem 4 in our analysis, we must choose our pseudo-metric $d_{\mathcal{G}}$, the class of graph embeddings $\mathcal{H}$, and the loss function so that the Lipschitzness of the graph embeddings and the loss function guarantee robustness; see Corollary 6. Since we are primarily interested in the generalization abilities of MPNN architectures, the chosen pseudo-metric should be aligned with the MPNN outputs to satisfy a Lipschitz property. A minimal requirement is that whenever the distance between two graphs through the chosen pseudo-metric is zero, the MPNN outputs should be identical. We derive the following result by combining this observation with the results from Morris et al. (2019) regarding the equivalence of MPNNs and the 1-WL algorithm.

**Observation 2.** *Let $(\mathcal{G}, d_{\mathcal{G}})$ be a set of graph $\mathcal{G}$ paired with a pseudo-metric $d_{\mathcal{G}}$ and $\mathcal{H}$ be the hypothesis class of all possible MPNNs. Then, if $h$ is $C_h$-Lipschitz for all $h \in \mathcal{H}$, we have that the pseudo-metric $d_{\mathcal{G}}$ is* at least as expressive as *the 1-WL, i.e., for all $G_1, G_2 \in \mathcal{G}$ with $d_{\mathcal{G}}(G_1, G_2) = 0$, we have that $G_1, G_2$ are 1-WL indistinguishable.* $\qquad\square$

The observation shows some minimal expressivity requirements for the pseudo-metric must be met to satisfy the Lipschitz property on MPNNs. Thus, for the rest of this work, we focus on pseudo-metrics that are at least as expressive as the 1-WL or its variants such that the 1-MWL.

## K. Graphon analysis

In the following section, we define graphons, the extension of graphs to graphs with infinitely many (and uncountable) vertices. Formally, the space of graphons is the completion of the space of graphs to a compact space concerning the cut-distance; see Frieze & Kannan (1999) and Lovász (2012, Theorem 9.23). Next, we extend the tree distance to the space of graphons such that the graphon tree distance between two induced graphons is precisely the tree distance between the original graphs. We further introduce the Prokhorov pseudo-metric $\delta_P$ on the space of graphons as defined in (Böker et al., 2023). We proceed by describing a message-passing mechanism on the space of graphons such that applying this mechanism to a graphon induced by a graph is equivalent to using the MPNN defined in Equation (1) to the original graph. Finally, we present Theorems 29 and 31 from Böker et al. (2023), showing the equicontinuity of MPNNs on graphons concerning the Prokhorov pseudo-metric and the uniform continuity of the Prokhorov pseudo-metric concerning the graphon tree distance, respectively. Combining these two results, we establish the equicontinuity of MPNNs concerning the tree distance, as stated in Theorem 8 of the main paper.

**Graphons**    We begin with the definition of a graphon, which is just a symmetric measurable function $W \colon [0,1]^2 \to [0,1]$. We denote the set of all graphons by $\mathcal{W}$. Graphons generalize graphs in the following sense. Every $n$-order graph $G$ can be viewed as a graphon $W_G$ by partitioning $[0,1]$ into $n$ intervals $(I_v)_{v \in V(G)}$, each of mass $1/n$, and letting $W_G(x, y)$ for $x \in I_u, y \in I_v$ be one if $\{u, v\} \in E(G)$ and zero otherwise. We call $W_G$ the induced graphon by graph $G$.

**Graphon tree distance**    A graphon $W$ defines an operator $T_W \colon L^2([0,1]) \to L^2([0,1])$ on the space $L^2([0,1])$ of square-integrable functions modulo equality almost everywhere as

$$(T_W f)(x) := \int_{[0,1]} W(x, y) f(y) \, d\lambda(y), \quad x \in [0,1], \ f \in L^2([0,1]).$$

Following Böker (2021) we define the *graphon tree distance* of two graphons $U$ and $W$ as

$$\delta_\Box^{\mathcal{T}}(U, W) := \inf_S \sup_{f,g} |\langle f, (T_U \circ S - S \circ T_W)g\rangle|,$$

where the supremum is taken over all measurable functions $f, g : [0,1] \to [0,1]$ and the infimum is taken over all *Markov operators* $S$, i.e., operators $S : L^2([0,1]) \to L^2([0,1])$ such that $S(f) \geq 0$ for every $f \geq 0$, $S(1_{[0,1]}) = 1_{[0,1]}$, and $S^*(1_{[0,1]}) = 1_{[0,1]}$ also for the Hilbert adjoint $S^*$. Markov operators are the infinite-dimensional analog to doubly stochastic matrices (Eisner et al., 2015).

The following result by Böker (2021) shows that the graphon tree distance specializes to the tree distance defined on graphs when applied to the corresponding induced graphons.

**Lemma 39** ((Böker, 2021, Lemma 15)). *Let $G, H \in \mathcal{G}_n$ for some $n \in \mathbb{N}$. Then,*

$$\delta_\Box^{\mathcal{T}}(G, H) = n^2 \cdot \delta_\Box^{\mathcal{T}}(W_G, W_H). \qquad \Box$$

**Prokhorov pseudo-metric on graphons** Without delving into technical details, we denote the *Prokhorov* pseudo-metric on $\mathcal{W}$ as $\delta_P$. The formal definition can be found in Böker et al. (2023). Intuitively, the Prokhorov pseudo-metric is a pseudo-metric on the space of graphons and is based on the Prokhorov distance between the color histograms produced by a continuous counterpart of the 1-WL algorithm applied to graphons.

**Message passing graph neural networks on graphons** For a graphon $W \in \mathcal{W}$, an $L$-layer MPNN initializes a feature $\boldsymbol{h}_x^{(0)} := \varphi_0 \in \mathbb{R}^{d_0}$ for $x \in [0,1]$. Then, for $t \in [L]$, we compute $\boldsymbol{h}_x^{(t)} : [0,1] \to \mathbb{R}^{d_t}$ and the single graphon-level feature $\boldsymbol{h}_W \in \mathbb{R}^{1 \times d}$ after $L$ layers by

$$\boldsymbol{h}_x^{(t)} := \varphi_t \left( \int_{[0,1]} W(x,y) \boldsymbol{h}_y^{(t-1)} \, d\lambda(y) \right), \quad \text{and} \quad \boldsymbol{h}_W := \psi \left( \int_{[0,1]} \boldsymbol{h}_x^{(L)} \, d\lambda(x) \right). \tag{14}$$

Where, $(\varphi)_{t=1}^L, \psi$ are the functions of Equation (1). This definition generalizes Equation (1) on graphons. More precisely, we have the following result.

**Lemma 40** ((Böker et al., 2023, Theorem 9)). *Let $G \in \mathcal{G}_n$, for some $n \in \mathbb{N}$, let $(\varphi_{t=1}^L)$ be an $L$-layer MPNN model, and $\psi$ be Lipschitz. Then,*

$$\boldsymbol{h}_G = \boldsymbol{h}_{W_G}. \qquad \Box$$

We are now ready to present Theorems 29 and 31 from (Böker et al., 2023), presenting the uniform continuity of MPNNs on graphons for the Prokhorov pseudo-metric and the uniform continuity of the Prokhorov pseudo-metric for the graphon tree distance, respectively.

**Theorem 41** ((Böker et al., 2023, Theorem 29)). *For all $L \in \mathbb{N}$, $\varepsilon > 0$, there exists a $\delta_1 > 0$ such that, for all graphons $U$ and $W$, if $\delta_P(U, W) \leq \delta_1$, then $\|\boldsymbol{h}_U - \boldsymbol{h}_W\|_2 \leq \varepsilon$ for every $L'$-layer MPNN model given by Equation (14) with $L \leq L'$.* $\Box$

**Theorem 42** ((Böker et al., 2023, Theorem 31)). *For every $\varepsilon > 0$, there exists a $\delta_2 > 0$ such that, for all graphons $U$ and $W$, if $\delta_\Box^{\mathcal{T}}(U, W) \leq \delta_2$, then $\delta_P(U, W) \leq \varepsilon$.* $\Box$

We now combine the above two results to show the equicontinuity of $\mathsf{MPNN}_{L,M',L_{\mathsf{FNN}}}^{\mathrm{ord}}(\mathcal{G}_n)$, as stated in Theorem 8 in the main paper.

**Theorem 43** (Theorem 8 in the main paper). *For all $\varepsilon > 0$ and $L \in \mathbb{N}$, there exists $\gamma(\varepsilon) > 0$ such that for $n \in \mathbb{N}$, $M' \in \mathbb{R}$, $G, H \in \mathcal{G}_n$, and all MPNN architectures in $\mathsf{MPNN}_{L,M',L_{\mathsf{FNN}}}^{\mathrm{ord}}(\mathcal{G}_n)$, the following implication holds,*

$$\delta_\Box^{\mathcal{T}}(G, H) < n^2 \cdot \gamma(\varepsilon) \implies \|\boldsymbol{h}_G - \boldsymbol{h}_H\|_2 \leq \varepsilon.$$

*Proof.* Given $\varepsilon > 0$, let $\delta_1(\varepsilon)$ be as described in Theorem 41. Set this $\delta_1(\varepsilon)$ as the $\varepsilon$ in Theorem 42 and compute $\delta_2(\delta_1(\varepsilon)) = \delta_2(\varepsilon)$. Now, by setting $\gamma(\varepsilon) = \delta_2(\varepsilon)$, we have that for all $n \in \mathbb{N}$ and $G, H \in \mathcal{G}_n$, if $\delta_\Box^{\mathcal{T}}(W_G, W_H) \leq \gamma(\varepsilon)$, or equivalently, by Lemma 39 if,

$$\delta_\Box^{\mathcal{T}}(G, H) \leq n^2 \cdot \gamma(\varepsilon),$$

then $\delta_P(W_G, W_H) \leq \delta_1(\varepsilon)$. Hence, by Theorem 41, $\|\boldsymbol{h}_{W_G} - \boldsymbol{h}_{W_H}\|_2 \leq \varepsilon$, for all $L$-layer MPNN models given by Equation (14). Finally, by Lemma 40, the last inequality can equivalently be written as

$$\|\boldsymbol{h}_G - \boldsymbol{h}_H\|_2 \leq \varepsilon,$$

completing the proof. $\qquad\square$

## L. Robustness and VC dimension bounds

Here, we show how we can recover previously obtained bounds on the generalization abilities of MPNNs of Morris et al. (2023a) through the generalization framework of Section 3. That is, we show how the robustness framework of Section 3 allows us to recover the results in Morris et al. (2023a, Proposition 1 and 2) by fixing a particular pseudo-metric on the set of $n$-order graphs. Unlike Morris et al. (2023a), our results not only hold for the binary classification setting using the 0-1 loss but, under mild assumptions, for arbitrary loss functions, such as the cross entropy loss and also standard loss functions for the regression case.

**Analysis of generalization abilities of MPNNs of Morris et al. (2023a)** To state the results of Morris et al. (2023a), we first introduce the VC dimension of MPNNs. That is, for a class of MPNNs MPNN$(\mathcal{X})$ operating on a set of graphs $\mathcal{X}$, e.g., MPNN$_L(\mathcal{X})$, the *VC dimension* VC-dim(MPNN$(\mathcal{G})$) is the maximal number $m$ of graphs $G_1, \ldots, G_m$ in $\mathcal{X}$ that can be shattered by MPNN$(\mathcal{X})$. Here, $G_1, \ldots, G_m$ are *shattered* if for any $\boldsymbol{\tau}$ in $\{0,1\}^m$ there exists a MPNN mpnn $\in$ MPNN$(\mathcal{X})$ such that for all $i$ in $[m]$,

$$\mathsf{mpnn}(G_i) = \tau_i.$$

By standard learning-theoretic results, bounding the VC dimension of a class of functions directly implies a bound on the generalization error.[9]

**Theorem 44** (Vapnik & Chervonenkis (1964); Vapnik (1998, adapted to MPNNs)). *Let $\mathcal{X}$ be a set of graphs and let* MPNN$(\mathcal{X})$ *be a class of MPNNs operating on $\mathcal{X}$, with* VC-dim(MPNN$(\mathcal{X})$) $= d < \infty$. *Then, for all $\delta \in (0,1)$, with probability at least $1 - \delta$, the following holds for all* mpnn $\in$ MPNN$(\mathcal{X})$,

$$\ell_{\exp}(\mathsf{mpnn}) \leq \ell_{\mathrm{emp}}(\mathsf{mpnn}) + \sqrt{\frac{2d \log (eN/d)}{N}} + \sqrt{\frac{\log (1/\delta)}{2N}}. \qquad\square$$

To bound the VC dimension of MPNNs, Morris et al. (2023a) consider $\mathcal{X} = \mathcal{G}_{n,d}^{\mathbb{B}}$, the set of $n$-order graphs with $d$-dimensional Boolean vertex features and $\mathcal{Y} = \{0,1\}$. Like Morris et al. (2023a), for $d > 0$, by $m_{n,d,L}$ we denote the number of unique graphs in $\mathcal{G}_{n,d}^{\mathbb{B}}$ distinguishable by 1-WL after $L$ iterations. Now, Morris et al. (2023a) proved the following upper and lower bounds for the VC dimension of $L$-layer MPNNs on the set $\mathcal{G}_{n,d}^{\mathbb{B}}$ of graphs, showing that the VC dimension is bounded by the ability of the 1-WL to distinguish $n$-order graphs.

**Proposition 45** (Morris et al. (2023a, Proposition 1)). *For all $n$, $d$ and $L \in \mathbb{N}$, it holds that*

$$\mathsf{VC\text{-}dim}\left(\mathsf{MPNN}_L\left(\mathcal{G}_{n,d}^{\mathbb{B}}\right)\right) \leq m_{n,d,L}. \qquad\square$$

This upper bound holds regardless of the choice of aggregation, update, and readout functions used in the MPNN architecture. They also show a matching lower bound.

**Proposition 46** (Morris et al. (2023a, Proposition 2)). *For all $n$, $d$, and $L \in \mathbb{N}$, all $m_{n,d,L}$ 1-WL-distinguishable $n$-order graphs with $d$-dimensional boolean features can be shattered by sufficiently wide $L$-layer MPNNs. Hence,*

$$\mathsf{VC\text{-}dim}\left(\mathsf{MPNN}_L\left(\mathcal{G}_{n,d}^{\mathbb{B}}\right)\right) = m_{n,d,L}. \qquad\square$$

Hence, combining the above result with Theorem 44 implies that for all samples $\mathcal{S}$, with probability at least $1 - \delta$,

$$\ell_{\exp}(h_{\mathcal{S}}) \leq \ell_{\mathrm{emp}}(h_{\mathcal{S}}) + \sqrt{\frac{2m_{n,d,L} \log (e|\mathcal{S}|/m_{n,d,L})}{|\mathcal{S}|}} + \sqrt{\frac{\log (1/\delta)}{2|\mathcal{S}|}}.$$

---

[9]For ease of exposition, compared to Morris et al. (2023a), we use a slightly different definition of MPNNs' VC dimension. However, the proofs of Propositions 1 and 2 also work in the case of the above definition.

Note that the logarithm must be positive for the above bound to be meaningful. This requires the assumption that the sample size is significantly larger than the total number of 1-WL-distinguishable graphs, which is generally not feasible, especially for large graphs. We now largely recover the above generalization bound without any assumptions on the sample size through the theory outlined in Section 3 and extend it to the case of regression.

**Recovering Morris et al. (2023a) via the robustness framework**    Let us consider the binary classification setting with $\mathcal{Y} = \{0,1\}$ and the 0-1 loss function $\ell \colon \mathcal{Y} \times \mathcal{Y} \to \{0,1\}$, where $\ell(y_1, y_2) = 0$ if, and only if, $y_1 = y_2$. Given that the expressive power of any MPNN with $L$ layers is bounded by 1-WL after $L$ iterations (Morris et al., 2019), it is straightforward to verify that any graph learning algorithm for $\mathsf{MPNN}_L(\mathcal{G}_{n,d}^{\mathbb{B}})$ on $\mathcal{Z} \coloneqq \mathcal{G}_{n,d}^{\mathbb{B}} \times \mathcal{Y}$ is $(2m_{n,d,L}, 0)$-robust, leading to the following generalization bound. For $\delta \in (0,1)$ and for all samples $\mathcal{S}$, with probability at least $1 - \delta$, we have

$$|\ell_{\exp}(h_{\mathcal{S}}) - \ell_{\mathrm{emp}}(h_{\mathcal{S}})| \leq \sqrt{\frac{4m_{n,d,L} \log(2) + 2\log(1/\delta)}{|\mathcal{S}|}}.$$

To see this, observe that we can construct a partition of $\mathcal{Z}$ using the sets $C = G \times \{j\}$, for $G \in \mathcal{G}_n/_{\sim_{\mathsf{WL}_L}}$, $j \in \{0,1\}$.

**Regression setting**    In contrast to Morris et al. (2023a), the robustness framework is flexible enough to accommodate different loss functions straightforwardly. For example, consider the set of graphs $\mathcal{G}_{n,d}^{\mathbb{B}}$, we now extend to the regression setting where $\mathcal{Y} = (0,1)$, with the loss function $\ell \colon \mathcal{Y} \times \mathcal{Y} \to (0,1)$, defined by $\ell(y_1, y_2) = |y_1 - y_2|$. We then define the pseudo-metric $\mathsf{WL}_{1,L}$ on $\mathcal{G}_{n,d}^{\mathbb{B}}$ as follows,

$$\mathsf{WL}_{1,L}(G,H) \coloneqq \begin{cases} 1, & \text{if 1-WL distinguishes } G \text{ and } H \text{ after } L \text{ iterations,} \\ 0, & \text{otherwise.} \end{cases}$$

We further consider the metric $|\cdot|$ on $(0,1)$ as the absolute difference metric. Observing that: (1) the covering number $\mathcal{N}((0,1), |\cdot|, \varepsilon) < 2/\varepsilon$, for $\varepsilon < 1$, (2) the covering number $\mathcal{N}(\mathcal{G}_{n,d,L}^{\mathbb{B}}, \mathsf{WL}_{1,L}, \varepsilon) = m_{n,d,L}$, for $\varepsilon < 1$, (3) the loss function $\ell(x,y) = |x-y|$ is 2-Lipschitz, and (4) $L$-layer MPNNs are 1-Lipschitz with respect to the $\mathsf{WL}_{1,L}$ pseudo-metric, we can apply Corollary 6 to obtain the following bound. For $\delta \in (0,1)$, for all samples $\mathcal{S}$, with probability at least $1 - \delta$ and for $\epsilon < 1$, we have

$$|\ell_{\exp}(h_{\mathcal{S}}) - \ell_{\mathrm{emp}}(h_{\mathcal{S}})| \leq 4\epsilon + \sqrt{\frac{2/\epsilon \cdot m_{n,d,L} \log(2) + 2\log(1/\delta)}{|\mathcal{S}|}}.$$

# M. Fine-grained analysis of MPNNs with other aggregation functions

In this section, we extend the previous results analyzing MPNN layers using mean aggregation according to Equation (5). Let $\bar{m}_{n,d,q,L}$ denote the number of equivalence classes induced by the 1-MWL after $L$ iterations on $\mathcal{G}_{n,d,q}$. For the case when $L = n - 1$, we simply write $\bar{m}_{n,d,q}$. To establish generalization bounds, we rely on the mean-Forest distance $\mathrm{FD}_L^m$ defined in Section 2. Similar to the Forest distance, we can bound the mean-Forest distance between two graphs that differ by either a single edge or vertex label change, as shown in the following lemma, the proof proceeds identically to the proof of Lemma 13.

**Lemma 47.** *For $d, q, L \in \mathbb{N}$, there exists a constant $b^{(m)}(d,q,L)$ such that for $n \in \mathbb{N}$ and $G, H \in \mathcal{G}_{n,d,q}$, if $G$ can be obtained from $H$ by either deleting an edge or by replacing a vertex feature with a new one, then*

$$\mathrm{FD}_L^{(m)}(G,H) \leq b^{(m)}(d,q,L). \qquad \square$$

We next analyze the generalization power of the MPNN class $\mathsf{MPNN}_{L,M',L_{\mathsf{FNN}}}^{\mathrm{mean}}(\mathcal{G}_{n,w,q})$. Again, we consider the loss function $\ell \colon \mathbb{R} \times \{0,1\} \to \mathbb{R}$, defined by $\ell(x,y) = y \log(\sigma(x)) + (1-y)\log(1-\sigma(x))$, where $\sigma(x)$ is the sigmoid function. The following result shows the Lipschitz continuity property of MPNNs in $\mathsf{MPNN}_{L,M',L_{\mathsf{FNN}}}^{\mathrm{mean}}(\mathcal{G}_{n,w,q})$ concerning the mean-Forest distance. Similarly to sum aggregation MPNNs, we state and prove the Lipschitz property below for the more general setting with graphs in $\mathcal{G}_{n,d}^{\mathbb{R}}$.

**Lemma 48.** *For every $L, n, d \in \mathbb{N}, M' \in \mathbb{R}$ and for all MPNNs in $\mathsf{MPNN}_{L,M',L_{\mathsf{FNN}}}^{\mathrm{mean}}(\mathcal{G}_{n,d}^{\mathbb{R}})$, we have*

$$\|\boldsymbol{h}_G - \boldsymbol{h}_H\|_2 \leq \frac{1}{n} C^{(m)}(L) L_\psi \prod_{i=1}^{L} L_{\varphi_i} \mathrm{FD}_L^{\mathrm{m}}(G,H), \quad \text{for all } G, H \in \mathcal{G}_{n,d}^{\mathbb{R}},$$

*where $C^{(m)}(L)$ is a constant depending on L.* □

We now state the generalization bound for mean aggregation MPNNs.

**Proposition 49.** *For $\varepsilon > 0$ and $n, q, d, L \in \mathbb{N}$, $M' \in \mathbb{R}$ and any graph learning algorithm for $\mathrm{MPNN}^{\mathrm{mean}}_{L,M',L_{\mathsf{FNN}}}(\mathcal{G}_{n,d,q})$ is*

$$\left(2\mathcal{N}(\mathcal{G}_{n,d,q}, \mathrm{FD}^{(m)}_L, \varepsilon), L_\ell \cdot L_{\mathsf{FNN}} \cdot C_{\mathrm{FD}^{(m)}_L} \cdot 2\varepsilon\right)\text{-robust.}$$

*Hence, for any sample $\mathcal{S}$ and $\delta \in (0,1)$, with probability at least $1 - \delta$,*

$$|\ell_{\exp}(h_\mathcal{S}) - \ell_{\mathrm{emp}}(h_\mathcal{S})| \leq \widetilde{C}^{(m)}\varepsilon + M\sqrt{\frac{2\mathcal{N}(\mathcal{G}_{n,w,q}, \mathrm{FD}^{(m)}_L, \varepsilon)2\log 2 + 2\log(1/\delta)}{|\mathcal{S}|}}, \text{ for } \varepsilon > 0,$$

*where $\widetilde{C}^{(m)} = 2/n L_\ell L_{\mathsf{FNN}} C_{\mathrm{FD}^{(m)}}$, $C_{\mathrm{FD}^{(m)}_L} = C^{(m)}(L)L_\psi \prod_{i=1}^{L} L_{mlp_i}$ the Lipschitz constant by Lemma 48, and $M$, is an upper bound of the loss function $\ell$.* □

We similarly bound the covering number concerning the mean-Forest distance, deriving the following bound.

**Proposition 50.** *For $n, q, d, L \in \mathbb{N}$, and $M' \in \mathbb{R}$, for any graph learning algorithm for the class $\mathrm{MPNN}^{\mathrm{sum}}_{L,M',L_{\mathsf{FNN}}}(\mathcal{G}_{n,d,q})$ and for any sample $\mathcal{S}$ and $\delta \in (0,1)$, with probability at least $1 - \delta$, we have*

$$|\ell_{\exp}(h_\mathcal{S}) - \ell_{\mathrm{emp}}(h_\mathcal{S})| \leq 2\widetilde{C}^{(m)}b^{(m)}(d,q,L)k + M\sqrt{\frac{\overline{m_{n,d,q,L}}{k+1}4\log(2) + 2\log(1/\delta)}{|\mathcal{S}|}}, \quad \text{for } k \in \mathbb{N},$$

# N. Covering number dependency on graphs'-order

In previous sections, we showed that for different choices of $\varepsilon$, the covering number can be bounded by a function that always increases scales on $n$ as $m_n$, i.e., the number of 1-WL-indistinguishable classes. Specifically, when estimating the limit

$$\lim_{n \to \infty} \frac{\mathcal{N}(\mathcal{G}_n, \delta^{\mathcal{T}}_\square, \varepsilon)}{m_n},$$

the result will always be a constant that depends on $\varepsilon$. While one could use Equation (10) with radius $4n$ to obtain a bound that scales as $m_n/(n+1)$ in $n$, such an increase in radius would impact the first term in the generalization bound. Hence, the question arises of whether it is possible to derive a bound that grows more slowly with $n$ than $m_n$ for some sufficiently interesting graph class, using either a constant radius or a radius that depends only mildly on $n$.

More formally, a natural question is whether we can establish tighter bounds for the covering number for a given constant $\varepsilon$ that does not depend, or mildly depends, on $n$. For instance, is there a choice of $\varepsilon > 0$ and a graph class $\mathcal{F}_n$ such that

$$\mathcal{N}(\mathcal{F}_n, \delta^{\mathcal{T}}_\square, \varepsilon) \leq \frac{|\mathcal{F}_n/\sim_{\mathrm{WL}}|}{b_n}, \tag{15}$$

where $b_n \to \infty$ as $n$ grows? If such a graph class exists, then MPNNs could effectively handle the growth in graph size without significantly impacting their generalization performance. Below, we present a class of graphs $\mathcal{F}_n$ that satisfies Equation (15) for radius $\varepsilon = 4$.

We define the class $\mathcal{F}_n$ as follows, let $P_n$ denote a path on $n$ vertices, with vertex set $V(P_n) = \{v_1, \ldots, v_n\}$ and edge set $E(p_n) = \{\{v_1, v_2\}, \ldots, \{v_{n-1}, v_n\}\}$. We denote by $\mathcal{P}(n)$ the set consisting of disjoint unions of paths on $n$ vertices. Then, we define the graph class

$$\mathcal{F}_n := \{(V, E) \mid V := V(P_n) \dot\cup V(P), E := E(P) \dot\cup E(P_n)\dot\cup\{u, v\}, \text{ for } P \in \mathcal{P}(n),$$
$$u \in V(P), \text{ and } v \in V(P_n)\}.$$

We will now show that for the graph class $\mathcal{F}_n$ and $\varepsilon = 4$, Equation (15) holds.

**Proposition 51.** *For the the graph class $\mathcal{F}_n \subset \mathcal{G}_n$ constructed above, we have*

$$\mathcal{N}(\mathcal{F}_n, \delta^{\mathcal{T}}_\square, 4) \leq \frac{2|\mathcal{F}_n/\sim_{WL}|}{n}.$$
□

Note that in the above proposition, we established that the covering number is bounded by $|\mathcal{P}(n)|$, which has an interesting combinatorial interpretation. Specifically, $|\mathcal{P}(n)|$ represents the integer partition of $n$, i.e., the number of distinct ways to express $n$ as a sum of positive integers. While there is no known closed formula for the integer partition, it can be accurately approximated through asymptotic expansions and computed precisely via recurrence relations, making the bound efficiently computable (Andrews & Eriksson, 2004).

**Regression setting** Here, we lift the results from the previous section to the regression setting. Specifically, we use consider the loss function $\ell(x, y) = |x - y|$, resulting in the following results.

**Proposition 52.** *For every $L, n \in \mathbb{N}, M' \in \mathbb{R}$ and for all MPNNs in* $\mathsf{MPNN}^{\mathrm{ord}}_{L,M',L_{\mathsf{FNN}}}(\mathcal{G}_n)$, *we have the following generalization bound. For any sample $\mathcal{S}$, and any $\delta \in (0,1)$, with probability at least $1 - \delta$, for $\varepsilon < 2M'$,*

$$|\ell_{\exp}(h_{\mathcal{S}}) - \ell_{\mathrm{emp}}(h_{\mathcal{S}})| \le 2 \cdot L_{\mathsf{FNN}} \cdot \bar{\gamma}^{\leftarrow}\left(\frac{2\varepsilon}{n^2}\right) + M'\sqrt{\frac{\mathcal{N}(\mathcal{G}_n, \delta^{\mathcal{T}}_{\square}, \varepsilon)\frac{4M'\log(2)}{\varepsilon} + 2\log(1/\delta)}{|\mathcal{S}|}}. \qquad \square$$

# O. Missing proofs from section Section 4

The following result links robustness with the generalization error of order-normalized MPNNs.

**Proposition 53** (Proposition 9 in the main paper). *For $\varepsilon > \frac{1}{2} \cdot \inf\limits_{\delta^{\mathcal{T}}_{\square}(G,H)>0} \delta^{\mathcal{T}}_{\square}(G, H)$, $n, L \in \mathbb{N}$, and $M' \in \mathbb{R}$, any graph learning algorithm for the class* $\mathsf{MPNN}^{\mathrm{ord}}_{L,M',L_{\mathsf{FNN}}}(\mathcal{G}_n)$ *is*

$$\left(2\mathcal{N}(\mathcal{G}_n, \delta^{\mathcal{T}}_{\square}, \varepsilon), L_{\ell} \cdot L_{\mathsf{FNN}} \cdot \bar{\gamma}^{\leftarrow}\left(\frac{2\varepsilon}{n^2}\right)\right)\text{-robust.}$$

*Hence, for any sample $\mathcal{S}$ and $\delta \in (0,1)$, with probability at least $1 - \delta$,*

$$|\ell_{\exp}(h_{\mathcal{S}}) - \ell_{\mathrm{emp}}(h_{\mathcal{S}})| \le L_{\ell} \cdot L_{\mathsf{FNN}} \cdot \bar{\gamma}^{\leftarrow}\left(\frac{2\varepsilon}{n^2}\right) + M\sqrt{\frac{4\mathcal{N}(\mathcal{G}_n, \delta^{\mathcal{T}}_{\square}, \varepsilon)\log(2) + 2\log\left(\frac{1}{\delta}\right)}{|\mathcal{S}|}},$$

*where, $M$ is an upper bound for the loss function $\ell$ and $L_{\ell}$, the Lipschitz constant of $\ell(\cdot, y), y \in \{0,1\}$.*

*Proof.* We first note that since $\mathsf{FNN}$ is bounded by $M'$, we know that the loss function is also bounded from some $M \in \mathbb{R}$. Now, let $\mathcal{Z} = \mathcal{G}_n \times \{0,1\}$. For a given $\varepsilon > 0$, graphs $G_1, G_2 \in \mathcal{G}_n$, and $y_1, y_2 \in \{0,1\}$, if $\max\{\delta^{\mathcal{T}}_{\square}(G_1, G_2), \delta(y_1, y_2)\} < \bar{\gamma}(\varepsilon)n^2$, where $\delta$ is the Kronecker-delta metric on $\{0,1\}$, i.e.,

$$\delta(y_1, y_2) = \begin{cases} \infty, & \text{if } y_1 \ne y_2, \text{ and} \\ 0 & \text{if } y_1 = y_2, \end{cases}$$

we have that $\delta(y_1, y_2) < \bar{\gamma}(\varepsilon)n^2$, implies $y_1 = y_2$, and therefore, for any sample $\mathcal{S}$,

$$\begin{aligned} |\ell(h_{\mathcal{S}}(G_1), y_1) - \ell(h_{\mathcal{S}}(G_2), y_2)| &= |\ell(h_{\mathcal{S}}(G_1), y_1) - \ell(h_{\mathcal{S}}(G_2), y_1)| \\ &\le L_{\ell}|h_{\mathcal{S}}(G_1) - h_{\mathcal{S}}(G_2)| \\ &\le L_{\ell} \cdot L_{\mathsf{FNN}} \cdot \varepsilon. \end{aligned}$$

Therefore, by Proposition 7, any learning algorithm for $\mathsf{MPNN}^{\mathrm{ord}}_{L,M',L_{\mathsf{FNN}}}(\mathcal{G}_n)$ is

$$\left(2\mathcal{N}(\mathcal{G}_n, \delta^{\mathcal{T}}_{\square}, \frac{\bar{\gamma}(\varepsilon)n^2}{2}), L_{\ell} \cdot L_{\mathsf{FNN}} \cdot \varepsilon\right)\text{-robust.}$$

Now, we replace $\varepsilon$ with $\bar{\gamma}^{\leftarrow}\left(\frac{2\varepsilon}{n^2}\right)$. By definition of $\bar{\gamma}^{\leftarrow}$, if $\bar{\gamma}^{\leftarrow}\left(\frac{2\varepsilon}{n^2}\right) \ne 0$, we have $\bar{\gamma}\left(\bar{\gamma}^{\leftarrow}\left(\frac{2\varepsilon}{n^2}\right)\right) \ge \frac{2\varepsilon}{n^2}$. Thus, $\mathcal{N}(\mathcal{G}_n, \delta^{\mathcal{T}}_{\square}, \frac{\bar{\gamma}(\bar{\gamma}^{\leftarrow}(\frac{2\varepsilon}{n^2}))n^2}{2}) \le \mathcal{N}(\mathcal{G}_n, \delta^{\mathcal{T}}_{\square}, \varepsilon)$, leading to the following generalization bound:

$$|\ell_{\exp}(h_{\mathcal{S}}) - \ell_{\mathrm{emp}}(h_{\mathcal{S}})| \le L_{\ell} \cdot L_{\mathsf{FNN}} \cdot \bar{\gamma}^{\leftarrow}\left(\frac{2\varepsilon}{n^2}\right) + M\sqrt{\frac{4\mathcal{N}(\mathcal{G}_n, \delta^{\mathcal{T}}_{\square}, \varepsilon)\log(2) + 2\log\left(\frac{1}{\delta}\right)}{|\mathcal{S}|}}.$$

It remains to check the case where $\overline{\gamma}^{\leftarrow}\left(\frac{2\varepsilon}{n^2}\right) = 0$.

If $\overline{\gamma}^{\leftarrow}\left(\frac{2\varepsilon}{n^2}\right) = 0$, then $\overline{\gamma}(x) \geq \frac{2\varepsilon}{n^2}$ for all $x > 0$. Thus, for all graphs $G_1, G_2$, with $\delta_{\square}^{\mathcal{T}}(G_1, G_2) < \frac{2\varepsilon}{n^2} \cdot n^2$, we have:

$$\|h_{G_1} - h_{G_2}\|_2 \leq x, \quad \text{for all } x > 0, h \in \mathsf{MPNN}_{L,M',L_{\mathsf{FNN}}}^{\mathrm{ord}}(\mathcal{G}_n) \Leftrightarrow$$

$$\|h_{G_1} - h_{G_2}\|_2 = 0, \quad \text{for all } h \in \mathsf{MPNN}_{L,M',L_{\mathsf{FNN}}}^{\mathrm{ord}}(\mathcal{G}_n),$$

which, following Böker et al. (2023), implies that $\delta_{\square}^{\mathcal{T}}(G_1, G_2) = 0$.

Now, observe that since for all $G_1, G_2 \in \mathcal{G}_n$ with $\delta_{\square}^{\mathcal{T}}(G_1, G_2) < 2\varepsilon$, we have $\delta_{\square}^{\mathcal{T}}(G_1, G_2) = 0$, implying

$$\varepsilon \leq \frac{1}{2} \inf_{\substack{G,H \in \mathcal{G}_n \\ \delta_{\square}^{\mathcal{T}}(G,H) > 0}} \delta_{\square}^{\mathcal{T}}(G, H),$$

which leads to a contradiction, since we have assumed $\varepsilon > \frac{1}{2} \inf_{\substack{G,H \in \mathcal{G}_n \\ \delta_{\square}^{\mathcal{T}}(G,H) > 0}} \delta_{\square}^{\mathcal{T}}(G, H)$. $\qquad\square$

The following results provide a generalization bound using the number of graphs distinguished by 1-$\mathsf{WL}$ to derive an upper bound on the covering number.

**Theorem 54** (Theorem 10 in the main paper)**.** *For* $n, L \in \mathbb{N}, M' \in \mathbb{R}$*, for any graph learning algorithm for* $\mathsf{MPNN}_{L,M',L_{\mathsf{FNN}}}^{\mathrm{ord}}(\mathcal{G}_n)$ *and for any sample* $\mathcal{S}$ *and* $\delta \in (0,1)$*, with probability at least* $1 - \delta$*, we have*

$$|\ell_{\exp}(h_{\mathcal{S}}) - \ell_{\mathrm{emp}}(h_{\mathcal{S}})| \leq L_\ell \cdot L_{\mathsf{FNN}} \cdot \overline{\gamma}^{\leftarrow}\left(\frac{8k}{n^2}\right) + M\sqrt{\frac{4\frac{m_n}{k+1}\log(2) + 2\log(1/\delta)}{|\mathcal{S}|}},$$

*for* $k \in \mathbb{N}$*, where* $M$ *is an upper bound on the loss function* $\ell$*.*

*Proof.* First, note that $\delta_{\square}^{\mathcal{T}}(G, H) = 0$ if, and only, if the graphs $G, H \in \mathcal{G}_n$ are 1-$\mathsf{WL}$-equivalent. By the triangle inequality, we also have that if $G, H \in \mathcal{G}_n$, and $G' \in [G], H' \in [H]$ are two arbitrarily chosen graphs from their respective equivalence classes, then

$$\delta_{\square}^{\mathcal{T}}(G, H) = \delta_{\square}^{\mathcal{T}}(G', H').$$

Thus, we can define the Tree distance between two equivalence classes as the Tree distance between two randomly chosen representatives. Moreover, computing the covering number on $\mathcal{G}_n$ is equivalent to computing the covering number on the pseudo-metric space $(\mathcal{G}_n/_{\sim_{\mathsf{WL}}}, \delta_{\square}^{\mathcal{T}})$.

We now construct a labeled directed tree with labels from $\mathcal{G}_n/_{\sim_{\mathsf{WL}}}$ level-wise, starting from the root $r$ (level 0), as follows.

1. The root corresponds to the complete graph on $n$ vertices denoted as $K_n$, which is unique in its equivalence class.

2. The first level consists of a single vertex corresponding to the graph derived by deleting an edge from the complete graph on $n$ vertices. Denote this graph as $K_n^-$, which is also unique in its equivalence class.

3. In the next level, we compute all graphs obtained by deleting an edge from $K_n^-$. Denote these graphs as $G_1, \ldots, G_r$, for $r \in \mathbb{N}$. We then de-duplicate the set $\{G_1, \ldots, G_r\}$ for the 1-$\mathsf{WL}$, keeping one representative for each equivalence class. This reduces the set to $[G_1], \ldots, [G_{r'}]$, where $r' \leq r$. These equivalence classes form the children of the vertex at level 1.

4. For each vertex corresponding to a graph class $[G_i]$ in level 2, we compute its children as follows. We first compute all graphs in the set.

$$[G_i]_c = \{G' \mid \exists\, G \in [G_i], e \in E(G) \text{ such that } G \setminus \{e\} \simeq G'\}.$$

   We connect these graphs to the vertex $[G_i]$. We then prune the tree by arbitrarily dropping all 1-$\mathsf{WL}$-equivalent graphs, keeping only one as the representative.

5. We continue this process until we reach the vertex corresponding to the empty graph.

Observe that in the resulting tree, the total number of vertices is exactly $m_n$, since for each $l \in \left[\frac{n(n-1)}{2}\right]$, level $l$ contains all 1-WL-equivalence classes for graphs with $n$ vertices and $\frac{n(n-1)}{2} - l$ edges. Additionally, the tree satisfies the following two properties:

1. Each parent is at most at a distance 2 from its children.

2. All siblings are at most at a distance of 4 from each other.

To verify this, consider the following. If $[G_1]$ is a child of $[G]$, then there exists $G_1 \in [G_1]$, $G \in [G]$, and an edge $e \in E(G)$ such that $G \setminus \{e\} \simeq G_1$. By the triangle inequality and the fact that

$$\delta_\Box^{\mathcal{T}}(G, G') = 0,$$

since $G$ and $G'$ are 1-WL-indistinguishable, we have

$$\delta_\Box^{\mathcal{T}}([G], [G_1]) = \delta_\Box^{\mathcal{T}}(G, G_1) = 2.$$

Similarly, if $[G_1]$ and $[G_2]$ are siblings, the triangle inequality gives

$$\delta_\Box^{\mathcal{T}}([G_1], [G_2]) \leq \delta_\Box^{\mathcal{T}}([G], [G_1]) + \delta_\Box^{\mathcal{T}}([G], [G_2]) \leq 4.$$

This follows from the fact that if the adjacency matrices of two graphs differ in exactly two entries, then their Tree distance is bounded above by 2. Now, we construct a cover of $\mathcal{G}_n/\sim_{\mathsf{WL}}$ as follows.

1. We start from the leaves of the tree and merge each leaf with its parent. We repeat this step $k$ times. The resulting graph is still a tree.

2. In this induced tree, we create as many disjoint sets of 1-WL-equivalence classes as there are leaves in the induced tree by considering all classes corresponding to a leaf as a set. Each group contains at least $k + 1$ elements.

3. We then remove all leaves, creating a new tree $T'$. For each leaf in $T'$, we repeat the merging process as many times as necessary until each leaf contains at least $k + 1$ equivalence classes (requiring at most $k$ merges).

At the end of this process, each group will have at least $k + 1$ elements, and the Tree distance between any two elements in the same group will be at most $4k$. To see this, note that for any two graph classes $[G]$ and $[H]$ in the same partition set, if we consider the tree to be undirected, there exists a path of length at most $2k$ in the initial tree connecting $[G]$ and $[H]$. Thus, we can find graphs $G \in [G]$ and $H \in [H]$, differing by at most $2k$ edge additions or deletions, implying that $G$ and $H$ are at Tree distance at most $4k$. This implies that the covering number

$$N(\mathcal{G}_n, \delta_\Box^{\mathcal{T}}, 4k) \leq \frac{m_n}{k+1}. \qquad \Box$$

The following result derives tighter bounds for Otter trees.

**Proposition 55** (Proposition 12 in the main paper). *For $L \in \mathbb{N}$, $M' \in \mathbb{R}$ and sufficiently large $n \in \mathbb{N}$, for any graph learning algorithm on $\mathsf{MPNN}^{\mathrm{ord}}_{L,M',L_{\mathsf{FNN}}}(\mathcal{T}^{(2)}_{2n+1})$ and for any sample $\mathcal{S}$, with $\delta \in (0,1)$, with probability at least $1 - \delta$, we have*

$$|\ell_{\exp}(h_\mathcal{S}) - \ell_{\mathrm{emp}}(h_\mathcal{S})| \leq L_\ell \cdot L_{\mathsf{FNN}} \cdot \overline{\gamma}^{\leftarrow}\left(\frac{16k}{(2n+1)^2}\right) + M\sqrt{\frac{4^{w_n}/b^{2k}\log(2) + 2\log(1/\delta)}{|\mathcal{S}|}},$$

*where $k \in \mathbb{N}$, $M$ is an upper bound on the loss function $\ell$, $b \approx 2\,4832$, and $w_n = |\mathcal{T}^{(2)}_{2n+1}|$.*

*Proof.* As in the proof of Theorem 10, we begin by constructing a rooted, labeled, directed tree. However, the labels are now graphs from $\bigcup_{j=1}^{n} \mathcal{T}^{(2)}_{2j+1}$, and we do not need to use 1-WL equivalence classes, as 1-WL can distinguish any pair of non-isomorphic trees.

1. The root (level 0) corresponds to the isolated vertex graph.

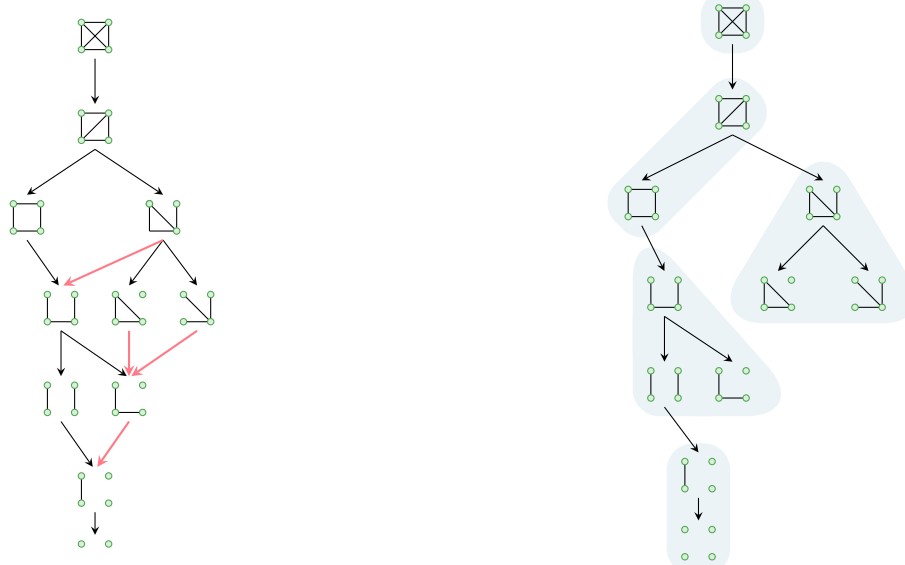

(a) An illustration of tree construction as described in the proof of Theorem 10. On the left, we show all possible trees that can be computed according to the description in the proof, represented in a directed acyclic graph (DAG) format. On the right, we display an arbitrarily chosen tree from this set.

(b) An illustration of the grouping process applied to the chosen tree, which leads to the desired upper bound for the covering number. All vertices belong to a group with at least two elements (possibly excluding from the root).

*Figure 5.* Illustrations related to tree construction and the grouping process.

2. The first level consists of a single vertex corresponding to the 3-path graph.

3. At level $l + 1$, we compute the children of vertices at level $l$ as the graphs obtained by connecting a leaf vertex of the graphs in level $l$ with two additional vertices. We then prune the tree by removing isomorphic graphs.

4. We continue this process until all binary trees at the current level have $2n + 1$ vertices.

The above construction shows that level $l$ consists of all Otter trees with $2l + 1$ vertices. Additionally, each child is at most at Tree distance 4 from its parent and at most 8 from its siblings. Therefore, we can cover $\mathcal{T}_{2n+1}^{(2)}$ using $\mathcal{T}_{2n-1}^{(2)}$ balls of radius 8. Recursively, we can show that $\mathcal{T}_{2n+1}^{(2)}$ can be covered using $\mathcal{T}_{2(n-2k)+1}^{(2)}$ balls of radius $8k$. Using the asymptotic formula from Lemma 11, we obtain

$$\mathcal{N}(\mathcal{T}_{2n+1}^{(2)}, 8k) \leq w_{n-2k} = w_n \frac{w_{n-2k}}{w_n} = \frac{w_n}{b^{2k}}. \qquad \square$$

The following result derives tighter bounds for a specialized graph class constructed from paths.

**Proposition 56** (Proposition 51 in the main paper). *For the the graph class $\mathcal{F}_n \subset \mathcal{G}_n$ constructed above, we have*

$$\mathcal{N}(\mathcal{F}_n, \delta_\square^{\mathcal{T}}, 4) \leq \frac{2|\mathcal{F}_n/\sim_{\text{WL}}|}{n}.$$

*Proof.* First note that since 1-WL is complete in the space of forests, $|\mathcal{F}_n/\sim_{\text{WL}}|$ is exactly the number of non-isomorphic graphs in $\mathcal{F}_n$. We prove that $\mathcal{F}_n$ can be partitioned into $|\mathcal{P}(n)|$ subsets, each containing at least $n/2$ non-isomorphic graphs. Furthermore, within each partition set, all graphs are pairwise at Tree distance of at most 4. This result implies that while the total number of non-isomorphic graphs in $\mathcal{F}_n$ is greater than $|\mathcal{P}(n)| \cdot n/2$, we can cover them using only $|\mathcal{P}(n)|$ sets.

Now, for each $P \in \mathcal{P}(n)$, we define a set $S_P$ containing $n/2$ non-isomorphic graphs from $\mathcal{F}_n$ such that for all $P, P' \in \mathcal{P}(n)$, we have $S_P \cap S_{P'} = \emptyset$. We construct $S_P$ as follows. For each $P \in \mathcal{P}(n)$, let $P_{\max}$ denote the largest connected component of $P$. Choose, without loss of generality, a vertex $u \in V(P_{\max})$ with degree 1, i.e., a non-interior vertex) and define

$$S_P := \left\{ (V, E) \;\middle|\; V := V(p_n) \mathbin{\dot{\cup}} V(P), E := E(P) \mathbin{\dot{\cup}} E(p_n) \cup \{u, v\}, \text{ where } v = v_j \text{ for } j \in \lceil n/2 \rceil \right\}.$$

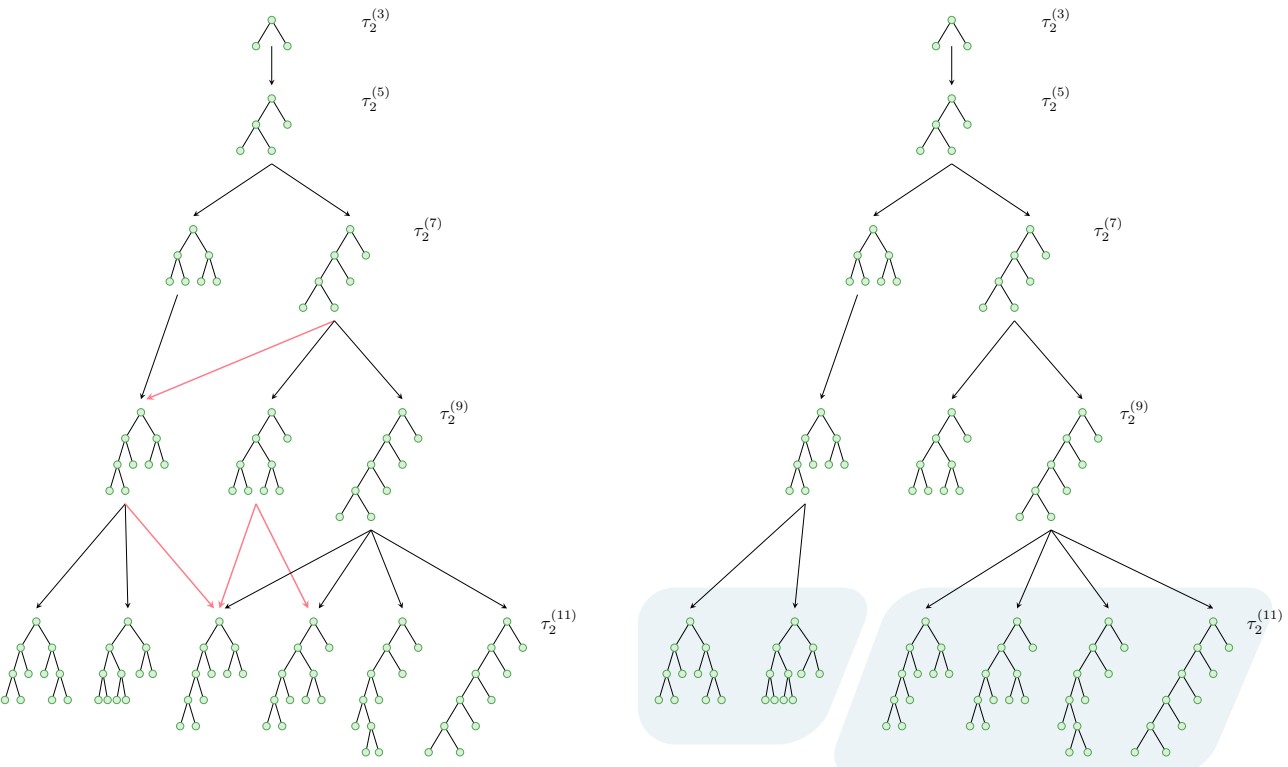

*Figure 6.* An illustration related to tree construction (left) and the grouping process for Otter trees (right) asymptotically leading to an exponential decrease of the covering number with the radius.

Clearly, $|S_P| = n/2$. We now verify the following claims.

1. For all $G_1, G_2 \in S_P$, $G_1$ and $G_2$ are non-isomorphic.

2. For all $P, P' \in \mathcal{P}(n)$, $S_P \cap S_{P'} = \emptyset$.

*Proof of (1).* Let $G_1, G_2 \in S_P$ with edge sets $E(G_1) = E(P) \cup \{u, v_{j_1}\}$ and $E(G_2) = E(P) \cup \{u, v_{j_2}\}$, where $j_1 \neq j_2$ and $j_1, j_2 \in \left\lceil \frac{n}{2} \right\rceil$. Assume further that $j_1, j_2 \neq 1$. If $j_1 < j_2$, then in $G_2$, starting from vertex $v_1$, we can trace a path of length $j_2 + |E(P_{\max})|$. In $G_1$, however, such a path does not exist. If $j_1 = 1 \neq j_2$, it is evident that $G_1$ has a connected component consisting of a path of length $n + |V(P_{\max})|$, which is absent in $G_2$.

*Proof of (2).* If $P$ and $P'$ have different numbers of disconnected components, then for all $G_1 \in S_P$ and $G_2 \in S_{P'}$, $G_1$ and $G_2$ will differ in edge count. Furthermore, if $P$ and $P'$ have the same number of connected components but $|P_{\max}| \neq |P'_{\max}|$, then the size of the largest connected component in $G_1$ will differ from that in $G_2$. Finally, if $P$ and $P'$ have identical numbers of connected components and $|P_{\max}| = |P'_{\max}|$, then the connected component size histograms of $G_1$ and $G_2$ must differ; otherwise, $P$ and $P'$ would be isomorphic. $\square$

The following result establishes that MPNNs using sum aggregation are Lipschitz continuous regarding the Forest distance.

**Lemma 57** (Lemma 14 in the main paper)*. For $L, n, d \in \mathbb{N}, M' \in \mathbb{R}$ and for all MPNNs in $\mathsf{MPNN}^{\mathrm{sum}}_{L,M',L_{\mathsf{FNN}}}(\mathcal{G}^{\mathbb{R}}_{n,d})$, we have*

$$\|\boldsymbol{h}_G - \boldsymbol{h}_H\|_2 \leq \frac{1}{n} C(L) L_\psi \prod_{i=1}^{L} L_{\varphi_i} \cdot \mathrm{FD}_L(G, H), \text{ for all } G, H \in \mathcal{G}^{\mathbb{R}}_{n,d},$$

*where $C(L)$ is a constant depending on $L$*

*Proof.* Let $\theta$ be an edge-preserving bijection between the padded forests $F_{G,L}$ and $F_{H,L}$. This bijection $\theta$ induces another bijection, $\theta^{(l)}$, between vertices at layer $l$ of trees in $F_{G,L}$ and vertices at layer $l$ of trees in $F_{H,L}$, for each $l \in \{0, \ldots, L\}$. We

*Figure 7.* An example of graphs within the class $\mathcal{F}_n$ for n=7.

further define $\bar{\theta}^{(l)}(u,v) = 1$ iff $\theta^{(l)}(u) = v$ and 0 elsewhere. We present the proof for the case $L = 2$ and let $\boldsymbol{W}_t^{(1)}, \boldsymbol{W}_t^{(2)}$ be identical matrices; for $L > 2$ and any choice of matrices $\boldsymbol{W}_t^{(1)}, \boldsymbol{W}_t^{(2)}$, the proof is analogous, though the notation becomes more complex. Specifically, one may generalize by replacing $\boldsymbol{W}_t^{(1)}$ and $\boldsymbol{W}_t^{(2)}$ with Lipschitz continuous functions, and the proof would still hold. We show the Lipschitz property by recursively bounding the following difference:

$$
\begin{aligned}
\|\boldsymbol{h}_G - \boldsymbol{h}_H\| &= \left\| \psi\left( \frac{1}{n} \sum_{u \in V(G)} \boldsymbol{h}_u^{(2)} \right) - \psi\left( \frac{1}{n} \sum_{v \in V(H)} \boldsymbol{h}_v^{(2)} \right) \right\| \\
&\leq \frac{1}{n} L_\psi \sum_{u \in V(G), v \in V(H)} \bar{\theta}^{(0)}(u,v) \left\| \boldsymbol{h}_u^{(2)} - \boldsymbol{h}_v^{(2)} \right\|,
\end{aligned}
$$

where in the last step, we applied the triangle inequality, as implied by the bijection $\theta^{(0)}$ between the roots.

Thus, we have that $\|\boldsymbol{h}_G - \boldsymbol{h}_H\|$ is less than

$$
\begin{aligned}
&\frac{1}{n} L_\psi \sum_{u \in V(G), v \in V(H)} \bar{\theta}^{(0)}(u,v) \left\| \varphi_2\left( \boldsymbol{h}_u^{(1)} + \sum_{u' \in N(u)} \boldsymbol{h}_{u'}^{(1)} \right) - \varphi_2\left( \boldsymbol{h}_v^{(1)} + \sum_{v' \in N(v)} \boldsymbol{h}_{v'}^{(1)} \right) \right\| \\
&\leq \frac{1}{n} L_\psi L_{\varphi_2} \sum_{u \in V(G), v \in V(H)} \bar{\theta}^{(0)}(u,v) \left\| \boldsymbol{h}_u^{(1)} - \boldsymbol{h}_v^{(1)} + \sum_{u' \in N(u)} \boldsymbol{h}_{u'}^{(1)} - \sum_{v' \in N(v)} \boldsymbol{h}_{v'}^{(1)} \right\| \\
&\leq \underbrace{\frac{1}{n} L_\psi L_{\varphi_2} \sum_{u \in V(G), v \in V(H)} \bar{\theta}^{(0)}(u,v) \left\| \boldsymbol{h}_u^{(1)} - \boldsymbol{h}_v^{(1)} \right\|}_{A} \\
&\quad + \underbrace{\frac{1}{n} L_\psi L_{\varphi_2} \sum_{u \in V(G), v \in V(H)} \bar{\theta}^{(0)}(u,v) \sum_{u' \in N(u), v' \in N(v)} \bar{\theta}^{(1)}(u',v') \left\| \boldsymbol{h}_{u'}^{(1)} - \boldsymbol{h}_{v'}^{(1)} \right\|}_{B},
\end{aligned}
$$

where, in the last step, we again applied the triangle inequality, this time induced by the bijection $\theta^{(1)}$ between the vertices at the first layer. Note that it is possible that $|N(u)| > |N(v)|$, meaning some neighbors of $u$ are mapped by $\theta^{(1)}$ to padded vertices. In this case, we can trivially apply the triangle inequality by setting $\boldsymbol{h}_{v'}^{(1)} = 0$ in the expression above.

Next, we bound term $A$:

$$
\begin{aligned}
A &\leq \frac{1}{n} L_\psi L_{\varphi_2} L_{\varphi_1} \sum_{u \in V(G), v \in V(H)} \bar{\theta}^{(0)}(u,v) \left\| \boldsymbol{h}_u^{(0)} - \boldsymbol{h}_v^{(0)} + \sum_{u' \in N(u)} \boldsymbol{h}_{u'}^{(0)} - \sum_{v' \in N(v)} \boldsymbol{h}_{v'}^{(0)} \right\| \\
&\leq \frac{1}{n} L_\psi L_{\varphi_2} L_{\varphi_1} \sum_{u \in V(G), v \in V(H)} \bar{\theta}^{(0)}(u,v) \left\| \boldsymbol{h}_u^{(0)} - \boldsymbol{h}_v^{(0)} \right\|
\end{aligned}
$$

$$+ \frac{1}{n} L_\psi L_{\varphi_2} L_{\varphi_1} \sum_{u \in V(G), v \in V(H)} \bar{\theta}^{(0)}(u,v) \sum_{u' \in N(u), v' \in N(v)} \bar{\theta}^{(1)}(u',v') \left\| \boldsymbol{h}_{u'}^{(0)} - \boldsymbol{h}_{v'}^{(0)} \right\|.$$

Similarly, the term $B$ is bounded by

$$\leq \frac{1}{n} L_\psi L_{\varphi_2} L_{\varphi_1} \sum_{u \in V(G), v \in V(H)} \bar{\theta}^{(0)}(u,v) \sum_{u' \in N(u), v' \in N(v)} \bar{\theta}^{(1)}(u',v') \left\| \boldsymbol{h}_{u'}^{(0)} - \boldsymbol{h}_{v'}^{(0)} + \sum_{u'' \in N(u')} \boldsymbol{h}_{u''}^{(0)} - \sum_{v'' \in N(v')} \boldsymbol{h}_{v''}^{(0)} \right\|$$

$$\leq \frac{1}{n} L_\psi L_{\varphi_2} L_{\varphi_1} \sum_{u \in V(G), v \in V(H)} \bar{\theta}^{(0)}(u,v) \sum_{u' \in N(u), v' \in N(v)} \bar{\theta}^{(1)}(u',v') \left\| \boldsymbol{h}_{u'}^{(0)} - \boldsymbol{h}_{v'}^{(0)} \right\|$$

$$+ \frac{1}{n} L_\psi L_{\varphi_2} L_{\varphi_1} \sum_{u \in V(G), v \in V(H)} \bar{\theta}^{(0)}(u,v) \sum_{u' \in N(u), v' \in N(v)} \bar{\theta}^{(1)}(u',v') \sum_{u'' \in N(u'), v'' \in N(v')} \bar{\theta}^{(2)}(u'',v'') \left\| \boldsymbol{h}_{u''}^{(0)} - \boldsymbol{h}_{v''}^{(0)} \right\|.$$

Therefore, choosing as $\theta$ the edge-preserving bijection minimizes Equation (9) we get,

$$\| \boldsymbol{h}_G - \boldsymbol{h}_H \| \leq 2 \frac{1}{n} L_\psi L_{\varphi_2} L_{\varphi_1} \mathrm{FD}_L(G,H). \qquad \square$$

The following result derives a bound on the generalization error for MPNNs using sum aggregation.

**Proposition 58** (Proposition 52 in the main paper). *For every $L, n \in \mathbb{N}$, $M' \in \mathbb{R}$ and for all MPNNs in $\mathrm{MPNN}_{L,M',L_{\mathsf{FNN}}}^{\mathrm{sum}}(\mathcal{G}_n)$, we have the following generalization bound: For any sample $\mathcal{S}$, and any $\delta \in (0,1)$, with probability at least $1 - \delta$,*

$$|\ell_{\exp}(h_{\mathcal{S}}) - \ell_{\mathrm{emp}}(h_{\mathcal{S}})| \leq 2 \cdot L_{\mathsf{FNN}} \cdot \bar{\gamma}^{\leftarrow}\left(\frac{2\varepsilon}{n^2}\right) + M' \sqrt{\frac{\mathcal{N}(\mathcal{G}_n, \delta_\square^{\mathcal{T}}, \varepsilon) \frac{4M' \log(2)}{\varepsilon} + 2\log\left(\frac{1}{\delta}\right)}{|\mathcal{S}|}},$$

*for $\varepsilon < 2M'$.*

*Proof.* Similarly to the classification setting we apply Proposition 7. First note that $\ell$ is 2-Lipschitz since,

$$\begin{aligned}
|\ell(x_1, y_1) - \ell(x_2, y_2)| &= ||x_1 - y_1| - |x_2 - y_2|| \\
&\leq |x_1 - x_2| + |y_1 - y_2| \\
&\leq 2 \max\{|x_1 - x_2|, |y_1 - y_2|\}.
\end{aligned}$$

Now, let $\mathcal{Z} = \mathcal{G}_n \times [-M', M']$. For a given $\varepsilon > 0$, graphs $G_1, G_2 \in \mathcal{G}_n$, and $y_1, y_2 \in [0,1]$, if $\max\{\delta_\square^{\mathcal{T}}(G_1, G_2), |y_1, y_2| < \bar{\gamma}(\varepsilon)n^2$, we have that for any sample $\mathcal{S}$,

$$\begin{aligned}
|\ell(h_{\mathcal{S}}(G_1), y_1) - \ell(h_{\mathcal{S}}(G_2), y_2)| &\leq 2 \max\{|h_{\mathcal{S}}(G_1) - h_{\mathcal{S}}(G_2)|, |y_1, y_2|\} \\
&\leq 2 \cdot L_{\mathsf{FNN}} \cdot \varepsilon.
\end{aligned}$$

Therefore, by Proposition 7, any learning algorithm for $\mathrm{MPNN}_{L,M',L_{\mathsf{FNN}}}^{\mathrm{ord}}(\mathcal{G}_n)$ is

$$\left( \mathcal{N}\left((0,1), |\cdot|, \frac{\bar{\gamma}(\varepsilon)n^2}{2}\right) \cdot \mathcal{N}\left(\mathcal{G}_n, \delta_\square^{\mathcal{T}}, \frac{\bar{\gamma}(\varepsilon)n^2}{2}\right), 2 \cdot L_{\mathsf{FNN}} \cdot \varepsilon \right)\text{-robust.}$$

Similarly we replace $\varepsilon$ with $\bar{\gamma}^{\leftarrow}\left(\frac{2\varepsilon}{n^2}\right)$ and we get $\mathcal{N}(\mathcal{G}_n, \delta_\square^{\mathcal{T}}, \frac{\bar{\gamma}(\bar{\gamma}^{\leftarrow}(\frac{2\varepsilon}{n^2}))n^2}{2}) \leq \mathcal{N}(\mathcal{G}_n, \delta_\square^{\mathcal{T}}, \varepsilon)$, leading to the following generalization bound: For any sample $\mathcal{S}$, and any $\delta \in (0,1)$, with probability at least $1 - \delta$, $|\ell_{\exp}(h_{\mathcal{S}}) - \ell_{\mathrm{emp}}(h_{\mathcal{S}})|$ is less than

$$2 \cdot L_{\mathsf{FNN}} \cdot \bar{\gamma}^{\leftarrow}\left(\frac{2\varepsilon}{n^2}\right) + M' \sqrt{\frac{\mathcal{N}([-M', M'], |\cdot|, \varepsilon) \cdot \mathcal{N}(\mathcal{G}_n, \delta_\square^{\mathcal{T}}, \varepsilon) 2 \log(2) + 2 \log\left(\frac{1}{\delta}\right)}{|\mathcal{S}|}},$$

or,

$$|\ell_{\exp}(h_{\mathcal{S}}) - \ell_{\mathrm{emp}}(h_{\mathcal{S}})| \leq 2 \cdot L_{\mathsf{FNN}} \cdot \bar{\gamma}^{\leftarrow}\left(\frac{2\varepsilon}{n^2}\right) + M' \sqrt{\frac{\mathcal{N}(\mathcal{G}_n, \delta_\square^{\mathcal{T}}, \varepsilon) \frac{4M' \log(2)}{\varepsilon} + 2\log\left(\frac{1}{\delta}\right)}{|\mathcal{S}|}},$$

for $\varepsilon < 2M'$. $\qquad \square$

The following results upper bounds the covering number regarding the Forest distance using the number of 1-WL-distinguishable graphs after $L$ iterations.

**Proposition 59** (Proposition 16 in the main paper). *For $n, q, d, L \in \mathbb{N}$, and $M' \in \mathbb{R}$, for any graph learning algorithm for the class* $\mathsf{MPNN}^{\mathrm{sum}}_{L,M',L_{\mathsf{FNN}}}(\mathcal{G}_{n,d,q})$ *and for any sample $\mathcal{S}$ and $\delta \in (0,1)$, with probability at least $1 - \delta$, we have*

$$|\ell_{\exp}(h_{\mathcal{S}}) - \ell_{\mathrm{emp}}(h_{\mathcal{S}})| \leq 2\widetilde{C}b(d,q,L)k + M\sqrt{\frac{\frac{m_{n,d,q,L}}{k+1}4\log(2) + 2\log(1/\delta)}{|\mathcal{S}|}}, \quad \text{for } k \in \mathbb{N},$$

*where $\widetilde{C} = 2/nL_\ell L_{\mathsf{FNN}}C_{\mathrm{FD}_L}$, $C_{\mathrm{FD}_L} = C(L)L_\psi \prod_{i=1}^{L} L_{\varphi_i}$ is the Lipschitz constant in Lemma 14 and $M$ is an upper bound of the loss function $\ell$.* $\qquad\square$

*Proof.* It suffices to show that for all $n, L, d, q \in \mathbb{N}$, we have the following bound for the covering number.

$$\mathcal{N}(\mathcal{G}_{n,d,q}, \mathrm{FD}_L, 2b(d,q,L)k) \leq \frac{m_{n,d,q,L}}{k+1}.$$

Let $\mathcal{G}_{n,d,q}/_{\sim\mathsf{WL}_L}$ be the quotient space defined by 1-WL after $L$ iterations on $\mathcal{G}_{n,d,q}$, i.e., $m_{n,d,q,L} = |\mathcal{G}_{n,d,q}/_{\sim\mathsf{WL}_L}|$. We denote the equivalence classes in this space by $[G]_L$. Following the proof in Theorem 10, it suffices to construct a rooted tree with labels from $\mathcal{G}_{n,d,q}/_{\sim\mathsf{WL}_L}$ satisfying the following properties,

- The total number of vertices is $m_{n,d,q,L}$, and each vertex has a unique label.
- The graph corresponding to each parent vertex is at most at Forest distance (depth $L$) of $b(d,q,L)$ from the graphs corresponding to its children.
- Graphs corresponding to sibling vertices are at most at Forest distance (depth $L$) of $2b(d,q,L)$.

We construct such a tree as follows:

1. The root (at level $l = 0$) corresponds to the empty graph on $n$ vertices, where each vertex has the same feature. Without loss of generality, we assume this feature to be $(1,0,\ldots,0) \in \mathbb{R}^{1 \times d}$.

2. At the next level $(l + 1)$, for a vertex at level $l$ with label, say, $[G_i]_L$, we compute its children as follows. We first calculate all graphs in the set
$$[G_i]_{L,c} := [G_i]^{(e)}_{L,c} \cup [G_i]^{(f)}_{L,c},$$
where
$$[G_i]^{(e)}_{L,c} = \{G' \in \mathcal{G}_{n,d,q} \mid \exists\, G \in [G_i]_L,\, e \in E(G') \text{ such that } G' \setminus \{e\} \simeq G\},$$
and
$$[G_i]^{(f)}_{L,c} = \Big\{G' \in \mathcal{G}_{n,d,q} \mid \exists\, G \in [G_i]_L,\, u \in V(G),\, u' \in V(G') \text{ such that }$$
$$\text{changing } \ell_G(u) \text{ to } \ell_{G'}(u') \text{ implies } G \simeq G'\Big\}.$$

   We connect all graphs in $[G_i]_{L,c}$ to the vertex $[G_i]_L$. Then, we prune the tree by arbitrarily removing all 1-WL-equivalent (after $L$ iterations) graphs, retaining only one as the representative.

3. We continue this process until all graph classes in $\mathcal{G}_{n,d,q}/_{\sim\mathsf{WL}_L}$ have been used as labels.

$\qquad\square$

The following result shows the Lipschitz continuity property of MPNNs using mean aggregation regarding the mean-Forest distance.

**Lemma 60** (Lemma 48 in the main paper). *For $L, n, d \in \mathbb{N}, M' \in \mathbb{R}$ and for all MPNNs in* $\mathsf{MPNN}^{\mathrm{mean}}_{L,M',L_{\mathsf{FNN}}}(\mathcal{G}^{\mathbb{R}}_{n,d})$, *we have*

$$\|\boldsymbol{h}_G - \boldsymbol{h}_H\|_2 \leq \frac{1}{n}C^{(m)}(L)L_\psi \prod_{i=1}^{L} L_{\varphi_i}\mathrm{FD}^{\mathrm{m}}_L(G,H), \quad \text{for all } G, H \in \mathcal{G}^{\mathbb{R}}_{n,d},$$

*where $C^{(m)}(L)$ is a constant depending on $L$.*

*Proof.* Let $\theta$ be an edge-preserving bijection between the padded forests $F_{G,L}^{(m)}$ and $F_{H,L}^{(m)}$. This bijection $\theta$ induces another bijection, $\theta^{(l)}$, between vertices at layer $l$ of trees in $F_{G,L}^{(m)}$ and vertices at layer $l$ of trees in $F_{H,L}^{(m)}$, for $l \in \{0, \ldots, L\}$. Similarly, we define $\bar{\theta}^{(l)}(u, v) = 1$ iff $\theta^{(l)}(u) = v$ and 0 elsewhere. We present the proof for the case $L = 2$ with $\boldsymbol{W}_t^{(1)}, \boldsymbol{W}_t^{(2)}$ as identical matrices; for $L > 2$ and any choice of matrices $\boldsymbol{W}_t^{(1)}, \boldsymbol{W}_t^{(2)}$, the proof is analogous, though the notation becomes more complex. Specifically, one may generalize by replacing $\boldsymbol{W}_t^{(1)}$ and $\boldsymbol{W}_t^{(2)}$ with Lipschitz continuous and positive homogeneous functions, and the proof would still hold. For each $u \in V(G)$, we denote by $N_u$ the number of neighbors of $u$ (i.e., $N_u = |N(u)|$). Then,

$$
\begin{aligned}
\|\boldsymbol{h}_G - \boldsymbol{h}_H\| &= \left\| \psi\left( \frac{1}{n} \sum_{u \in V(G)} \boldsymbol{h}_u^{(2)} \right) - \psi\left( \frac{1}{n} \sum_{v \in V(H)} \boldsymbol{h}_v^{(2)} \right) \right\| \\
&\leq \frac{1}{n} L_\psi \sum_{u \in V(G), v \in V(H)} \bar{\theta}^{(0)}(u, v) \left\| \boldsymbol{h}_u^{(2)} - \boldsymbol{h}_v^{(2)} \right\|,
\end{aligned}
$$

where we have just applied the triangle inequality, as implied by the bijection $\theta^{(0)}$ between the roots. Thus, we have that $\|\boldsymbol{h}_G - \boldsymbol{h}_H\|$ is bounded by

$$
\frac{1}{n} L_\psi \sum_{u \in V(G), v \in V(H)} \bar{\theta}^{(0)}(u, v) \left\| \varphi_2\left( \boldsymbol{h}_u^{(1)} + \frac{1}{N_u} \sum_{u' \in N(u)} \boldsymbol{h}_{u'}^{(1)} \right) - \varphi_2\left( \boldsymbol{h}_v^{(1)} + \frac{1}{N_v} \sum_{v' \in N(v)} \boldsymbol{h}_{v'}^{(1)} \right) \right\|
$$

$$
\leq \frac{1}{n} L_\psi L_{\varphi_2} \sum_{u \in V(G), v \in V(H)} \bar{\theta}^{(0)}(u, v) \left\| \boldsymbol{h}_u^{(1)} - \boldsymbol{h}_v^{(1)} + \sum_{u' \in N(u)} \frac{1}{N_u} \boldsymbol{h}_{u'}^{(1)} - \sum_{v' \in N(v)} \frac{1}{N_v} \boldsymbol{h}_{v'}^{(1)} \right\|
$$

$$
\underbrace{\leq \frac{1}{n} L_\psi L_{\varphi_2} \sum_{u \in V(G), v \in V(H)} \bar{\theta}^{(0)}(u, v) \left\| \boldsymbol{h}_u^{(1)} - \boldsymbol{h}_v^{(1)} \right\|}_{A}
$$

$$
+ \underbrace{\frac{1}{n} L_\psi L_{\varphi_2} \sum_{u \in V(G), v \in V(H)} \bar{\theta}^{(0)}(u, v) \sum_{u' \in N(u), v' \in N(v)} \bar{\theta}^{(1)}(u', v') \left\| \frac{\boldsymbol{h}_{u'}^{(1)}}{N_u} - \frac{\boldsymbol{h}_{v'}^{(1)}}{N_v} \right\|}_{B},
$$

where, in the last step, we again applied the triangle inequality induced by the bijection $\theta^{(1)}$ between the vertices at the first layer. Note that it is possible that $N_u > N_v$, meaning some neighbors of $u$ are mapped by $\theta^{(1)}$ to padded vertices. In this case, we can trivially apply the triangle inequality by setting $\boldsymbol{h}_{v'}^{(1)} = 0$ in the expression above.

Next, we bound term $A$, i.e.,

$$
\begin{aligned}
A &\leq \frac{1}{n} L_\psi L_{\varphi_2} L_{\varphi_1} \sum_{u \in V(G), v \in V(H)} \bar{\theta}^{(0)}(u, v) \left\| \boldsymbol{h}_u^{(0)} - \boldsymbol{h}_v^{(0)} + \frac{1}{N_u} \sum_{u' \in N(u)} \boldsymbol{h}_{u'}^{(0)} - \frac{1}{N_v} \sum_{v' \in N(v)} \boldsymbol{h}_{v'}^{(0)} \right\| \\
&\leq \frac{1}{n} L_\psi L_{\varphi_2} L_{\varphi_1} \sum_{u \in V(G), v \in V(H)} \bar{\theta}^{(0)}(u, v) \left\| \boldsymbol{h}_u^{(0)} - \boldsymbol{h}_v^{(0)} \right\| \\
&\quad + \frac{1}{n} L_\psi L_{\varphi_2} L_{\varphi_1} \sum_{u \in V(G), v \in V(H)} \bar{\theta}^{(0)}(u, v) \sum_{u' \in N(u), v' \in N(v)} \bar{\theta}^{(1)}(u', v') \left\| \frac{\boldsymbol{h}_{u'}^{(0)}}{N_u} - \frac{\boldsymbol{h}_{v'}^{(0)}}{N_v} \right\|.
\end{aligned}
$$

Similarly, we can bound the term $B$ by

$$
\frac{1}{n} L_\psi L_{\varphi_2} \sum_{u \in V(G), v \in V(H)} \bar{\theta}^{(0)}(u, v) \sum_{u' \in N(u), v' \in N(v)}
$$

$$\bar{\theta}^{(1)}(u', v') \cdot \left\| \varphi_1 \left( \frac{\boldsymbol{h}_{u'}^{(0)}}{N_u} + \frac{1}{N_u N_{u'}} \sum_{u'' \in N(u')} \boldsymbol{h}_{u''}^{(0)} \right) - \varphi_1 \left( \frac{\boldsymbol{h}_{v'}^{(0)}}{N_v} + \frac{1}{N_v N_{v'}} \sum_{v'' \in N(v')} \boldsymbol{h}_{v''}^{(0)} \right) \right\|,$$

where we have used the positive homogeneity property of $\varphi_1$. Therefore,

$$
\begin{aligned}
B \leq {} & \frac{1}{n} L_\psi L_{\varphi_2} L_{\varphi_1} \sum_{u \in V(G), v \in V(H)} \bar{\theta}^{(0)}(u, v) \sum_{u' \in N(u), v' \in N(v)} \bar{\theta}^{(1)}(u', v') \left\| \frac{\boldsymbol{h}_{u'}^{(0)}}{N_u} - \frac{\boldsymbol{h}_{v'}^{(0)}}{N_v} \right\| \\
& + \frac{1}{n} L_\psi L_{\varphi_2} L_{\varphi_1} \sum_{u \in V(G), v \in V(H)} \bar{\theta}^{(0)}(u, v) \sum_{u' \in N(u), v' \in N(v)} \bar{\theta}^{(1)}(u', v') \\
& \qquad\qquad\qquad \sum_{u'' \in N(u'), v'' \in N(v')} \bar{\theta}^{(2)}(u'', v'') \cdot \left\| \frac{\boldsymbol{h}_{u''}^{(0)}}{N_u N_{u'}} - \frac{\boldsymbol{h}_{v''}^{(0)}}{N_v N_{v'}} \right\|.
\end{aligned}
$$

Hence, choosing as $\theta$ the edge preserving bijection minimizes Appendix G we get,

$$\|\boldsymbol{h}_G - \boldsymbol{h}_H\| \leq 2\frac{1}{n} L_\psi L_{\varphi_2} L_{\varphi_1} \mathrm{FD}_L(G, H),$$

as desired. $\qquad\qquad\qquad\qquad\qquad\qquad\qquad\qquad\qquad\qquad\qquad\qquad\qquad\qquad\qquad\qquad$ $\square$

## P. Experimental results

In the following section, we describe the experiments in detail and report the results.

**Datasets**     To investigate **Q1**, we conducted experiments using the graph classes $\mathcal{G}_n$, $\mathcal{T}_n^{(2)}$, and $\mathcal{F}_n$ (defined in Appendix N). Additionally, we experimented with the binary classification real-world datasets MUTAG, NCI1, MCF-7H (Morris et al., 2020a), and OGBG-MOLHIV (Hu et al., 2020), to match our theoretical setup and address **Q1**, **Q2**, and **Q3**. For real-world datasets analyzed under **Q1**, we computed the covering number across all graphs up to a specified order, with smaller graphs being padded with isolated vertices, aligning their order. Similarly, for **Q3**, graphs were padded to match the order of the largest graph in the dataset; see Table 2 for dataset statistics and properties.

**Neural architectures**     To address **Q2**, we used randomly initialized GIN architectures (Feng et al., 2022) with varying numbers of layers and matching parameters for the Forest distance. Conversely, for **Q3**, we trained GIN architecture with three layers across five random seeds to estimate variance. Both architectures incorporated ReLU activation functions and sum pooling, disregarding potential edge labels. We tuned the feature dimension across the set 32, 64, 128, 256 based on validation set performance, training MUTAG and NCI1 for 100 epochs and MCF-7H and OGBG-MOLHIV for 20 epochs using the Adam optimizer (Kingma & Ba, 2015). The training setup included a learning rate of 0.001, a batch size of 128, and no learning rate decay or dropout across all datasets. All models were implemented with PyTorch Geometric (Fey & Lenssen, 2019) and executed on a system with 128GB of RAM and an Nvidia Tesla A100 GPU with 48GB of memory.

**Experimental protocol and model configuration**     For **Q1**, we calculated the covering number $\mathcal{N}(\cdot, \mathrm{FD}_3, \varepsilon)$ for various datasets, graph orders, and radii by solving the corresponding set cover problem. We also compare the bound on the covering number presented in Theorem 10 against the covering number $\mathcal{N}(\cdot, \delta_1^{\mathcal{T}}, \varepsilon)$ using the 1-norm instance of the cut norm as computing the cut norm is MaxSNP-hard. For **Q2**, we measure whether the Forest distance input perturbations lead to perturbations in the MPNN outputs as expected in Lemma 14. We used a random 80/10/10 split for training/validation/testing. Each plot included 1 000 data points generated by selecting a random graph size. The Forest distance for two graphs of the chosen size, randomly sampled from the dataset, can be compared with the Euclidean distance between their MPNN outputs. For **Q3**, we evaluated the generalization gap by estimating the train and test losses and compared it against our generalization bound derived for the Forest distance in Proposition 15. We utilized an upper estimate of the Lipschitz constant obtained from **Q2** and an $L_\ell = 1$ (Mao et al., 2023) to compute the bound. We selected the radii that yielded the tightest bound. See Table 3 for the values of $\widetilde{C}, C_{\mathrm{FD}_3}$, and $m_{n,d,3}$ used in the computation of our bound.

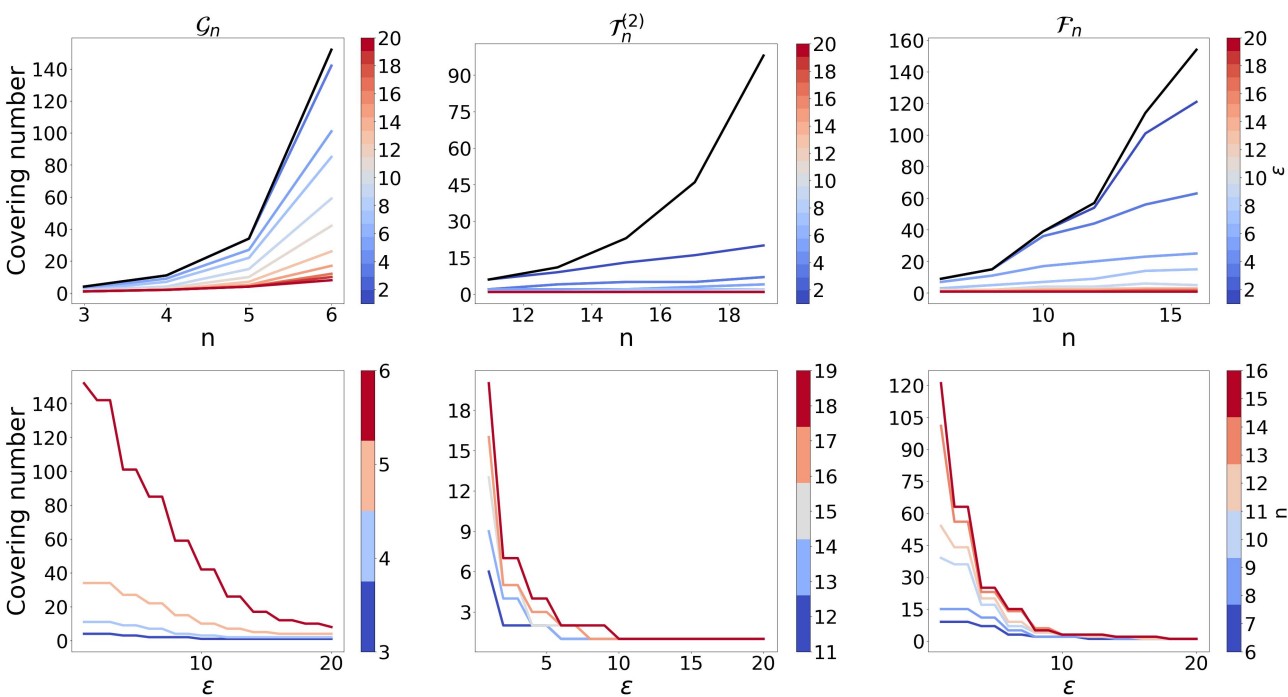

**Figure 8.** The covering number $\mathcal{N}(\cdot, \mathrm{FD}_3, \varepsilon)$ and the number of 1-WL indistinguishable graphs $m_n$ (upper row, black) for the graph families $\mathcal{G}_n$, $\mathcal{T}_n^{(2)}$, and $\mathcal{F}_n$ across different graph sizes $n$ and radii $\varepsilon$.

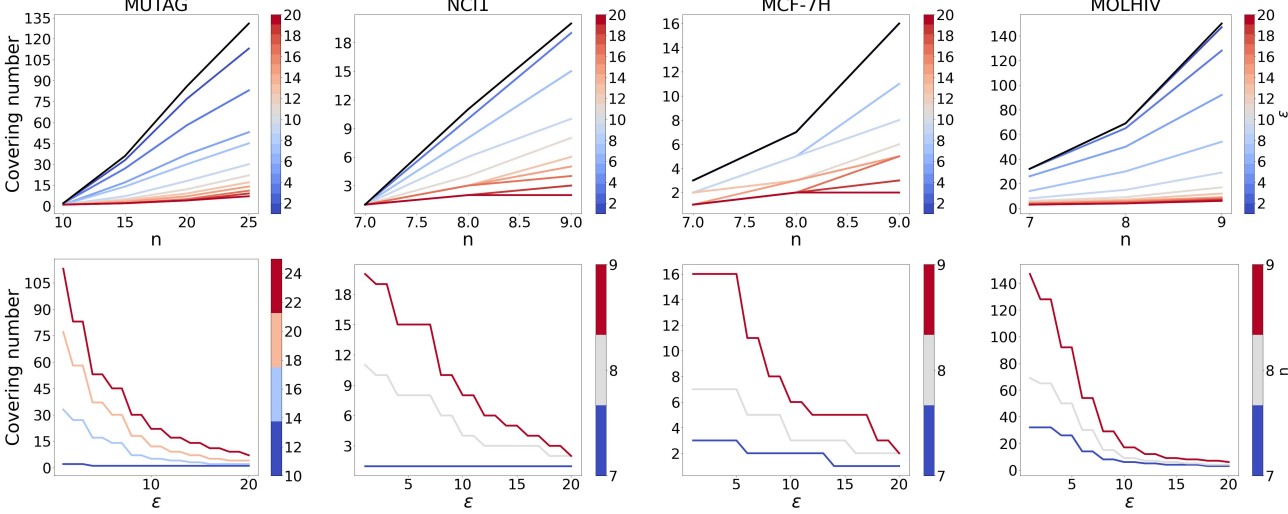

**Figure 9.** The covering number $\mathcal{N}(\cdot, \mathrm{FD}_3, \varepsilon)$ and the number of 1-WL indistinguishable graphs $m_n$ (upper row, black) for real-world datasets across varying graph sizes $n$ and radii $\varepsilon$.

# Q. Dataset statistics

The statistics of the real-world datasets can be found in Table 2, and estimates of the coefficients in Theorem 10 can be found in Table 3.

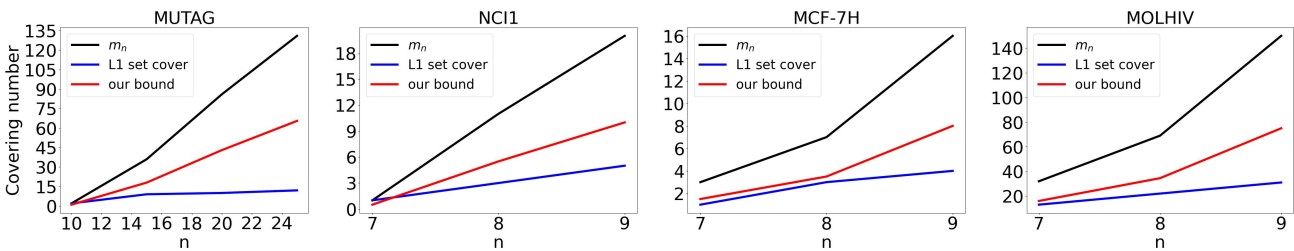

*Figure 10.* The covering number $\mathcal{N}(\cdot, \mathrm{FD}_3, 0\,7)$ bound, the 1-norm set cover and the number of 1-WL indistinguishable graphs $m_n$ for real-world datasets with varying graph orders.

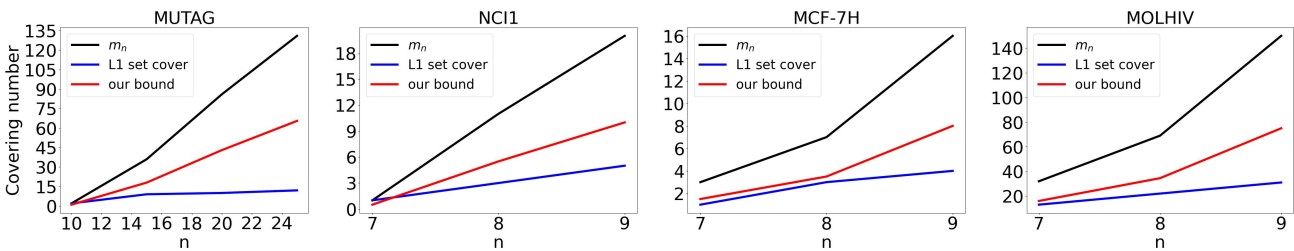

*Figure 11.* Correlation between MPNN outputs and the Forest distance FD across real-world datasets using a varying number of MPNN layers. The Pearson correlation coefficient $r$ between $\|h(G_1) - h(G_2)\|_2$ and the Forest distance $\mathrm{FD}_3$ are shown on the top left of the figures

*Table 2.* Statistics for the MUTAG, NCI1, MCF-7H, and OGBG-MOLHIV datasets.

|  | Dataset | | | |
|---|---|---|---|---|
|  | MUTAG | NCI1 | MCF-7H | OGBG-MOLHIV |
| # Graphs | 188 | 4 110 | 27 770 | 41 127 |
| # Classes | 2 | 2 | 2 | 2 |
| Avg. # vertices | 17.9 | 29.9 | 26.4 | 25.5 |
| Avg. # edges | 39.6 | 32.3 | 28.5 | 27.5 |

*Table 3.* Estimates of the coefficients in Proposition 15 for real-world datasets. We compute $m_{n,d,3}$ in a data-dependent fashion for each dataset, i.e., not for the whole graph class.

| | Dataset | | | |
|---|---|---|---|---|
| | MUTAG | NCI1 | MCF-7H | OGBG-MOLHIV |
| $\mathcal{S}$ | 150 | 3 288 | 22 216 | 32 902 |
| $m_{n,d,3}$ | 139 | 3 808 | 25 448 | 34 433 |
| $M$ | 5.890 | 2.389 | 0.497 | 1.704 |
| $C_{\mathrm{FD}_3}$ | 0.377 | 0.202 | 0.101 | 0.107 |
| $n$ | 28 | 111 | 244 | 222 |

