# OpenReview forum: "Covered Forest: Fine-grained generalization analysis of graph neural networks"
_ICML.cc/2025/Conference — ICML 2025 spotlightposter_

### Official Review · Reviewer_EBjh · 2025-03-13

**Overall Recommendation:** 3

**Summary:**

This paper presents a study on the generalization abilities of sum-aggregation message passing graph neural networks (GNNs) based on a covering number approach. Towards this goal, they employ the so-called forest distance pseudometric, which is an intuitive re-formulation of the tree mover’s distance by Chuang and Jegelka. In particular, they exploit the Lipschitzness of GNNs wrt this pseudometric, and use the results from Xu and Mannor to obtain generalization bounds. Moreover, the paper deals with GNNs with mean (rather than sum) aggregation by defining a slightly modified metric.

**Claims And Evidence:**

All claims are supported by formal proofs. Moreover, the theoretical results are supported by experimental evidence.

**Essential References Not Discussed:**

None

**Experimental Designs Or Analyses:**

The datasets chosen for the experimental evaluation are few, but for a theory-driven paper this is not a problem. The experimental setup described in section Q is sensible, and the code is available.

**Methods And Evaluation Criteria:**

The proposed theoretical approach of combining a metric for graphs such that GNNs are Lipschitz (or equicontinuous) to it with the robustness framework of Xu and Mannor indeed makes sense for deriving data-dependent generalization bounds for GNNs.
The datasets chosen for the experimental evaluation are few, but for a theory-driven paper this is not a problem.

**Other Comments Or Suggestions:**

Minor comments:
* In eq. 7, A(G) is never defined (only in the Appendix). Please double check that everything is defined properly.
* in lemma 2 the comma in the "only if" is misplaced
* the footnote 7 on the "technical reasons" should really be part of the Proposition.

**Other Strengths And Weaknesses:**

Strengths:
- the paper seems to be technically sound, far as I can tell the proofs seem correct.
- the paper addresses a very relevant problem, obtaining tight generalization bounds for GNNs.

 Weaknesses:
- as already mentioned, the paper is somewhat incremental on the papers it is based on.
- the paper is very heavy and difficult to understand, and at times it lacks clarity in what it wants to achieve and how to prove it. It might be beneficial to add some more intuitive discussion between the various propositions. For example:
    -  an intuition of the behavior of $\bar\gamma^{\leftarrow}$ would help understand the results of Proposition 9 and its consequences
    - an explanation on the choice of using the Tree distance rather than the Forest distance (see also questions)
    - an intuition on the construction of Theorem 10 (again, see questions)
- the fact that an estimated Lipschitz constant (rather than an upper bound for the entire GNN class) is used in the results of Table 1 should be stated clearly, and not hidden in Section Q.

**Questions For Authors:**

- Why do you use the Tree distance for unlabeled graphs and the Forest distance for labeled graphs? Could one not use always the Forest distance? This distinction should be clarified.
- I don't understand the relevance of the subsection on Otter trees, as this is an artificial setting with little practical relevance. Could you motivate your choice of inserting this in the main paper? I think that, e.g., the construction of Theorem 10 that allows you to obtain a cover of size m/(k+1) is much more interesting.


----
#### Update
After the rebuttal, my recommendation remains mostly positive.

**Relation To Broader Scientific Literature:**

The paper addresses a gap in the literature, namely obtaining tight data-dependent generalization bounds for GNNs. The results are non-trivial, but somewhat incremental on the papers it is based on. In particular, the tree distance is obtained from Böker 2021 and the forest distance is a re-formulation of the tree over distance by Chuang and Jegelka 2022. Moreover, the generalization bounds are obtained from the pre-existing work of Xu and Mannor. Nonetheless, I think this is an important advancement in the learning theory of GNNs.

**Theoretical Claims:**

I checked the proofs for most of the claims in the main paper, albeit not in depth, and as far as I can tell they seem correct.

---

> ### Author Rebuttal · Authors · 2025-03-27
>
> **We thank the reviewer for their fair and constructive review.**
>
> > the paper is very heavy and difficult to understand, and at times it lacks clarity in what it wants to achieve and how to prove it. It might be beneficial to add some more intuitive discussion between the various propositions. For example:
> an intuition of the behavior of \widebar{\gamma}^{\leftarrow} would help understand the results of Proposition 9 and its consequences
> an explanation on the choice of using the Tree distance rather than the Forest distance (see also questions)
>
> We agree that the paper is relatively dense. We will use the additional page in the camera-ready version of the paper to add more intuition and guide the reader. For example, we will add intuitive explanations before a theorem/proposition.
>
> > the fact that an estimated Lipschitz constant (rather than an upper bound for the entire GNN class) is used in the results of Table 1 should be stated clearly, and not hidden in Section Q.
>
> We will explicitly mention that we do not compute the uniform Lipschitz constant for our experiments but instead use an empirical Lipschitz constant.
>
> >Minor comments:
> >In eq. 7, A(G) is never defined (only in the Appendix). Please double check that everything is defined properly.
> >in lemma 2 the comma in the "only if" is misplaced
> >the footnote 7 on the "technical reasons" should really be part of the Proposition.
>
> Thank you for your comments. We will consider all minor comments.
>
> > Why do you use the Tree distance for unlabeled graphs and the Forest distance for labeled graphs? Could one not use always the Forest distance? This distinction should be clarified.
>
> The decision to use both the tree distance and forest distance is motivated by:
>
> (i) Computational complexity: The tree distance with the L2-norm can be computed in polynomial time (see our response to Reviewer e2Pe), and we will move this explanation to the main text.
>
> (ii) The tree distance allows us to derive generalization bounds for graph classes without requiring bounded degree assumptions on graphs, which are necessary for the finiteness of the covering number using the forest distance.
>
> > I don't understand the relevance of the subsection on Otter trees, as this is an artificial setting with little practical relevance. Could you motivate your choice of inserting this in the main paper? I think that, e.g., the construction of Theorem 10 that allows you to obtain a cover of size m/(k+1) is much more interesting.
>
> The motivation was to show that in this specific graph class, the covering number admits an upper bound that decreases exponentially with the radius. This shows that our framework yields significantly tighter generalization bounds to structurally more simple graphs.
>
>
> **Please consider updating your score if you are satisfied with our answer. We are happy to answer any remaining questions.**

---

> > ### Comment · Reviewer_EBjh · 2025-04-02
> >
> > Thanks for the reply. I'm confident that by addressing my and the other reviewer's comments, the paper will improve even further. Because of this, my recommendation remains positive after the rebuttal. However, I'm not fully convinced by some answers. I still find the result on Otter trees too artificial. Moreover, I understand the advantages of the tree distance, but not why you applied this only to unlabeled graphs. Couldn't you use it also on labeled graphs and obtain results for both distances on both classes of graphs?
> > Because of this, but mostly due to the general lack of clarity in the paper, and since at the moment I cannot assess how this will be addressed, I don't feel comfortable raising the score beyond a weak accept.

---

> > > ### Author Response · Authors · 2025-04-02
> > >
> > > Thanks for your response regarding the extension of the tree distance for labeled graphs. We agree that a uniform treatment would be preferred; however, proving uniform continuity of tree distance to labeled graphs is highly nontrivial and open. We are actively working on this problem.
> > >
> > > We will make this clearer in the revised version of the paper to streamline presentation.

---

### Official Review · Reviewer_e2Pe · 2025-03-13

**Overall Recommendation:** 4

**Summary:**

This paper presents a new framework for analyzing generalization properties of Message-Passing Neural Networks (MPNNs) via fine-grained graph pseudo-metrics. These distances capture subtle structural similarities that the usual 1-WL equivalence classes overlook. The key theoretical results show that MPNNs of varying aggregation schemes (sum, mean) remain Lipschitz with respect to these distances, allowing the authors to derive covering number based generalization bounds that account for partial graph similarity rather than a purely binary (same/different) classification. Empirically, the paper demonstrates that these refined metrics yield more accurate coverings and tighter bounds, thereby better predicting real-world generalization gaps.

**Claims And Evidence:**

The authors provide clear definitions and statements of theoretical results and provide proofs for all theoretical claims.

Their experiments, while somewhat limited in scope and scale, show results consistent with the theoretical discussion; in particular, the coverage size shrinks significantly under these pseudo-metrics, which corresponds to smaller generalization gaps (Tab. 1). This addresses (though does not exhaustively prove) the claim of better real-world bounds.

Overall, the main claims are supported by theoretical analysis and experiments, prior work on 1-WL alignment and generalization bounds is discussed in detail.

**Essential References Not Discussed:**

N/A

**Experimental Designs Or Analyses:**

The experiments mostly use standard GNN benchmarks (MUTAG, NCI1, etc.) and includes a demonstration of how the covering number changes with the radius parameter. The authors show correlations (Figures 8–11 in appendix) between the distances and MPNN output differences, which supports the Lipschitz-type claim.

While sample sizes in these experiments are typical for classification tasks on molecular graphs, it might be useful to see a broader range of data sets, possibly with more variety in graph topologies. But the presented experiments are overall adequate and support the theoretical results.

**Methods And Evaluation Criteria:**

The theoretical analysis is overall sound. The data selected for experiments seems appropriate (though somewhat limited in scale and scope).

**Other Comments Or Suggestions:**

N/A

**Other Strengths And Weaknesses:**

Strengths:
- The introduction of refined metrics that preserve WL-like expressivity but provide more “granularity” is conceptually novel and addresses gaps in prior analyses.
- The Lipschitz continuity argument for sum- and mean-aggregation MPNNs is handled neatly and supports a variety of common network layers.
- The paper is well-written.

Weaknesses:
- The paper’s experimental scope is somewhat narrow, focusing on a small set of molecules/graphs; the empirical portion might be strengthened by more extensive analysis, including for other graph topologies.
- One potential weakness is the computational complexity of computing pseudo-metrics at scale on large graphs. The authors do mention computational complexity but might elaborate more on possible approximations.

**Questions For Authors:**

see weaknesses

**Relation To Broader Scientific Literature:**

The paper clearly connects to the established line of work about 1-WL expressivity (Morris et al., Xu et al.). It complements prior work that used coarser, purely discrete distances (i.e., 0 vs. 1 if the graphs are 1-WL distinguishable). The authors also provide comprehensive references to relevant works on GNN generalization. I did not notice major omissions in related literature.

**Theoretical Claims:**

The paper provides clear statements of theoretical claims and proves. Proof sketches/ short descriptions of the proof idea are given in the main text, more detailed analysis is in the appendix.

While I have not checked every detail in the proofs in the appendix, the arguments and line of reasoning seems correct, I did not see any obvious gaps or errors.

---

> ### Author Rebuttal · Authors · 2025-03-27
>
> **We thank the reviewer for their detailed and constructive review.**
>
> > One potential weakness is the computational complexity of computing pseudo-metrics at scale on large graphs. The authors do mention computational complexity but might elaborate more on possible approximations.
>
> Thank you for highlighting computational complexity. Throughout the paper, we discuss two main distances: the tree distance (with either the cut norm or the l2-norm) and the forest distance.
>
> As shown in Lemma 3, the forest distance is equivalent to the Tree Mover’s Distance (TMD) introduced by Chuang et al. This allows us to compute it efficiently using dynamic programming (as shown in the original paper) in time: $\mathcal{O}(t(n) + Lnt(q))$, where $q$ is the maximum degree of a node (in both graphs), $n$ the number of nodes, and $t(m) = O(m^3 log(m))$.
>
> The tree distance with the L2-norm can be computed in polynomial time (more precisely, an $\varepsilon$-solution), as detailed in “Interior-Point Polynomial Algorithms in Convex Programming” (Section 6.3.3).
>
> On the other hand, the cut norm poses a much more difficult combinatorial optimization problem. Its computation is indeed challenging. One of the most well-known approximation techniques is given in “Approximating the Cut-Norm via Grothendieck’s Inequality”. Additional relevant works will be added to the main paper (“Random Sampling and Approximation of MAX-CSP Problems”, Alon et al., “Quick Approximation to Matrices and Applications”, Frieze & Kannan.)
>
> It is worth mentioning that for the theoretical results in our paper, either the tree distance with the cut norm or with the l2-norm can be used interchangeably, as they define the same topology—this is discussed in “Fine-Grained Generalization Analysis.”
>
> The revised paper will add a detailed discussion regarding the pseudo-metrics computational complexity.
>
> > The paper’s experimental scope is somewhat narrow, focusing on a small set of molecules/graphs; the empirical portion might be strengthened by more extensive analysis, including for other graph topologies
>
> This is a good point. However, we view our work as theoretical work. We will try our best to include experiments with other graph structures,
>
> **Please consider updating your score if you are satisfied with our answer. We are happy to answer any remaining questions.**

---

> > ### Comment · Reviewer_e2Pe · 2025-04-03
> >
> > Thanks to the authors for the detailed rebuttal. I will maintain my score of acceptance.

---

### Official Review · Reviewer_QFxN · 2025-03-14

**Overall Recommendation:** 4

**Summary:**

This paper first defines three pseudo-distances on graphs compatible with 1-WL or its variants. First, the labeled tree distance, which is an extension of the tree distance to graphs with node features, is defined, and the equivalence with 1-WL indistinguishability is shown. Next, the forest distance is defined using graph unrolling, and the equivalence with 1-WL indistinguishablity up to L-time unrolling is shown. Finally, the mean forest distance is defined as a variant of the forest distance, and the equivalence with 1-MWL is shown.

The generalization error bounds for MPPNs are derived using these distances by showing that MPPNs are equicontinuous with respect to the tree distance and Lipschitz continuous with respect to the forest distance. These guarantee the robustness of MPPNs, which in turn gives generalization error bounds in terms of the covering number.

Numerical experiments evaluate the relationship between the covering number and WL-indistinguishable graphs, the relationship between the forest distance and the output of MPN, and the relationship between the generalization performance and the generalization bounds.


## update after rebuttal

I thank the authors for responding to my review comments. I am satisfied with them. Although the numerical evaluations align with theoretical analyses, I think we need more significance to be eligible for strong acceptance. Therefore, I keep my score (4. accept)

**Claims And Evidence:**

The main claim of this paper is that introducing similarity-based distances, such as the labeled tree and forest distances, makes generalization bounds tighter than existing VC-dimension-based bounds. This claim is supported by Proposition 9 (for MPPNs with order-normalized sum aggregation on unlabelled $n$-order graphs), Proposition 15 (for MPPNs with sum aggregation on attributed bounded-degree graphs), and their corollaries.

It is claimed that these theoretical results provide insight into the generalization performance of empirical MPPNs. However, as discussed below, I have questions about the design and interpretation of the numerical experiments in Section 5 and think that they are not sufficient as evidence.

**Essential References Not Discussed:**

As far as I checked the related studies, the coverage of existing studies on generalization analysis for the node and graph prediction tasks is OK. However, I may have missed some important literature since I am less familiar with graph prediction tasks.

**Ethical Review Concerns:**

N.A.

**Experimental Designs Or Analyses:**

I checked Sections Q and R.

Although $\mathcal{F}_n$ is referred to as a set of graphs, it is not defined in the main text or appendix. I suggest adding its definition.

**Methods And Evaluation Criteria:**

The theoretical analysis is appropriate to support the claim because, in general, the generalization bounds by the covering number are tighter than that by the VC dimension. In addition, Appendix L, in which the bound using the VC dimension is derived from the result of this paper, justifies the claim.

On the other hand, the three numerical experiments in Section 5 are not convincing. In particular, I have questions about Q1 and Q3:

Q1. We can expect the experiment's results from the definition of the covering number and WL-indistinguishability. Therefore, I have a question whether the observations from this experiment are new.
Q2. We see that forest distance and changes in the MPNN output are correlated and that small changes in forest distance do not significantly change output. This observation is consistent with Lemma 14.
Q3. The experiment shows that the generalization performance gap and theoretical upper bounds have the same scale. However, we cannot say the tightness of the bound from this observation.

**Other Comments Or Suggestions:**

1. this paper uses two terms (labeled and attributed) to describe a node with a feature. I want to clarify whether they are different concepts.
2. l.400(right): Unless I am missing something, $\mathcal{F}_n$ is undefined.
3. L.431(left): *How are the Forest distance and MPNN outputs correlated?* is better
4. L.1652 : *The above generalization bound*: I could not identify the bound this sentence refers to.
5. L.1696: *Theorem 35 in the main paper*: Theorem 35 is not in the main text but in the appendix.

**Other Strengths And Weaknesses:**

One of the weaknesses of this paper is the excessive appendix (37 pages), which reduces its clarity. For example, as this paper mainly focuses on generalization of MPNNs, the discussion on the mean forest distance seems to be different from its main scope.

On the other hand, I am positive about adding explanations on the cut norm, graphon, and WL algorithm in the appendix because they allow readers unfamiliar with these concepts to understand the paper in a self-contained manner.

**Questions For Authors:**

N.A.

**Relation To Broader Scientific Literature:**

This paper extends the tree distance proposed in [Böker, 2021] and defines labeled tree distance and forest distance. Then, as analogs of the tree distance, the equivalence between these distances and 1-WL indistinguishability (or its variants).

Furthermore, the robustness of MPNNs with respect to those pseudo distances is used to show the generalization performance bound of MPNNs for the covering number. This result is a refinement of the bound obtained in [Morris et al. (2023a)], which uses VC dimension.

**Theoretical Claims:**

I read the proofs of the theorems in this paper, specifically Section E, Propositon 38 in Section J, Propositon 43 in Section K, and Section O, and the lemmas associated with them. However, due to time constraints, I could not carry out a detailed check line by line.

---

> ### Author Rebuttal · Authors · 2025-03-27
>
> **We thank the reviewer for their fair and constructive review.**
>
> > Q1. We can expect the experiment's results from the definition of the covering number and WL-indistinguishability. Therefore, I have a question whether the observations from this experiment are new.
>
> You are correct that, by definition, we expect the covering number to decrease with the radius and increase with the order. What is new in our work is that we derive explicit upper bounds on the covering number for different graph families (n-order graphs, otter trees, and an artificial graph class), capturing how it scales with radius and graph order.
>
> Specifically:
>
> i) Proposition 10 shows that for n-order graphs, the covering number is upper bounded by $m_n/(k+1)$, showing a linear decrease with the radius, and an $\mathcal{O}(m_n)$ increase with $n$.
>
> ii) Proposition 12 establishes a stronger bound for Otter trees, showing exponential decay in the radius and again O(m_n) dependence on n.
>
> iii) Appendix N introduces an artificial graph class where the covering number scales with m_n/n instead of m_n.
>
> Hence, we empirically compute covering numbers in Figure 8 and observe trends that align with the theoretical bounds. These results help validate the tightness of our derived bounds. Indeed, for n-order graphs, the covering number can be bounded by a linear function on the radius, and for otter trees, it can be bounded by an exponential function on the radius.
>
> However, the current phrasing of Q1 may not fully reflect this. We will revise it to “To what extent do the empirical covering numbers for different graph families match the theoretical upper bounds derived in Section 4?.”
>
> > Q3. The experiment shows that the generalization performance gap and theoretical upper bounds have the same scale. However, we cannot say the tightness of the bound from this observation.
>
> Our goal in Q3 was not to claim that the bound is tight in the formal sense, but to show that the bound and the generalization gap are of the same scale in practice. We will clarify this point.
>
> > I checked Sections Q and R.
> > Although $\mathcal{F}_n$  is referred to as a set of graphs, it is not defined in the main text or appendix. I suggest adding its definition.
>
> Thank you for pointing this out. $\mathcal{F}_n$  is an artificial graph class defined in Appendix N. We will include its definition in the main text of the revised papers.
>
> > One of the weaknesses of this paper is the excessive appendix (37 pages), which reduces its clarity. For example, as this paper mainly focuses on generalization of MPNNs, the discussion on the mean forest distance seems to be different from its main scope.
> On the other hand, I am positive about adding explanations on the cut norm, graphon, and WL algorithm in the appendix because they allow readers unfamiliar with these concepts to understand the paper in a self-contained manner.
>
> We acknowledge that the appendix is quite extensive. However, our goal was to ensure that all notations and definitions used in the proofs are rigorously defined and that the paper remains self-contained for readers with varying backgrounds. For that reason, we will provide some additional discussion on Appendices D, K, and C about the intuition behind cut-norm, graphons, and the WL algorithm, respectively.
> We defined the mean forest distance to show that mean aggregation MPNNs satisfy the Lipschitz property for a coarser pseudo-metric (mean forest distance), leading to tighter generalization bounds. We will clarify this motivation in the main paper.
>
> Overall, we will use the additional page to make the main part more self-contained.
>
> > this paper uses two terms (labeled and attributed) to describe a node with a feature. I want to clarify whether they are different concepts.
> > l.400(right): Unless I am missing something, $\mathcal{F}_n$ is undefined.
> > L.431(left): How are the Forest distance and MPNN outputs correlated? is better
> > L.1652 : The above generalization bound: I could not identify the bound this sentence refers to.
> > L.1696: Theorem 35 in the main paper: Theorem 35 is not in the main text but in the appendix.
>
> Thank you for pointing this out.
> 1. The terms labeled and attributed refer to the same underlying concept, with a minor distinction noted in the background section: labeled typically refers to discrete labels drawn from a finite set, whereas attributed generally refers to continuous feature vectors in $\mathbb{R}^d$.
>
> 2.,5. Thank you. We will correct this in the camera-ready version.
>
> 3. Thank you for the suggestion.
>
> 4. Should have been “improved the generalization bound in \cref{robustness_bound}” referring to the one introduced by Xu & Manor.  We will correct this in the camera-ready version.
>
> **Please consider updating your score if you are satisfied with our answer. We are happy to answer any remaining questions.**

---

> > ### Comment · Reviewer_QFxN · 2025-04-04
> >
> > I thank the authors for answering my questions and comments. I am satisfied with the authors' responses.

---

### Official Review · Reviewer_LBby · 2025-03-18

**Overall Recommendation:** 3

**Summary:**

This paper studies the generalization bound for message passing network. The author extend the generalization framework of using pseudo-metric to graph, with a focus on studying what graph pseudo metric is suitable to obtain a tight bound for the generalization error of MPNNs. Specifically, the author studied tree distance for non-labeled case and tree mover's distance for labeled case, showing that fine-grained and 1-WL expressive graph distance can provide much tighter generalization bound comparing to 1-WL discrete metric. The main contribution contains several parts: 1) successfully applied the generalization framework of Xu & Mannor to graph setting, with defining graph pseudo metric. 2) based on the framework, the key part of tight the bound is to find proper metric with smaller covering number; the author have show the advantage of using tree distance and tree mover's distance. 3) the author also have done fine-grained study for different model architecture and loss function. In the experiment section, the paper shows that their bound is tight for the empirical generalization gap.

**Claims And Evidence:**

Without checking the correctness of proofs, I do think the author have made their claims supported, both theoretically and empirically.

**Essential References Not Discussed:**

I'm not aware of that. But I should also acknowledge that I'm not very familiar with generalization bound literature.
When I check the related work section in appendix, I feel the author have mentioned many recent papers of the area.

**Experimental Designs Or Analyses:**

Yes I have checked. I think the experiments, while being limited, are aligning with the theory presented in the paper.

**Methods And Evaluation Criteria:**

1. The author have designed several experiments to show that their derived bounds and insights can provide guidance in real-world graph MPNN setting.
2. Nevertheless, it would be interesting to see how sum and ordinary aggregation differ in theory and experiment.

**Other Comments Or Suggestions:**

I think the author can consider add additional experiments to cover the influence of graph structure, aggregation and loss function. Which is claimed in theory and contribution, but not empirically studied in experiment.

**Other Strengths And Weaknesses:**

### Strength
1. The author successfully applied modern generalization framework to graph setting, and linked the framework with many recent graph metrics. With good theoretical contribution, the author also confirms that the derived bound is relatively tight in empirical setting.
2. Comparing with previous literature, the usage of fine-grained graph similarity metric moves a great step forward for this GNN generalization area. If all claims and theories shown are correct, the contribution should be great for future studies.
3. The author did great job in literature review and background introduction.
4. Figures presented in the paper are very helpful for understanding equations and key ideas.

### Weakness
1.  Some part is not clear enough, and the main content is not self contained without looking at the appendix. For example, the equation in line 187 does not explain what is the definition of unr. Also, the Figure 4 referenced in line 211 are essential for understanding the presented metric, I think the author should move it to the main paper.
2. In the meantime, I think many claims and theorms can be improved with presentation, with simplifying notification and moving minor bounds to appendix. This will make the reading and understanding much better.
3. The section 4, while being the main section of the paper, lacks good structure. The author presents all findings and derivations sequentially in a single flat level. I think this make the reading harder. The author should think about how to present this section much structured and simpler.

**Questions For Authors:**

N/A

**Relation To Broader Scientific Literature:**

I think the author made a good step of linking several previous works of generalization bound, and make a forward step for MPNN's generalization bound. I value the theory contribution of it to make the generalization study of GNNs more complete.

**Theoretical Claims:**

No I have not checked. I will not provide confident review on their theorems.

---

> ### Author Rebuttal · Authors · 2025-03-27
>
> **We thank the reviewer for their detailed and constructive review.**
>
> > Some part is not clear enough, and the main content is not self contained without looking at the appendix. For example, the equation  in line 187 does not explain what is the definition of unr. Also, the Figure 4 referenced in line 211 are essential for understanding the > presented metric, I think the author should move it to the main paper.
> > In the meantime, I think many claims and theorms can be improved with presentation, with simplifying notification and moving minor > bounds to appendix. This will make the reading and understanding much better.
> > The section 4, while being the main section of the paper, lacks good structure. The author presents all findings and derivations >sequentially in a single flat level. I think this make the reading harder. The author should think about how to present this section much
> > structured and simpler.
>
> Thank you for pointing out these omissions. We will use an additional page in the camera-ready version to explain the unrolling trees in the main text and refer the reader to their definition in Appendix C.
>
> Given the extra page available, we will include an introductory paragraph in Section 4 highlighting the main results and guiding the reader through the section's flow to improve readability.
>
> > I think the author can consider adding additional experiments to cover the influence of graph structure, aggregation, and loss function, which is claimed in theory and contribution but not empirically studied in the experiments.
>
> Thank you for this suggestion, we will try our best to include these experiments to align theory and practice more.
>
> **Please consider updating your score if you are satisfied with our answer. We are happy to answer any remaining questions.**

---

### Decision · Program_Chairs · 2025-05-01

**Decision:**

Accept (spotlight poster)

**Comment:**

This paper presents a generalization analysis framework for MPGNNs by introducing new graph pseudometrics aligned with 1-WL expressivity. Building on prior robustness frameworks, the authors derive covering number-based generalization bounds that reflect structural similarities between graphs more accurately than discrete WL-based metrics. Theoretical results are complemented by empirical observations.

I recommend acceptance. The paper offers a meaningful advance in our understanding of GNN generalization, wit theoretical contributions in recent developments in graph similarity. The analysis is rigorous & the chosen metrics are well-motivated.